# Tokenisation over Bounded Alphabets is Hard

**Violeta Kastreva**,[*,†,1,2] **Philip Whittington**,[*,1] **Dennis Komm**,[1] **Tiago Pimentel**[1]
[1]ETH Zürich, [2]Sofia University "St. Kliment Ohridski"
vkastreva@uni-sofia.bg, {philip.whittington, dennis.komm, tiago.pimentel}@inf.ethz.ch

## Abstract

Recent works have shown that tokenisation is NP-complete. However, these works assume tokenisation is applied to inputs with unboundedly large alphabets—an unrealistic assumption, given that in practice tokenisers operate over fixed-size alphabets, such as bytes or Unicode-characters. We close this gap by analysing tokenisation over bounded alphabets, considering two natural variants: **bottom-up tokenisation** and **direct tokenisation**, where we must, respectively, select a sequence of merge operations or a vocabulary whose application optimally compresses a dataset. We prove that even with binary alphabets, both variants are not only NP-complete, but also APX-hard and thus admit no polynomial-time approximation scheme (unless P = NP). We further show that direct tokenisation remains NP-complete even when applied to unary alphabets. These results establish that the computational intractability of tokenisation is not an artifact of large alphabets or complex constructions, but a fundamental barrier. Overall, our results explain why current practical algorithms such as BPE and UnigramLM are heuristic, and point toward approximation algorithms being an important path going forward for tokenisation research.

## 1 Introduction

Tokenisation is the first step in most natural language processing pipelines. Given a string of characters **c**, a tokeniser maps it to a sequence of subwords **s**. Language models then operate on these subword sequences rather than the raw characters. Despite its central role, however, we still lack a comprehensive understanding of tokenisation; e.g., which properties of the produced strings of subwords **s** actually help downstream modelling. A common property to aim for is **compression** (Sennrich et al., 2016; Uzan et al., 2024; Zouhar et al., 2023b), as using shorter subword-strings to encode a dataset allows for more efficient training and inference—more data can be passed through the model with the same number of flops. While not a silver bullet (Schmidt et al., 2024; Ali et al., 2024), compression has been shown to correlate with downstream model performance (Gallé, 2019; Rust et al., 2021; Zouhar et al., 2023a; Goldman et al., 2024) and will be our work's focus.

A practical concern follows immediately: once an objective (e.g., compression) is fixed, can an optimal tokeniser be found efficiently? Popular algorithms such as BPE and UnigramLM are greedy or heuristic and need not return an optimal tokeniser for the metrics they are designed to optimise. Recent work has sharpened this picture, proving NP-completeness of finding an optimal tokeniser under a compression-style objective (Kozma and Voderholzer, 2024; Whittington et al., 2025; Lim et al., 2025). These papers, however, assume tokenisation of strings over unboundedly large alphabets. Conversely, in practice the strings we care about are typically composed of Unicode-characters or bytes, thus using bounded alphabets. Whether it is possible to efficiently find optimal tokenisers over Unicode-strings (which have an alphabet size of roughly 170,000), byte-strings (with an alphabet size of 256), or bit-strings (with an alphabet size of 2) are open questions of practical relevance.

In this paper, we first define the **$n$-ary tokenisation problem**: the problem of finding an optimal tokeniser on strings constrained to alphabets of size $n$. We examine this problem under two variants: direct and bottom-up tokenisation, where—given a dataset over an $n$-ary alphabet and a vocabulary size $K$—we must find the vocabulary (in direct tokenisation) or sequence of merges (in bottom-up tokenisation) which, when applied to the dataset, maximally compresses it. We prove that (i) both

---

*Equal contribution. † Work was done during a research internship at ETH Zürich.

direct and bottom-up binary tokenisation are APX-hard, i.e., they are not in the **polynomial-time approximation scheme** (PTAS) class and thus cannot be approximated arbitrarily well in polynomial time (unless P = NP); and that (ii) for the direct case, even unary tokenisation is NP-complete. Notably, unary and binary are the easiest of the $n$-ary tokenisation problems, and thus these hardness results also trivially extend to tokenisation problems with larger alphabets.

Our results thus indicate that the computational hardness of tokenisation is not an artifact of large alphabets or elaborate merge operations: it already appears under direct tokenisation over unary alphabets. This helps explain why practical algorithms (e.g., BPE) rely on approximations, and suggests that future work should focus on provably good approximate methods or on relaxations for this problem.

## 2 TOKENISATION

> **Our notation's colour-coding (following Whittington et al., 2025)**
>
> - Blue for raw data (i.e., characters $\mathbf{c} \in \Sigma^*$);
> - Magenta for tokeniser-specific data (i.e., subwords $\mathbf{s} \in \mathcal{S}^*$ and merges $\mathbf{m} \in \mathcal{M}^*$);
> - Orange for functions (e.g., tok).

Let $\mathbf{c} \in \Sigma^*$ be[1] a **character-string**, composed of characters $c$ from an alphabet $\Sigma$; for notational convenience, we may write one such string as $\mathbf{c} = c_1 c_2 \ldots c_{|\mathbf{c}|}$. Character-strings compose the raw text data found, say, on the web, which make up the datasets on which language models are trained. We denote one such dataset by $\mathcal{D} = \{\mathbf{c}_m\}_{m=1}^M$. Before feeding data to our models, however, we typically convert them to strings of subwords, which is the job of a tokeniser.

Formally, a tokeniser can be defined as a tuple $\langle \mathcal{S}, \mathtt{detok}, \mathtt{tok} \rangle$, composed of a vocabulary, a decoding and an encoding function. A **vocabulary** $\mathcal{S}$ is a finite set of subwords, each of which is a non-empty span of characters; we thus write $\mathcal{S} \subset \Sigma^+$. A **subword-string** is then a sequence $\mathbf{s} \in \mathcal{S}^*$ and represents a character-string via the concatenation of its subwords' characters. We say that a pair of character- and subword-strings are equivalent if

$$\mathbf{c} \overset{\circ}{=} \mathbf{s} \iff \mathbf{c} = \mathtt{concat}(\mathbf{s}), \qquad \mathtt{concat}(\mathbf{s}) = s_1 \circ s_2 \circ \cdots \circ s_{|\mathbf{s}|} \qquad (1)$$

where $\mathbf{s} = \langle s_1, s_2, \cdots, s_{|\mathbf{s}|} \rangle$, each $s_t \in \mathcal{S}$ is a subword, and operator $\circ$ denotes string concatenation. Notably, $\Sigma \subseteq \mathcal{S}$ is typically enforced to guarantee that every $\mathbf{c} \in \Sigma^*$ can be represented by at least one subword-string $\mathbf{s} \in \mathcal{S}^*$, and we say that a vocabulary's size is $|\mathcal{S}| = |\Sigma| + K$. Second in the tuple above, a **decoding function** is defined as $\mathtt{detok} \colon \mathcal{S}^* \to \Sigma^*$, and given a subword-string it outputs the character-string it represents. This function thus is simply defined as $\mathtt{detok}(\mathbf{s}) \overset{\text{def}}{=} \mathtt{concat}(\mathbf{s})$.

Finally, an **encoding function** $\mathtt{tok} \colon \Sigma^* \to \mathcal{S}^*$ maps character- to subword-strings while ensuring the equivalence $\mathbf{c} \overset{\circ}{=} \mathbf{s}$ for $\mathbf{s} = \mathtt{tok}(\mathbf{c})$. Several encoding functions may respect this constraint, as many subword-strings may be equivalent to a specific character-string. For instance, given $\mathcal{S} = \{a, aa, aaa\}$, the string $\mathbf{c} = aaa$ could be tokenised as $\mathbf{s} = \langle aaa \rangle$ or as $\mathbf{s} = \langle a, aa \rangle$. We focus on two encoding functions in this paper, which we follow Whittington et al. (2025) in labelling as direct and bottom-up. The **direct encoding function** ($\mathtt{tok}_\diamond$) only requires a vocabulary, which it applies optimally to encode a character-string. In turn, the **bottom-up encoding function** ($\mathtt{tok}_\uparrow$) takes a merge sequence $\mathbf{m} = \langle m_1, \ldots, m_K \rangle$ as input, which it applies in order to a character-string; each of these merges $m_k$ is composed of a pair of subwords, which we represent as $s_k^{[1]} \circledcirc s_k^{[2]}$, and we write $\mathtt{merge}_m \colon \mathcal{S}^* \to \mathcal{S}^*$ to represent a function which, given a subword-string, processes it left-to-right and replaces any consecutive occurrence of the pair $s_k^{[1]}, s_k^{[2]}$ with a new token $s_k^{[\text{new}]} = s_k^{[1]} \circ s_k^{[2]}$. Defining $\mathcal{M} \overset{\text{def}}{=} \mathcal{S} \times \mathcal{S}$, we say $m \in \mathcal{M}$ and $\mathbf{m} \in \mathcal{M}^*$; we formalise the encoding functions as

$$\mathtt{tok}_\diamond[\mathcal{S}](\mathbf{c}) \overset{\text{def}}{=} \underset{\substack{\mathbf{s} \in \mathcal{S}^* \\ \text{s.t. } \mathbf{c} \overset{\circ}{=} \mathbf{s}}}{\arg \min} |\mathbf{s}|, \qquad \mathtt{tok}_\uparrow[\mathbf{m}](\mathbf{c}) \overset{\text{def}}{=} \left( \underset{m \in \mathbf{m}}{\bigodot} \mathtt{merge}_m \right)(\mathbf{c}) \qquad (2)$$

where $\bigodot$ represents function composition. A tokeniser is thus fully determined by a vocabulary or merge-sequence; for the direct case we have $\mathtt{tok} \overset{\text{def}}{=} \mathtt{tok}_\diamond[\mathcal{S}]$, while for bottom-up we have

---

[1] We note that $\Sigma^*$ denotes the Kleene star of $\Sigma$ (i.e., $\cup_{i=0}^\infty \Sigma^i$), and $\Sigma^+$ denotes its Kleene plus (i.e., $\cup_{i=1}^\infty \Sigma^i$).

$\mathtt{tok}\overset{\text{def}}{=}\mathtt{tok}_\uparrow[\mathbf{m}]$. Importantly, the direct encoding function ($\mathtt{tok}_\phi[\mathcal{S}]$) can be efficiently computed in $O(|\mathbf{c}|^2)$ time using the methods from Schmidt et al. (2024).

## 2.1 Objective Functions and their Optimisation

As described above, a direct tokeniser is fully determined by a vocabulary, while a bottom-up tokeniser is identified by a merge-sequence. How to select a specific tokeniser, though? This is typically done via defining an objective function $\mathfrak{G}$ which, given an encoding function ($\mathtt{tok}$) and a dataset ($\mathcal{D}$), returns a value representing the cost of that particular choice. Choosing a tokeniser then "simply" requires optimising this objective: e.g., for direct tokenisation we must find $\mathcal{S}_{\mathtt{opt}} = \arg\min_{\mathcal{S}\subset\Sigma^+} \mathfrak{G}(\mathtt{tok}_\phi[\mathcal{S}], \mathcal{D})$ under the constraint that $|\mathcal{S}| = |\Sigma| + K$.

Several objective functions exist. UnigramLM (Kudo, 2018), for instance, selects a vocabulary that optimises a dataset's unigram negative log-probability. Other work has proposed alternative measures, such as the frequency of the 5-th % least frequent token (Gowda and May, 2020), or the tokeniser's Rényi efficiency (Zouhar et al., 2023a). As mentioned above, we focus on compression in this paper. We do so following a battery of previous work which formally analyses tokenisers (Zouhar et al., 2023b; Kozma and Voderholzer, 2024; Whittington et al., 2025; Lim et al., 2025). Prior work has shown that a tokeniser's compression correlates with the downstream performance of language models trained on its output subword-strings (Gallé, 2019; Zouhar et al., 2023a). We note, however, that other recent work has criticised compression as the sole objective for tokenisation, showing that these two properties (compression and downstream performance) may have a more complex relationship than originally suspected (Ali et al., 2024; Schmidt et al., 2024).

There are two natural ways to define a compression objective: **compressed length**, which measures the number of remaining symbols after a string is tokenised, and **compression reduction**, which measures how many symbols are reduced in the string by a tokeniser. These are formalised as:

$$\underbrace{\mathfrak{G}_\ell(\mathtt{tok}, \mathcal{D}) \overset{\text{def}}{=} \sum_{\mathbf{c}\in\mathcal{D}} |\mathtt{tok}(\mathbf{c})|,}_{\textbf{compressed length, } \text{size of remaining string}} \qquad \underbrace{\mathfrak{G}_r(\mathtt{tok}, \mathcal{D}) \overset{\text{def}}{=} \sum_{\mathbf{c}\in\mathcal{D}} \Big( |\mathtt{tok}(\mathbf{c})| - |\mathbf{c}| \Big)}_{\textbf{compression reduction, } \text{number of reduced symbols}} \qquad (3)$$

where $|\mathtt{tok}(\mathbf{c})|$ denotes the length of the subword-sequence produced by the tokeniser for string $\mathbf{c}$. While equivalent in how they rank tokenisers, this choice can make a big difference when evaluating the quality of an approximation. When using minimisation objectives, such as $\mathfrak{G}_\ell$, the **approximation ratio** of an algorithm upper-bounds the ratio between the objective value achieved by the algorithm's solution and an optimal solution, being thus at least 1 by definition. A similar definition applies when using maximisation objectives, such as $\mathfrak{G}_r$, but the approximation ratio is inverted. We say we have a **$\rho$-approximation algorithm** if, for every possible input, this ratio is bounded from above by $\rho$. If a dataset has 1,000 characters and would have 100 symbols if optimally compressed, a suboptimal tokeniser which instead reduces it to at most 200 symbols would have an approximation ratio of 2 under $\mathfrak{G}_\ell$ but of 1.125 under $\mathfrak{G}_r$. Notably, prior work has analysed both these measures. We argue here that compressed length is the more natural objective, as it directly relates to the throughput achieved by a language model processing the given text, being thus connected to the model's training and inference costs. A 2-approximation for $\mathfrak{G}_\ell$ implies that a language model using that tokeniser may take twice as long as optimal when processing the same text.[2]

After deciding on an objective function, such as $\mathfrak{G}_\ell$, we must select a vocabulary ($\mathcal{S}\subset\Sigma^+$) or merge-sequence ($\mathbf{m}\in\mathcal{M}^*$) which optimises it. Unfortunately, both these optimisation problems have infinite search spaces (respectively, $\mathcal{P}(\Sigma^+)$ and $\mathcal{M}^*$, where $\mathcal{P}$ denotes the powerset operation), which begs the question: is there an efficient way to find these optima? Recent work has shown that, in general, this is not possible, proving compression-based tokenisation to be NP-complete; more specifically, Kozma and Voderholzer (2024) showed this for bottom-up tokenisation, Whittington et al. (2025) for direct and bottom-up tokenisation, and Lim et al. (2025) for direct tokenisation with candidate tokens. This means that, unless P = NP, there exists no polynomial-time algorithm to find compression-optimal tokenisers. Beyond that, using the $\mathfrak{G}_r$ objective function, Kozma and Voderholzer (2024) showed that bottom-up tokenisation is not only NP-complete but also APX-hard, which implies that it is not in the polynomial-time approximation scheme (PTAS) complexity class (unless P = NP). A problem is in the PTAS class if, for every constant $\varepsilon > 0$, there exists a polynomial-time

---

[2]Assuming that language models cannot achieve sub-linear computational complexity on their input's length.

algorithm (whose time complexity may depend on $\varepsilon$), which solves it with an approximation ratio upper-bounded by $1 + \varepsilon$. Not being in PTAS thus implies that there is no polynomial-time algorithm which can approximate the optimal solution arbitrarily well. Notably, all of the above complexity proofs apply to tokenisation problems over alphabets of arbitrarily large size. We show that these results hold even if the alphabet size is bounded by a constant, which was left open by prior work.

## 2.2 A Note on Decision vs. Optimisation Problems

In this paper, we will discuss two broad types of tokenisation problems: decision and optimisation versions. A **decision problem** is typically framed as a true–false existence question: e.g., does a tokeniser (within a certain class $\mathcal{T}_K$) exist which achieves a minimum target compression $\delta$ on a dataset $\mathcal{D}$? Alternatively, an **optimisation problem** asks what is the optimal solution to a problem: e.g., what is the optimal compression achieved by any tokeniser within $\mathcal{T}_K$ on this dataset? More formally:

$$\texttt{decision}: \delta \overset{?}{\geq} \min_{\texttt{tok} \in \mathcal{T}_K} \mathfrak{G}_\ell(\texttt{tok}, \mathcal{D}), \qquad \texttt{optimisation}: \delta_{\text{opt}} = \min_{\texttt{tok} \in \mathcal{T}_K} \mathfrak{G}_\ell(\texttt{tok}, \mathcal{D}) \qquad (4)$$

Typically, NP-hardness is discussed mainly as a property of decision problems, while hardness of approximation (and consequently, being contained in PTAS or not) is a property of optimisation problems. There is, however, a notion of equivalence between these classes of problems: if a polynomial-time algorithm exists to solve a decision problem, it can usually be leveraged to also find an efficient algorithm for its associated optimisation problem, and *vice-versa*. Similarly, if no polynomial-time algorithm can solve an optimisation problem with an approximation ratio arbitrarily close to 1 (i.e., if the problem is not in PTAS), this implies that there must be some **hardness-of-approximation constant** $\varepsilon$ such that it is NP-hard to distinguish between instances that admit a solution with cost $x$ and those that admit a solution with cost $(1 + \varepsilon)x$.

Our paper will focus on showing three main results for the analysed tokenisation problems. First, we will show that these problems are NP-hard. This is done by proving that another NP-hard problem can be **reduced** to tokenisation: i.e., there exists a **reduction function** that receives an instance of this other problem and, in polynomial time, outputs an instance of the tokenisation problem with the same truth condition. Second, we will show that a large subset of these problems is APX-hard. This can be done by showing that another APX-hard problem can be **L-reduced** to tokenisation: i.e., beyond a reduction function, there also exists a **reconstruction function** that receives a solution of the tokenisation problem (i.e., a tokeniser) and, in polynomial time, outputs a solution of this other APX-hard problem within a certain quality bound (see App. A for a formal description).

Finally, we will also compute an explicit hardness-of-approximation constant $\varepsilon$ for these problems, relying on **gap-preserving reductions**. To this end, it will be useful to also define gap versions of the problems we discuss. Formally, we will define such gap versions similarly to their decision versions, but while providing two decision boundaries, i.e., $(\delta^-, \delta^+)$ instead of simply $\delta$. In minimisation gap problems, the task is then to decide whether their optimal value is at most $\delta^+$ or at least $\delta^-$, with $\delta^+ \leq \delta^-$; if a value falls between these, any answer is acceptable. For tokenisation problems, for instance, we would require an algorithm which returns: (i) true, if $\delta^+ \geq \min_{\texttt{tok} \in \mathcal{T}} \mathfrak{G}_\ell(\texttt{tok}, \mathcal{D})$, (ii) false, if $\delta^- \leq \min_{\texttt{tok} \in \mathcal{T}} \mathfrak{G}_\ell(\texttt{tok}, \mathcal{D})$, and (iii) anything, if neither condition holds.

## 3 Tokenisation over Bounded Alphabets

We now move to the analysis of tokenisation over bounded alphabets. Let an **$n$-ary alphabet** be an alphabet of size $|\Sigma| = n$. We define the tokenisation problem over such bounded alphabets as follows.

**Definition 1.** *Let $K$ be a vocabulary size and $\mathcal{D}$ be a dataset composed of character-strings from an $n$-ary alphabet $|\Sigma|$. For a given $\delta$, the $n$-ary tokenisation decision problem requires deciding whether there exists a vocabulary $\mathcal{S}_{\text{opt}} \subseteq \Sigma^+$ (for direct tokenisation) or a merge-sequence $\mathbf{m}_{\text{opt}} \in \mathcal{M}^*$ (for bottom-up tokenisation) which compresses $\mathcal{D}$ to at most $\delta$ symbols. The $n$-ary tokenisation optimisation problem is to find what the maximal such compression of $\mathcal{D}$ is. Defining $\mathcal{T}_K \overset{\text{def}}{=} \{\texttt{tok}_\phi[\mathcal{S}] \mid \mathcal{S} \subset \Sigma^+, |\mathcal{S}| = |\Sigma| + K\}$ for direct tokenisation or $\mathcal{T}_K \overset{\text{def}}{=} \{\texttt{tok}_\uparrow[\mathbf{m}] \mid \mathbf{m} \in \mathcal{M}^*, |\mathbf{m}| = K\}$ for bottom-up:*

$$\underbrace{\delta \overset{?}{\geq} \min_{\texttt{tok} \in \mathcal{T}_K} \mathfrak{G}_\ell(\texttt{tok}, \mathcal{D})}_{n\text{-ary tokenisation decision problem}} , \qquad \underbrace{\delta_{\text{opt}} = \min_{\texttt{tok} \in \mathcal{T}_K} \mathfrak{G}_\ell(\texttt{tok}, \mathcal{D})}_{n\text{-ary tokenisation optimisation problem}} \qquad (5)$$

We will more specifically call these the $n$-ary direct tokenisation problem and the $n$-ary bottom-up tokenisation problem when dealing with, respectively, direct and bottom-up tokenisers, writing $\text{Tok}_\leftrightarrow^n(\mathcal{D}, K, \delta)$ and $\text{Tok}_\uparrow^n(\mathcal{D}, K, \delta)$ for the functions which return the solution to their decision problems, and $\text{Tok}_\leftrightarrow^{n,\star}(\mathcal{D}, K)$ and $\text{Tok}_\uparrow^{n,\star}(\mathcal{D}, K)$ for functions which return the solution of their optimisation problems. Finally, we define the tokenisation gap problem as:

$$\text{Tok}(\mathcal{D}, K, (\delta^-, \delta^+)) = \begin{cases} \texttt{T} & \text{if} \;\; \delta^+ \geq \min_{\texttt{tok} \in \mathcal{T}_K} \mathfrak{G}_\ell(\texttt{tok}, \mathcal{D}) \\ \texttt{F} & \text{elif} \;\; \delta^- \leq \min_{\texttt{tok} \in \mathcal{T}_K} \mathfrak{G}_\ell(\texttt{tok}, \mathcal{D}) \\ \texttt{?} & \text{else} \end{cases} \tag{6}$$

Notably, the $n$-ary tokenisation problems form a clear hierarchy from easiest ($n = 1$) to hardest ($n \to \infty$), with unary tokenisation being the easiest such problem. In the next sections, we first prove that both binary direct and binary bottom-up tokenisation are APX-hard (in §4). We then prove that unary direct tokenisation is NP-complete (in §5).

**Fact 1.** *If $n$-ary tokenisation is hard, all $n'$-ary tokenisation problems for $n' > n$ are hard.*

*Proof.* Let $n, n' \in \mathbb{N}$ with $n' \geq n$. Any instance of the $n$-ary tokenisation problem is a valid instance of the $n'$-ary problem with the same solutions, allowing for a trivial reduction between them. Thus, any proof of hardness for the $n$-ary tokenisation problem immediately applies to $n'$-ary problems. $\quad\square$

## 4 Binary Tokenisation is Hard to Decide and Approximate

In this section, we will prove NP-hardness and APX-hardness of the two binary tokenisation decision problems above, as well as NP-hardness of their corresponding gap problems (for specific gaps). To this end, we will use a reduction from the **3-occurrence maximum 2-satisfiability** problem (3-OCC-MAX2SAT), which we define in §4.1. We then move on to proving the hardness of the binary direct and binary bottom-up tokenisation problems (in §4.2 and §4.3, respectively).

### 4.1 3-Occurrence Maximum 2-Satisfiability

Let $X$ be a Boolean variable assigned a value $x \in \{\texttt{F}, \texttt{T}\}$, and let $\mathcal{X} = \{X_j\}_{j=1}^J$ be a set of such variables, with joint assignment $\chi = \{x_j\}_{j=1}^J$. Further, let $\mathcal{C} = \{(L_i^1 \vee L_i^2)\}_{i=1}^I$ be a set of clauses,[3] where each literal $L_i$ is either a variable $X_j$ or its negation $\neg X_j$. We define 3-OCC-MAX2SAT as follows.

**Definition 2.** *Let $\mathcal{X} = \{X_j\}_{j=1}^J$ be a set of Boolean variables and $\mathcal{C} = \{(L_i^1 \vee L_i^2)\}_{i=1}^I$ be a set of clauses. Further, let each variable $X_j$ occur in exactly three clauses. Given a target $\gamma \in \mathbb{N}$, the* **3-OCC-MAX2SAT** *decision problem requires deciding whether there exists an assignment $\chi \in \{\texttt{F}, \texttt{T}\}^J$ such that at least $\gamma$ clauses are satisfied. The* **3-OCC-MAX2SAT** *optimisation problem requires finding the maximum number of satisfiable clauses. Formally:*

$$\underbrace{\gamma \overset{?}{\leq} \max_{\chi \in \{\texttt{F},\texttt{T}\}^J} \sum_{i=1}^I \mathbb{1}_\chi \{L_i^1 \vee L_i^2\}}_{\text{3-OCC-MAX2SAT decision problem}} \qquad\qquad \underbrace{\gamma_{\texttt{opt}} = \max_{\chi \in \{\texttt{F},\texttt{T}\}^J} \sum_{i=1}^I \mathbb{1}_\chi \{L_i^1 \vee L_i^2\}}_{\text{3-OCC-MAX2SAT optimisation problem}} \tag{7}$$

We write $\text{3OM2S}(\mathcal{X}, \mathcal{C}, \gamma)$ to denote a function that, given an instance of the 3-OCC-MAX2SAT decision problem, returns its solution. Similarly, we write $\text{3OM2S}^\star(\mathcal{X}, \mathcal{C})$ for a function that returns $\gamma_{\texttt{opt}}$. The 3-OCC-MAX2SAT problem was proven to be APX-hard by Berman and Karpinski (1999), with their result also implying that this problem is NP-hard.

### 4.2 Binary Direct Tokenisation is Hard to Decide and Approximate

In this section, we prove that the binary direct tokenisation problem is both hard to decide and to approximate beyond a certain constant greater than $1$. First, we will prove that the *decision* version is NP-hard (in §4.2.1). Second, we will use this initial result to prove both that this problem's optimisation version is APX-hard and that a *gap* version of the problem is similarly NP-hard (in §4.2.2). This will complete our proof that this problem's *optimisation* version is hard to approximate arbitrarily well, while providing an explicit hardness-of-approximation constant.

---

[3]In some formalisations, 3-OCC-MAX2SAT allows clauses of size one. We work here, more specifically, with the 3-occurrence maximum exact-2-satisfiability variant of this problem, thus not allowing single-literal clauses.

### 4.2.1 THE BINARY DIRECT TOKENISATION DECISION PROBLEM IS NP-HARD

We now prove NP-completeness of binary direct tokenisation, which requires two things: inclusion in NP and being NP-hard. Inclusion in NP follows from the general (unbounded) case, which was previously proven by Whittington et al. (2025). Proving NP-hardness requires a polynomial-time reduction from another NP-hard problem to this problem, which we will design in what follows.

**Reduction 1.** *Consider an instance of the* `3-OCC-MAX2SAT` *decision problem and the binary alphabet* $\Sigma = \{0, 1\}$. *Now, for each variable* $X_j$, *let* $\mathbf{x}_j^{\mathrm{T}} = 0^{2j-1}$ *and* $\mathbf{x}_j^{\mathrm{F}} = 0^{2j}$, *i.e., these are character-strings formed of* $0$ *repeated* $2j - 1$ *or* $2j$ *times. Then we build subdatasets:*

$$\mathcal{D}_1 = \{1\mathbf{x}_j^{\mathrm{T}},\ \mathbf{x}_j^{\mathrm{T}}1,\ 1\mathbf{x}_j^{\mathrm{F}},\ \mathbf{x}_j^{\mathrm{F}}1 \mid 1 \le j \le J\} \ \times f \quad \mathcal{D}_2 = \{1\mathbf{x}_j^{\mathrm{T}}1,\ 1\mathbf{x}_j^{\mathrm{F}}1 \mid 1 \le j \le J\} \ \times f' \quad \text{(8a)}$$

$$\mathcal{D}_3 = \{1\mathbf{x}_j^{\mathrm{T}}1\mathbf{x}_j^{\mathrm{F}}1 \mid 1 \le j \le J\} \hspace{2.2cm} \times f'' \quad \mathcal{D}_4 = \{1L_i^1 1 L_i^2 1 \mid 1 \le i \le I\} \hspace{1.1cm} \times 1 \quad \text{(8b)}$$

*where* $L_i^1$ *and* $L_i^2$ *are replaced by their respective variable characters as they appear in the* $i$-th *clause (i.e.,* $L_i$ *is replaced by* $\mathbf{x}_j^{\mathrm{T}}$ *if it is equal to* $X_j$ *or by* $\mathbf{x}_j^{\mathrm{F}}$ *if it equals* $\neg X_j$*). Further,* $\times f$ *denotes that a set of strings should be repeated* $f$ *times in the corresponding dataset. These multiplicities are* $f'' \stackrel{\text{def}}{=} 7$, $f' \stackrel{\text{def}}{=} 2(f'' + 3) + 1 = 21$, *and* $f \stackrel{\text{def}}{=} 2(f' + f'' + 3) + 1 = 63$. *A full dataset is then formed by joining these subdatasets:* $\mathcal{D} = \mathcal{D}_1 \cup \mathcal{D}_2 \cup \mathcal{D}_3 \cup \mathcal{D}_4$. *Finally, we set the number of allowed tokens to* $K = 5J$ *and the target compression to* $\delta = 4fJ + 3f'J + 2f''J + 3I - \gamma = 329J + 3I - \gamma$.[4]

We write $\mathrm{R1}(\mathcal{X}, \mathcal{C}, \gamma)$ to represent the `D-2-TOK` instance that is output by this reduction, represented by the tuple $(\mathcal{D}, K, \delta)$. Notably, this reduction runs in polynomial time. By proving its correctness, thus, we prove that binary direct tokenisation is NP-hard. For this reduction to be correct, the given `3-OCC-MAX2SAT` instance must be satisfiable if and only if its reduced tokenisation instance is as well, i.e., $3\mathrm{OM2S}(\mathcal{X}, \mathcal{C}, \gamma) \iff \mathrm{Tok}_\delta^2(\mathrm{R1}(\mathcal{X}, \mathcal{C}, \gamma))$. We now prove both directions of this iff clause.

**Theorem 1.** *The binary direct tokenisation decision problem is* NP-*complete.*

*Proof sketch.* This proof is done in two steps.

**Forward step** ($3\mathrm{OM2S}(\mathcal{X}, \mathcal{C}, \gamma) \implies \mathrm{Tok}_\delta^2(\mathrm{R1}(\mathcal{X}, \mathcal{C}, \gamma))$)**.** See a formal proof in Lemma 1 in App. B. Assuming an instance of `3-OCC-MAX2SAT` is satisfied by assignment $\chi^\star = \{x_j^\star\}_{j=1}^J$, we build a direct tokeniser with tokens $1\mathbf{x}_j^{\mathrm{T}}, \mathbf{x}_j^{\mathrm{T}}1, 1\mathbf{x}_j^{\mathrm{F}}, \mathbf{x}_j^{\mathrm{F}}1$, and with token $1\mathbf{x}_j^{\mathrm{T}}1$ if $x_j^\star = \mathtt{T}$, and $1\mathbf{x}_j^{\mathrm{F}}1$ otherwise. This tokeniser compresses $\mathcal{D}_1$ to $252J$, $\mathcal{D}_2$ to $63J$, $\mathcal{D}_3$ to $14J$, and $\mathcal{D}_4$ to $3I - \gamma^\star$ tokens, where $\gamma^\star$ is the number of clauses satisfied by $\chi^\star$. Adding these compressed lengths together, we find that they satisfy the direct tokenisation problem, as $\gamma^\star \ge \gamma$ by assumption.

**Backward step** ($\mathrm{Tok}_\delta^2(\mathrm{R1}(\mathcal{X}, \mathcal{C}, \gamma)) \implies 3\mathrm{OM2S}(\mathcal{X}, \mathcal{C}, \gamma)$)**.** See full proof in Lemma 2 in App. C. We first show that an optimal tokeniser for the `D-2-TOK` instance is always **sat-compliant**:[5] it contains all tokens of the form $1\mathbf{x}_j^{\mathrm{T}}, \mathbf{x}_j^{\mathrm{T}}1, 1\mathbf{x}_j^{\mathrm{F}}, \mathbf{x}_j^{\mathrm{F}}1$, and either $1\mathbf{x}_j^{\mathrm{T}}1$ or $1\mathbf{x}_j^{\mathrm{F}}1$ for each $j \in \{1, \ldots, J\}$. We do this by showing that $\mathcal{D}_1$ guarantees that any optimal solution includes tokens $1\mathbf{x}_j^{\mathrm{T}}, \mathbf{x}_j^{\mathrm{T}}1, 1\mathbf{x}_j^{\mathrm{F}}, \mathbf{x}_j^{\mathrm{F}}1$; $\mathcal{D}_2$ guarantees that any optimal solution further only includes tokens of the form $1\mathbf{x}_j^{\mathrm{T}}1, 1\mathbf{x}_j^{\mathrm{F}}1$; and $\mathcal{D}_3$ guarantees that either token $1\mathbf{x}_j^{\mathrm{T}}1$ or $1\mathbf{x}_j^{\mathrm{F}}1$ exist for each $j \in \{1, \ldots, J\}$. Then, we show that if such a sat-compliant tokeniser reaches the desired compression, it must correspond to an assignment $\chi^\star$ that satisfies the desired number of clauses. □

### 4.2.2 BINARY DIRECT TOKENISATION IS HARD TO APPROXIMATE

We now prove that the binary direct tokenisation problem is not only NP-hard, but APX-hard, implying that this problem is not in PTAS (unless P = NP). Further, we also prove that its gap version is NP-hard, computing a hardness-of-approximation constant for this problem in the process.

**Theorem 2.** *The binary direct tokenisation optimisation problem is* APX-*hard and cannot be approximated in polynomial time with an approximation ratio* $\rho < 1.000002$, *unless* P = NP.

*Proof sketch.* This proof is done in two steps.

---

[4]This reduction is inspired by Whittington et al.'s (2025) reduction, which we update to (i) rely on binary, as opposed to unbounded, alphabets; (ii) use constant-sized $f$'s, which allow us to prove approximation hardness.

[5]This concept is inspired by Kozma and Voderholzer's (2024) use of well-formed tokenisers.

**The optimisation problem is** APX-**hard.** See a formal proof in Lemma 3 in App. D. Our proof rests on an L-reduction from 3-OCC-MAX2SAT, which was shown to be APX-hard by Berman and Karpinski (1999).We again reduce from this problem to D-2-TOK using Reduction 1. To define a reconstruction function, we consider two cases: a solution to D-2-TOK (i.e., a tokeniser) will be either sat-compliant or not. Given a sat-compliant tokeniser, it is easy to build a solution for 3-OCC-MAX2SAT with bounded quality. Given a sat-noncompliant tokeniser, we first transform it into sat-compliant via quality-improving transformations; we then use this sat-compliant tokeniser to build a solution for 3-OCC-MAX2SAT with bounded quality. This concludes this step of the proof.

**The gap problem is** NP-**hard.** See a formal proof in Lemma 4 in App. E. As shown by Berman and Karpinski (1998; 1999), the 3-OCC-MAX2SAT gap problem is NP-hard for instances with $I = 2016n$ clauses, $\gamma^- = (2011 + \varepsilon)n$ lower bound, and $\gamma^+ = (2012 - \varepsilon)n$ upper bound. We use Lemmas 1 and 2 to prove a reduction from this gap problem to D-2-TOK's gap problem. Notably, our reduction sets $\delta = 329J + 3I - \gamma$, for both $\gamma^-$ and $\gamma^+$. Analysing the gap of the resulting tokenisation problem, we find that this problem is thus NP-hard for a gap $\delta^-/\delta^+$ of 1.000002. □

While the constant above (i.e., 1.000002) is remarkably small, we note that our proof makes no attempt to optimise this bound. Our lemma's main takeaway is that it is not possible to compute D-2-TOK with approximation ratios arbitrarily close to 1 in polynomial time. Other larger bounds likely exist and, in fact, it might even be possible that there is no constant-factor approximation for D-2-TOK at all. Note that whether D-2-TOK admits any polynomial-time constant-factor approximation algorithm (i.e., whether it lies in APX, meaning it would be APX-complete) remains an open problem.

### 4.3 BINARY BOTTOM-UP TOKENISATION IS HARD TO DECIDE AND APPROXIMATE

As for the direct case, our argument proceeds by first proving the hardness of the *decision* version, and then leveraging this result to show APX-hardness of the *optimisation* version as well as an explicit hardness-of-approximation constant via the gap problem.

#### 4.3.1 THE BINARY BOTTOM-UP TOKENISATION PROBLEM IS NP-COMPLETE

As before, we use a reduction from 3-OCC-MAX2SAT to prove NP-hardness.

**Reduction 2.** *Consider an instance of the* 3-OCC-MAX2SAT *decision problem and the binary alphabet* $\Sigma = \{0, 1\}$. *Again, for each variable* $X_j$, *let* $\mathbf{x}_j^\mathsf{T} = 0^{2j-1}$ *and* $\mathbf{x}_j^\mathsf{F} = 0^{2j}$. *Then we build subdatasets:*

$$\mathcal{D}_1 = \{11\} \cup \{\mathbf{x}_j^\mathsf{T}, \mathbf{x}_j^\mathsf{F}, 1\mathbf{x}_j^\mathsf{T}, \mathbf{x}_j^\mathsf{T}1, 1\mathbf{x}_j^\mathsf{F}, \mathbf{x}_j^\mathsf{F}1, \mathbf{x}_j^\mathsf{T}11, 11\mathbf{x}_j^\mathsf{F}\}_{j=1}^J \qquad \times f \tag{9a}$$

$$\mathcal{D}_2 = \{1\mathbf{x}_j^\mathsf{T}1, 1\mathbf{x}_j^\mathsf{F}1, 1\mathbf{x}_j^\mathsf{T}11, 11\mathbf{x}_j^\mathsf{F}1\}_{j=1}^J \qquad \times f' \tag{9b}$$

$$\mathcal{D}_3 = \{1\mathbf{x}_j^\mathsf{T}1\mathbf{x}_j^\mathsf{F}1, 11\mathbf{x}_j^\mathsf{F}1\mathbf{x}_j^\mathsf{T}11\}_{j=1}^J \qquad \times f'' \tag{9c}$$

$$\mathcal{D}_4 = \{1\mathbf{x}_j^\mathsf{F}1\mathbf{x}_j^\mathsf{T}11, 11\mathbf{x}_j^\mathsf{F}1\mathbf{x}_j^\mathsf{T}1\}_{j=1}^J \qquad \times f''' \tag{9d}$$

$$\mathcal{D}_5 = \left\{ \begin{array}{ll} 1\mathbf{x}_j^\mathsf{T}1\mathbf{x}_{j'}^\mathsf{F}1 & \texttt{if } L_i^1 = X_j,\ L_i^2 = \neg X_{j'} \\ 1\mathbf{x}_{j'}^\mathsf{T}1\mathbf{x}_j^\mathsf{F}1 & \texttt{if } L_i^1 = \neg X_j,\ L_i^2 = X_{j'} \\ 11\mathbf{x}_j^\mathsf{F}1\mathbf{x}_{j'}^\mathsf{F}1 & \texttt{if } L_i^1 = \neg X_j,\ L_i^2 = \neg X_{j'} \\ 1\mathbf{x}_j^\mathsf{T}1\mathbf{x}_{j'}^\mathsf{T}11 & \texttt{if } L_i^1 = X_j,\ L_i^2 = X_{j'} \end{array} \right\}_{i=1}^I \qquad \times 1 \tag{9e}$$

*The subdataset multiplicities are* $f''' \stackrel{\text{def}}{=} 4$, $f'' \stackrel{\text{def}}{=} 2(2f'''+3)+1 = 23$, $f' \stackrel{\text{def}}{=} 2(2f''+2f'''+3)+1 = 115$, *and* $f \stackrel{\text{def}}{=} 2(2f' + 2f'' + 2f''' + 3)+1 = 575$. *We set the vocabulary size to* $K = 10J$ *and the target compression length to* $\delta = (8J+1)f+6Jf'+4Jf''+4Jf'''+3I-\gamma = 5398J+575+3I-\gamma$.

We write $\text{R2}(\mathcal{X}, \mathcal{C}, \gamma)$ to represent the B-2-TOK instance $(\mathcal{D}, K, \delta)$ constructed by this reduction. As before, this is a polynomial-time reduction. We now prove the equivalence $\text{3OM2S}(\mathcal{X}, \mathcal{C}, \gamma) \iff \text{Tok}_\uparrow^2(\text{R2}(\mathcal{X}, \mathcal{C}, \gamma))$ which shows the reduction's correctness and thus that B-2-TOK is NP-hard.

**Theorem 3.** *The binary bottom-up tokenisation decision problem is* NP-*complete.*

*Proof sketch.* This proof is done in two steps.

**Forward step** ($\text{3OM2S}(\mathcal{X}, \mathcal{C}, \gamma) \implies \text{Tok}_\uparrow^2(\text{R1}(\mathcal{X}, \mathcal{C}, \gamma))$). See full proof in Lemma 5 in App. F. Assume the 3-OCC-MAX2SAT instance admits an assignment $\chi^\star = \{x_j^\star\}_{j=1}^J$ satisfying

at least $\gamma$ clauses. We construct a merge sequence $\mathbf{m} = \mathbf{m}_1 \circ \mathbf{m}_2 \circ \mathbf{m}_3 \circ \mathbf{m}_4 \circ \mathbf{m}_5 \circ \mathbf{m}_6$, where $\mathbf{m}_1, \mathbf{m}_2, \mathbf{m}_4, \mathbf{m}_6$ are **structural merges** that appear in every valid tokeniser solution and ensure that all strings corresponding to single variables are properly compressed; and $\mathbf{m}_3, \mathbf{m}_5$ are **assignment-dependent merges**, chosen according to $\chi^\star$: for each variable $x_j^\star$, we merge $1 \odot \mathbf{x}_j^\mathsf{T} 11, 1\mathbf{x}_j^\mathsf{T} \odot 1$ if $x_j^\star = \mathsf{T}$, and $11\mathbf{x}_j^\mathsf{F} \odot 1, 1 \odot \mathbf{x}_j^\mathsf{F} 1$ otherwise. Applying $\mathbf{m}$ to the string subdatasets $\mathcal{D}_1, \mathcal{D}_2, \mathcal{D}_3, \mathcal{D}_4$ gives the fixed compressed length $5398J + 575$. For the strings $\mathcal{D}_5$, the construction ensures that each clause compresses to 2 tokens if at least one of its two literals is true under $\chi^\star$, and remains at 3 tokens otherwise. Since $\chi^\star$ satisfies at least $\gamma$ clauses, we obtain at most $3I - \gamma$ symbols. Compression thus satisfies the budget $\delta$.

**Backward step** ($\mathrm{Tok}_\uparrow^2(\mathrm{R2}(\mathcal{X}, \mathcal{C}, \gamma)) \implies \mathrm{3OM2S}(\mathcal{X}, \mathcal{C}, \gamma)$)**.** See full proof in Lemma 6 in App. G. We consider sat-compliant *direct* tokenisers, which must contain all tokens of the form $11, \mathbf{x}_j^\mathsf{T}, \mathbf{x}_j^\mathsf{F}, 1\mathbf{x}_j^\mathsf{T}, \mathbf{x}_j^\mathsf{T}1, 1\mathbf{x}_j^\mathsf{F}, \mathbf{x}_j^\mathsf{F}1, \mathbf{x}_j^\mathsf{T}11, 11\mathbf{x}_j^\mathsf{F}$ and must contain either $1\mathbf{x}_j^\mathsf{T}1, 1\mathbf{x}_j^\mathsf{T}11$ or $1\mathbf{x}_j^\mathsf{F}1, 11\mathbf{x}_j^\mathsf{F}1$ for each $j \in \{1, \ldots, J\}$. Again, an optimal direct tokeniser for the B-2-TOK-instance is always sat-compliant, which is enforced by datasets $\mathcal{D}_1$ to $\mathcal{D}_4$. We then show that if such a tokeniser achieves the desired compression, it must correspond to an assignment $\chi^\star$ which satisfies the desired number of clauses. To finish, we show that, for any instance generated by Reduction 2, a sat-compliant direct tokeniser always corresponds to a bottom-up tokeniser with the same compression quality. $\square$

### 4.3.2 BINARY BOTTOM-UP TOKENISATION IS HARD TO APPROXIMATE

As for direct tokenisation, we now show the APX-hardness of binary bottom-up tokenisation, and compute an explicit hardness-of-approximation constant for it.

**Theorem 4.** *The binary bottom-up tokenisation optimisation problem is* APX-*complete and cannot be approximated in polynomial time with an approximation ratio $\rho < 1.0000001$, unless* P = NP.

*Proof sketch.* This proof is done in two steps, and relies on a similar strategy to Theorem 2, but while replacing the vocabularies of direct tokenisers with merge sequences. Additionally, Kozma and Voderholzer (2024) show that bottom-up tokenisation is contained in APX even for unbounded alphabets, which thus transfers to the problems we study here. Combined, these results prove APX-completeness.

**The optimisation problem is** APX-**hard.** See full proof in Lemma 7 in App. H. As in the proof of APX-hardness of binary direct tokenisation, we perform an L-reduction from 3-OCC-MAX2SAT, reusing the instance map of Reduction 2. We define a reconstruction map which first converts any feasible merge sequence into a sat-compliant one, and then decodes a Boolean assignment from this sat-compliant tokeniser with bounded quality. This completes the proof.

**The gap problem is** NP-**hard.** See full proof in Lemma 8 in App. I. Similarly to the direct case, we show that no polynomial-time algorithm can solve the binary bottom-up tokenisation gap problem with a gap of 1.0000001, unless P = NP. $\square$

## 5 UNARY TOKENISATION IS HARD TO DECIDE

We now move on to the unary tokenisation case. Here, we work with alphabets composed of a single symbol, i.e., $\Sigma = \{a\}$. As $\Sigma^* = \{a^\ell \mid \ell \in \mathbb{N}\}$, it follows that unary character-strings $\mathbf{c} \in \Sigma^*$ may only differ from one another in their length. There exists thus an isomorphism (given by the function $|\cdot|$ and its inverse) between these character-strings and their **string-lengths**, $\ell \in \mathbb{N}$. A natural notation for such problems is then to work directly with string-lengths. In this section, we will thus represent a character-string $\mathbf{c} \in \Sigma^*$ by its length $\ell \in \mathbb{N}$; a dataset $\mathcal{D} = \{\mathbf{c}_m\}_{m=1}^M$ by the lengths of its strings $\mathcal{D}_\mathbb{N} = \{\ell_m\}_{m=1}^M$, where $\mathbf{c}_m = a^{\ell_m}$; and a vocabulary $\mathcal{S} \subset \Sigma^+$ by a set of string-lengths $\mathcal{S}_\mathbb{N} \subset \mathbb{N}_+$. A subword-string is then a sequence of such string-lengths, $\mathbf{s}_\mathbb{N} \in \mathcal{S}_\mathbb{N}^*$ and we have:

$$\mathtt{detok}(\mathbf{s}_\mathbb{N}) \overset{\mathrm{def}}{=} \mathtt{sum}(\mathbf{s}_\mathbb{N}) \qquad \mathtt{tok}_\uparrow[\mathcal{S}_\mathbb{N}](\ell) \overset{\mathrm{def}}{=} \underset{\mathbf{s}_\mathbb{N} \in \mathcal{S}_\mathbb{N}^*}{\arg\min} |\mathbf{s}_\mathbb{N}|, \text{s.t.,} \ell = \mathtt{sum}(\mathbf{s}_\mathbb{N}) \qquad (10)$$

where we overload the functions $\mathtt{detok}$ and $\mathtt{tok}_\uparrow$ to handle this unary-strings representation. Note that all these definitions are equivalent (up to an isomorphism) to the definitions in §3.

When posing either the optimisation or decision version of the unary tokenisation problems, we could thus work with either representation of our data (as strings or string-lengths) and the solutions must be the same. However, the complexity of an algorithm is typically measured as a function of the

length of its input. If this input is a unary string, the input will be as long as this string's length. If this input is a number, however, this input's length behaves logarithmically on the value of the number itself (as this number would typically be encoded in a compact binary representation). When dealing with problems such as unary tokenisation, this introduces an important subtlety: the problem's complexity status may change depending on how we represent it (with strings or string-lengths). If such a problem is NP-hard when either representation is given, it is called **strongly NP-hard**. If this problem is NP-hard only in its string-length representation, but not when represented using unary strings, it is **weakly NP-hard**. Note that the opposite case—where a problem is NP-hard only when representing the data as strings, but not strings-lengths—is not possible, as strings have a larger size than their lengths. Importantly, Fact 1 applies only to strongly NP-hard unary problems; as the trivial identity we use in its proof would not be valid for unary problems with string-length representations. For unary tokenisation, the string representation (where strings are explicitly represented) is more natural, and we are thus interested in strong NP-hardness.

### 5.1 Unary Direct Tokenisation is Strongly NP-Complete

In this section, we prove that the unary direct tokenisation problem is strongly NP-complete. In App. J, we prove that the problem is in NP, even if the input is in string-length representation. To prove NP-hardness of unary direct tokenisation, we then design a polynomial-time reduction from the well-known NP-hard vertex cover problem (`vertex-cover`; Karp, 1972; Garey and Johnson, 1979). Let $(\mathcal{V}, \mathcal{E})$ represent a finite, simple, undirected graph with $\mathcal{V} = \{v_1, \ldots, v_J\}$ and $\mathcal{E} \subseteq \{(v, v') \mid v, v' \in \mathcal{V}, v \neq v'\}$. A set $\mathcal{C} \subseteq \mathcal{V}$ is a **vertex cover** if, for every edge $(v, v') \in \mathcal{E}$, we have that either $v$ or $v'$ is in $\mathcal{C}$. Given a budget $\psi \in \mathbb{N}$, the vertex cover problem requires deciding whether a graph has a vertex cover of at most $\psi$ vertices.

**Definition 3.** *Given a graph $(\mathcal{V}, \mathcal{E})$ and a budget $\psi \in \mathbb{N}$, the **vertex cover decision problem** asks whether there exists a vertex cover $\mathcal{C} \subseteq \mathcal{V}$ with $|\mathcal{C}| \leq \psi$ in this graph.*

For convenience, we will write $\mathrm{VC}(\mathcal{V}, \mathcal{E}, \psi)$ for a function that returns T if its input is a satisfiable instance of the `vertex-cover` decision problem, and F otherwise. We now provide a polynomial-time reduction from `vertex-cover` to `D-1-TOK`, which will prove `D-1-TOK`'s NP-hardness.

**Reduction 3.** *Consider an instance $(\mathcal{V}, \mathcal{E}, \psi)$ of `vertex-cover` and let $N \overset{\text{def}}{=} (J + I + 1)^4$, where $J = |\mathcal{V}|$ and $I = |\mathcal{E}|$. Now, let $\mathtt{enc}(v_j) = j + j^2 N + j^3 N^2$ and $B = N^4$. We construct three subdatasets from this graph as:*

$$\mathcal{D}_1 = \{a^{\ell_j} \mid v_j \in \mathcal{V}\} \cup \{B\}, \quad \text{where } \ell_j = \mathtt{enc}(v_j) \qquad\qquad \textit{vertex-strings} \quad (11\text{a})$$

$$\mathcal{D}_2 = \{a^{\ell'_j} \mid v_j \in \mathcal{V}\}, \qquad \text{where } \ell'_j = \mathtt{enc}(v_j) + B \qquad\qquad \textit{cover-strings} \quad (11\text{b})$$

$$\mathcal{D}_3 = \{a^{\ell''_{j,j'}} \mid (v_j, v_{j'} \in \mathcal{E})\}, \quad \text{where } \ell''_{j,j'} = \mathtt{enc}(v_j) + \mathtt{enc}(v_{j'}) + B \quad \textit{edge-strings} \quad (11\text{c})$$

*Finally, we merge these subdatasets to form a dataset $\mathcal{D} = \mathcal{D}_1 \cup \mathcal{D}_2 \cup \mathcal{D}_3$, and set $K = J + 1 + \psi$ and $\delta = 3J + 2I + 1 - \psi$.*

As before, we complete our NP-hardness proof by showing this to be a valid reduction, i.e., that $\mathrm{VC}(\mathcal{V}, \mathcal{E}, \psi) \iff \mathrm{Tok}^1_{\leftrightarrow}(\mathrm{R3}(\mathcal{V}, \mathcal{E}, \psi))$. Notably, our reduction outputs (in polynomial time, as all lengths are polynomially bounded in the size of the original instance) an instance of the unary direct tokenisation problem in string form. As such, by proving the correctness of this reduction, we prove the *strong* NP-hardness of `D-1-TOK`.

**Theorem 5.** *The unary direct tokenisation decision problem is strongly NP-complete.*

*Proof sketch.* This proof is done in two steps.

**Forward step ($\mathrm{VC}(\mathcal{V}, \mathcal{E}, \psi) \implies \mathrm{Tok}^1_{\leftrightarrow}(\mathrm{R3}(\mathcal{V}, \mathcal{E}, \psi))$).** See full proof in Lemma 10 in App. K. Suppose that the given instance of `vertex-cover` is true, i.e., that $\mathrm{VC}(\mathcal{V}, \mathcal{E}, \psi) = \mathtt{T}$. Now, let $\mathcal{C}^\star \subseteq \mathcal{V}$ be a vertex cover which satisfies this instance. Then we can build a tokeniser with vocabulary $\mathcal{S}_{\mathbb{N}} = \{a^{\ell_j} \mid v_j \in \mathcal{V}\} \cup \{a^B\} \cup \{a^{\ell'_j} \mid v_j \in \mathcal{C}^\star\}$. This tokeniser will encode: all strings in $\mathcal{D}_1$ as a single symbol; $\psi$ strings in $\mathcal{D}_2$ with a single symbol and others with 2; and all strings in $\mathcal{D}_3$ with two symbols (as, per our assumption, all edges have at least one vertex in $\mathcal{C}^\star$). This means that the total amount of tokens used is $(J + 1) + (2J - \psi) + 2I = \delta$. Therefore, $\mathrm{Tok}^1_{\leftrightarrow}(\mathcal{D}, K, \delta) = \mathtt{T}$.

**Backward step** ($\text{Tok}_\diamond^1(\text{R3}(\mathcal{V},\mathcal{E},\psi)) \implies \text{VC}(\mathcal{V},\mathcal{E},\psi)$)**.** See full proof in Lemma 11 in App. L. We prove this lemma in 4 steps. First, we show that all string-lengths in $\mathcal{D}_\mathbb{N}$ are unique. Second, we show that an optimal tokeniser's vocabulary must contain only full strings in $\mathcal{D}_\mathbb{N}$. Third, we show that an optimal tokeniser's vocabulary must include all strings in $\mathcal{D}_1$. Fourth, we show that if a compression of $\delta$ is achieved, than the corresponding vertex-cover instance must be true. Notably, three of these steps rely on the fact that we can use $N$ as a numerical base to prove the uniqueness of both: (i) individual string-lengths, as well as (ii) their pairwise summed values. $\qquad\square$

Interestingly, the unary direct tokenisation problem is tightly related to the problem of choosing denominations for a coin system. In fact, the application of the function $\text{tok}_\diamond[\mathcal{S}_\mathbb{N}](\ell)$ is equivalent to the change-making problem; a problem shown to be (weakly) NP-hard by Lueker (1975) and a classic example for dynamic programming. The unary direct tokenisation problem can thus be equivalently seen as a **general optimal denomination problem**, where—given a set of common currency transactions—one must select optimal coin denominations for a currency; see Shallit (2003) for a discussion of this problem.

**Corollary 1.** *The general optimal denomination decision problem is strongly* NP-*complete.*

## 5.2 A Variant of Unary Bottom-up Tokenisation is (at Least) Weakly NP-Hard

While direct tokenisation over a unary alphabet is strongly NP-complete, our current picture of the complexity of its bottom-up counterpart is more nuanced. In bottom-up tokenisation, one must find a merge sequence $\mathbf{m}$ which is then applied (by $\text{tok}_\uparrow[\mathbf{m}](\mathbf{c})$) *exhaustively* and *in sequence*, replacing all occurrences of each pair one at a time. A variant of this problem—termed **optimal pair encoding (OPE) tokenisation**—relaxes this requirement, using the merge sequence for a different purpose: to define a **merge-extracted vocabulary** $\mathcal{S}_\mathbf{m} = \Sigma \cup \{s_1 \circ s_2 \mid m \in \mathbf{m}, m = (s_1, s_2)\}$ The final tokenisation is then produced by optimally applying this vocabulary, which can be done using the direct encoding function ($\text{tok}_\diamond[\mathcal{S}_\mathbf{m}](\mathbf{c})$). This approach thus ensures that a merge is used only if it contributes to the most efficient segmentation overall. Notably, this variation was used by Schmidt et al. (2024) and formally analysed by Kozma and Voderholzer (2024). Now, let the **unary OPE tokenisation problem** be defined similarly to the other $n$-ary tokenisation problems (in Definition 1), but while constraining the search space to the set of OPE tokenisers: $\mathcal{T}_K \overset{\text{def}}{=} \{\text{tok}_\diamond[\mathcal{S}_\mathbf{m}] \mid \mathbf{m} \in \mathcal{M}^*, |\mathbf{m}| = K\}$. Having defined the decision problem, we now establish its computational hardness.

**Theorem 6.** *The unary optimal pair encoding decision problem is (at least) weakly* NP-*complete.*

*Proof sketch.* The full proof can be found in App. N. Inclusion in NP follows from Kozma and Voderholzer (2024). The proof of NP-hardness is achieved via a polynomial-time reduction from the addition chain sequence decision problem (see App. M for a formal definition), which is known to be NP-complete when its input numbers are encoded in binary (Downey et al., 1981). The reduction reveals a natural connection between the two problems: finding the shortest addition chain for a set of numbers is equivalent to a special case of unary optimal pair encoding where every string in the dataset must be compressed into a single token. $\qquad\square$

## 6 Conclusion and Limitations

We proved several hardness results on bottom-up and direct tokenisation with bounded alphabets, thus answering open questions posed by both Kozma and Voderholzer (2024) and Whittington et al. (2025). A number of open questions remain, however, in particular with respect to approximability. For instance, while we showed that the binary tokenisation optimisation problems are APX-hard and thus cannot be approximated arbitrarily well (unless P = NP), it is unclear whether binary *direct* tokenisation admits any polynomial-time constant-factor approximation algorithm (i.e., whether it lies in APX). With respect to decision problems, while we showed strong NP-hardness of unary direct tokenisation, we were so far only able to prove: (i) weak NP-hardness of unary OPE tokenisation, and (ii) no hardness result for unary (standard) bottom-up tokenisation. Finally, the results of our work are limited in that we consider (i) compression as objective, and (ii) bottom-up and direct tokenisation only; the hardness of other objectives and variants remains open. Overall, however, our results show that tokenisation remains a hard problem, even when restricted to small (even binary or unary) alphabets. Future work should thus explore provably good approximation algorithms.

## ACKNOWLEDGMENTS

We would like to thank Giulia Lanzillotta, Weronika Ormaniec, Dimitri von Rütte, and Felix Sarnthein for their helpful feedback on the introduction. We are also grateful to Pietro Lesci, Amit Moryossef, and Marius Mosbach for their comments on the manuscript, and to Thomas Hofmann for insightful discussions about the paper. We further thank Gregor Bachmann, Hans-Joachim Böckenhauer, Emanuel Skodinis, and Moritz Stocker for their contributions in the early stages of this work, and Stefan Gerdjikov for his valuable early input and discussions. The work of Tiago Pimentel was funded by the Data Analytics Lab at ETH Zürich. The work of Violeta Kastreva was partially supported by the European Union-NextGenerationEU project, through the National Recovery and Resilience Plan of the Republic of Bulgaria [Grant Project No. BG-RRP-2.004-0008].

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

## A    PROVING APX-HARDNESS AND L-REDUCTIONS

APX is the class of optimisation problems that admit polynomial-time constant-factor approximation algorithms. A problem is APX-hard if, under approximation-preserving reductions, it is at least as hard to approximate as every problem in APX. To prove APX-hardness, standard polynomial-time reductions are insufficient because they do not guarantee the preservation of approximation ratios. Instead, we rely on L-reductions. An L-reduction is a polynomial-time reduction that relates optimal objective values and approximation errors by fixed constants, ensuring that any constant-factor approximation for the target problem implies a constant-factor approximation for the source problem (Papadimitriou and Yannakakis, 1991). L-reductions achieve this by relying on two polynomial-timed functions. Given optimisation problems $A$ and $B$, a **reduction function** receives an instance of reduced problem $A$ and outputs an instance of the target problem $B$; in turn, a **reconstruction function** receives both an instance of problem $A$, and a solution to problem $B$, returning a solution to problem $A$. Formally, we define L-reductions as follows:

**Definition 4.** *Let $A$ and $B$ be optimization problems with objective-value functions $\mathfrak{G}_A(\cdot)$ and $\mathfrak{G}_B(\cdot)$, and optimal values $\mathrm{OPT}_A(\mathcal{I}_A)$ and $\mathrm{OPT}_B(\mathcal{I}_B)$ for instances $\mathcal{I}_A$ and $\mathcal{I}_B$. We say that $A$ L-reduces to $B$ (denoted $A \leq_L B$) if there exist polynomial-time computable reduction and reconstruction functions $f$ and $g$ and constants $\alpha, \beta > 0$ such that for every instance $\mathcal{I}_A$ of $A$:*

*(i) $\mathcal{I}_B = f(\mathcal{I}_A)$ is an instance of $B$ whose optimal solution is upper-bounded by $\alpha$ times the optimum of $\mathcal{I}_A$. Formally:*

$$|\mathrm{OPT}_B(f(\mathcal{I}_A))| \leq \alpha \cdot |\mathrm{OPT}_A(\mathcal{I}_A)| . \tag{12}$$

*(ii) Given $\mathcal{I}_B = f(\mathcal{I}_A)$ and any of its feasible solutions $s_B$ of $\mathcal{I}_B$, $s_A = g(\mathcal{I}_A, s_B)$ is a solution of $\mathcal{I}_A$ and has a quality difference to the optimum bounded by $s_B$'s quality:*

$$\left| \mathfrak{G}_A(g(\mathcal{I}_A, s_B)) - \mathrm{OPT}_A(\mathcal{I}_A) \right| \leq \beta \cdot \left| \mathfrak{G}_B(s_B) - \mathrm{OPT}_B(f(\mathcal{I}_A)) \right| . \tag{13}$$

L-reductions preserve APX-hardness; therefore, if $A$ is APX-hard and $A \leq_L B$, then $B$ is APX-hard (Papadimitriou and Yannakakis, 1991; Ausiello et al., 2012).

## B    PROOF OF FORWARD STEP OF THEOREM 1

**Lemma 1.** *If a* `3-OCC-MAX2SAT` *instance is satisfiable, then the* `D-2-TOK` *instance output by Reduction 1 is also satisfiable. Formally:* $3\mathrm{OM2S}(\mathcal{X}, \mathcal{C}, \gamma) \implies \mathrm{Tok}^2_{\circlearrowright}(\mathrm{R1}(\mathcal{X}, \mathcal{C}, \gamma))$

*Proof.* To prove this forward step of Theorem 1, we first establish that a satisfiable `3-OCC-MAX2SAT` instance guarantees the existence of a binary direct tokeniser that meets the compression target. Assume a $(\mathcal{X}, \mathcal{C}, \gamma)$ instance of the `3-OCC-MAX2SAT` problem is satisfiable, i.e., that $3\mathrm{OM2S}(\mathcal{X}, \mathcal{C}, \gamma)$ is true. We must prove that, in this case, $\mathrm{Tok}^2_{\circlearrowright}(\mathrm{R1}(\mathcal{X}, \mathcal{C}, \gamma))$ is also true. Now, let $\chi^\star = \{x_j^\star\}_{j=1}^J$ be any satisfying solution to the $(\mathcal{X}, \mathcal{C}, \gamma)$ instance. We will denote the number of clauses satisfied by $\chi^\star$ by $\gamma^\star$, noting that $\gamma^\star \geq \gamma$ by assumption. We can construct a tokeniser from this solution as follows:

$$\mathcal{S} = \Sigma \bigcup \left\{ 1\mathbf{x}_j^\mathtt{T}, \mathbf{x}_j^\mathtt{T}1, 1\mathbf{x}_j^\mathtt{F}, \mathbf{x}_j^\mathtt{F}1 \right\}_{j=1}^J \quad \bigcup \quad \left\{ 1\mathbf{x}_j^\mathtt{T}1 \text{ if } x_j^\star = \mathtt{T} \text{ else } 1\mathbf{x}_j^\mathtt{F}1 \right\}_{j=1}^J \tag{14}$$

Note that—as required by our reduction—this tokeniser has vocabulary size $|\mathcal{S}| = |\Sigma| + K$, since $K = 5J$ tokens were added. Under this tokeniser, we have:

$$\mathtt{tok}_{\circlearrowright}[\mathcal{S}](\mathcal{D}_1) = \left\{ \langle 1\mathbf{x}_j^\mathtt{T} \rangle, \langle \mathbf{x}_j^\mathtt{T}1 \rangle, \langle 1\mathbf{x}_j^\mathtt{F} \rangle, \langle \mathbf{x}_j^\mathtt{F}1 \rangle \quad \text{(length 1)} \quad \mid 1 \leq j \leq J \right\} \quad \times f \tag{15a}$$

$$\mathtt{tok}_{\circlearrowright}[\mathcal{S}](\mathcal{D}_2) = \left\{ \begin{array}{ll} \langle 1\mathbf{x}_j^\mathtt{T}1 \rangle, \langle 1\mathbf{x}_j^\mathtt{F}, 1 \rangle & \text{(length 3)} \quad \text{if } x_j^\star = \mathtt{T} \\ \langle 1\mathbf{x}_j^\mathtt{T}, 1 \rangle, \langle 1\mathbf{x}_j^\mathtt{F}1 \rangle & \text{(length 3)} \quad \text{else} \end{array} \mid 1 \leq j \leq J \right\} \quad \times f' \tag{15b}$$

$$\mathtt{tok}_{\circlearrowright}[\mathcal{S}](\mathcal{D}_3) = \left\{ \begin{array}{ll} \langle 1\mathbf{x}_j^\mathtt{T}1, \mathbf{x}_j^\mathtt{F}1 \rangle & \text{(length 2)} \quad \text{if } x_j^\star = \mathtt{T} \\ \langle 1\mathbf{x}_j^\mathtt{T}, 1\mathbf{x}_j^\mathtt{F}1 \rangle & \text{(length 2)} \quad \text{else} \end{array} \mid 1 \leq j \leq J \right\} \quad \times f'' \tag{15c}$$

$$\mathtt{tok}_{\circlearrowright}[\mathcal{S}](\mathcal{D}_4) = \left\{ \begin{array}{ll} \langle 1L_i^11, L_i^21 \rangle & \text{(length 2)} \quad \text{if } L_i^1 = \mathtt{T} \\ \langle 1L_i^1, 1L_i^21 \rangle & \text{(length 2)} \quad \text{elif } L_i^2 = \mathtt{T} \\ \langle 1L_i^1, 1, L_i^21 \rangle & \text{(length 3)} \quad \text{else} \end{array} \mid 1 \leq i \leq I \right\} \quad \times 1 \tag{15d}$$

where we override function $\mathtt{tok}_\looparrowright[\mathcal{S}]$ to apply elementwise to a full dataset of character-strings, instead of to a unique $\mathbf{c}$. Consequently, we get the compressed lengths:

$$\mathfrak{G}_\ell(\mathtt{tok}_\looparrowright[\mathcal{S}], \mathcal{D}_1) = 4\,J\,f = 252\,J\,f, \qquad \mathfrak{G}_\ell(\mathtt{tok}_\looparrowright[\mathcal{S}], \mathcal{D}_2) = 3\,J\,f' = 63\,J, \qquad (16a)$$

$$\mathfrak{G}_\ell(\mathtt{tok}_\looparrowright[\mathcal{S}], \mathcal{D}_3) = 2\,J\,f'' = 14\,J, \qquad \mathfrak{G}_\ell(\mathtt{tok}_\looparrowright[\mathcal{S}], \mathcal{D}_4) = 3\,I - \gamma^\star \qquad (16b)$$

We have that each character-string in dataset $\mathcal{D}_4$ is compressed to 2 symbols if either $L_i^1$ or $L_i^2$ are true, and otherwise is kept at 3 symbols; the $\gamma^\star$ satisfied clauses in $\chi^\star$ will thus be compressed to 2 symbols and the unsatisfied clauses to 3. Summing these values together, we get the compressed length of the entire dataset under this tokeniser: $\mathfrak{G}_\ell(\mathtt{tok}_\looparrowright[\mathcal{S}], \mathcal{D}) = 329J + 3\,I - \gamma^\star$. Finally:

$$\gamma^\star \geq \gamma \implies 329J + 3\,I - \gamma^\star \leq 329J + 3\,I - \gamma \qquad (17)$$

This completes this proof. $\qquad\square$

## C   PROOF OF BACKWARD STEP OF THEOREM 1

Before starting our lemma's proof, we define a few notions which will be useful throughout it. First, we define a **sat-compliant** tokeniser to be any tokeniser which: (i) contains all tokens of the form $1\mathbf{x}_j^\mathtt{T}, \mathbf{x}_j^\mathtt{T}1, 1\mathbf{x}_j^\mathtt{F}, \mathbf{x}_j^\mathtt{F}1$; and (ii) contains either $1\mathbf{x}_j^\mathtt{T}1$ or $1\mathbf{x}_j^\mathtt{F}1$ for each $j \in \{1, \ldots, J\}$. Otherwise, we call the tokeniser **sat-noncompliant**. Given the vocabulary of a sat-compliant tokeniser, we can easily build an assignment to a 3-OCC-MAX2SAT instance with the following function:

$$g(\mathcal{S}) = \{x_j\}_{j=1}^J, \text{ where } \begin{cases} x_j = \mathtt{T} & \text{if } 1\mathbf{x}_j^\mathtt{T}1 \in \mathcal{S} \\ x_j = \mathtt{F} & \text{elif } 1\mathbf{x}_j^\mathtt{F}1 \in \mathcal{S} \end{cases} \qquad (18)$$

Further, we will define as a **101-string** any character-string of the form $10^+1$, and as a **10101-string** any character-string of the form $10^+10^+1$. (The $0^+$ notation stands for a sequence of one or more $0$ characters.) Considering the datasets output by Reduction 1, we know that there are no 101-strings in $\mathcal{D}_1$. Further, we know that each unique 101-string appears in datasets $\mathcal{D}_2$ and $\mathcal{D}_3$ exactly $f'$ and $f''$ times, respectively, and exactly 3 times in $\mathcal{D}_4$. (This is due to us working with the three-occurrences variant of MAX2SAT and to the fact that $\mathbf{x}_j^\mathtt{T} = 0^{2j-1}$ and $\mathbf{x}_j^\mathtt{F} = 0^{2j}$.) We now prove the following lemma.

**Lemma 2.** *If the D-2-TOK instance output by Reduction 1 is satisfiable, then the 3-OCC-MAX2SAT instance which generated it is as well. Formally:* $\mathrm{Tok}_\looparrowright^2(\mathrm{R1}(\mathcal{X}, \mathcal{C}, \gamma)) \implies 3\mathrm{OM2S}(\mathcal{X}, \mathcal{C}, \gamma)$.

*Proof.* Assume this $(\mathcal{D}, K, \delta)$ instance of D-2-TOK—where $(\mathcal{D}, K, \delta) = \mathrm{R1}(\mathcal{X}, \mathcal{C}, \gamma)$—is satisfiable, i.e., that $\mathrm{Tok}_\looparrowright^2(\mathrm{R1}(\mathcal{X}, \mathcal{C}, \gamma))$ evaluates to true. We must prove that, in this case, $3\mathrm{OM2S}(\mathcal{X}, \mathcal{C}, \gamma)$ also evaluates to true. Now, let $\mathcal{S}_{\mathrm{opt}}$ be an arbitrary optimal solution to $(\mathcal{D}, K, \delta)$. We know, by definition, that:

$$\mathrm{Tok}_\looparrowright^2(\mathrm{R1}(\mathcal{X}, \mathcal{C}, \gamma)) \iff \left( \mathfrak{G}_\ell(\mathtt{tok}_\looparrowright[\mathcal{S}_{\mathrm{opt}}], \mathcal{D}) \leq \delta \right) \qquad (19)$$

We can thus prove this lemma by showing the following implication:

$$\left( \mathfrak{G}_\ell(\mathtt{tok}_\looparrowright[\mathcal{S}_{\mathrm{opt}}], \mathcal{D}) \leq \delta \right) \implies 3\mathrm{OM2S}(\mathcal{X}, \mathcal{C}, \gamma) \qquad (20)$$

We will proceed in four steps:

①  we prove that $\mathcal{S}_{\mathrm{opt}}$ must include all tokens of the form $1\mathbf{x}_j^\mathtt{T}, \mathbf{x}_j^\mathtt{T}1, 1\mathbf{x}_j^\mathtt{F}, \mathbf{x}_j^\mathtt{F}1$;

②  we prove that $\mathcal{S}_{\mathrm{opt}}$ must, in addition to the tokens above, only include tokens of the form $1\mathbf{x}_j^\mathtt{T}1, 1\mathbf{x}_j^\mathtt{F}1$;

③  we prove that $\mathcal{S}_{\mathrm{opt}}$ may only include, for each $j$, either token $1\mathbf{x}_j^\mathtt{T}1$ or $1\mathbf{x}_j^\mathtt{F}1$;

④  finally, we prove that, if $\left( \mathfrak{G}_\ell(\mathtt{tok}_\looparrowright[\mathcal{S}_{\mathrm{opt}}], \mathcal{D}) \leq \delta \right)$, we can build a variable assignment which satisfies this 3-OCC-MAX2SAT instance $(\mathcal{X}, \mathcal{C}, \gamma)$.

Note that, together, steps ① to ③ show that $\mathcal{S}_{\text{opt}}$ must be the vocabulary of a sat-compliant tokeniser; in step ④, we will then rely on the function $g$ (defined above in Eq. (18)) to convert this vocabulary into a satisfying assignment $\chi = g(\mathcal{S}_{\text{opt}})$ of the 3-OCC-MAX2SAT instance. □

**LemmaProofStep 1.** (Step ①). *An optimal tokeniser must include all tokens of the form* $1\mathbf{x}_j^{\text{T}}, \mathbf{x}_j^{\text{T}}1, 1\mathbf{x}_j^{\text{F}}, \mathbf{x}_j^{\text{F}}1$, *i.e.:*

$$\left\{1\mathbf{x}_j^{\text{T}}, \mathbf{x}_j^{\text{T}}1, 1\mathbf{x}_j^{\text{F}}, \mathbf{x}_j^{\text{F}}1\right\}_{j=1}^{J} \subseteq \mathcal{S}_{\text{opt}} \tag{21}$$

*Proof.* We prove this step by contradiction. Assume there exists an optimal tokeniser with vocabulary $\mathcal{S}_{\mathsf{x}}$ which does not include $t > 0$ of the tokens above. Now, choose an arbitrary set of $t$ tokens in this vocabulary which are not of the above form, and replace them with the missing tokens in this set. We denote this new tokeniser's vocabulary by $\mathcal{S}_{\checkmark}$. Note that the strings in $\mathcal{D}_1$ with these missing tokens were represented with at least 2 symbols under $\mathcal{S}_{\mathsf{x}}$, but with a single token under $\mathcal{S}_{\checkmark}$, i.e.:

$$\mathfrak{G}_\ell(\texttt{tok}_\looparrowright[\mathcal{S}_{\mathsf{x}}], \mathcal{D}_1) \geq (4J + t)f, \qquad \mathfrak{G}_\ell(\texttt{tok}_\looparrowright[\mathcal{S}_{\checkmark}], \mathcal{D}_1) = 4Jf \tag{22}$$

Further, note that under $\mathcal{S}_{\checkmark}$, we have that strings in dataset $\mathcal{D}_2$ are compressed to at most two symbols, while strings in $\mathcal{D}_3$ and $\mathcal{D}_4$ are compressed to at most three symbols:

$$\forall \mathbf{c} \in \mathcal{D}_2 : \mathfrak{G}_\ell(\texttt{tok}_\looparrowright[\mathcal{S}_{\checkmark}], \mathbf{c}) \leq 2, \quad \forall \mathbf{c} \in \mathcal{D}_3 \cup \mathcal{D}_4 : \mathfrak{G}_\ell(\texttt{tok}_\looparrowright[\mathcal{S}_{\checkmark}], \mathbf{c}) \leq 3 \tag{23}$$

To improve on this compressed length, $\mathcal{S}_{\mathsf{x}}$ must, thus, compress strings in $\mathcal{D}_2$ to a single symbol, or strings in $\mathcal{D}_3$ and $\mathcal{D}_4$ to one or two symbols. Notably, this can only be done if the non-compliant tokens in $\mathcal{S}_{\mathsf{x}}$ contain 101-strings. This is because, to compress a string in $\mathcal{D}_2$ to a single symbol, the full character-string must become a token, and $\mathcal{D}_2$ only includes 101-strings. Moreover, under $\mathcal{S}_{\checkmark}$, strings in $\mathcal{D}_3$ and $\mathcal{D}_4$ are already compressed to at most $\langle 1\mathbf{x}_j^{\text{T}}, 1, \mathbf{x}_j^{\text{T}}1 \rangle$. To further compress them, tokeniser $\mathcal{S}_{\mathsf{x}}$ must include tokens which cross the "middle" of this character-string, which would make this tokens at least have a 101 prefix or suffix. We consider the best case scenario, which is if they are exactly 101-strings, as any longer string will be at most as frequent as it.

As discussed above, however, each 101-string appears at most: $f'$ times in $\mathcal{D}_2$, $f''$ times in $\mathcal{D}_3$, and 3 times in $\mathcal{D}_4$. This gives us a best case scenario—in which all the strings in which a new token appears are compressed to a single symbol—where:

$$\mathfrak{G}_\ell(\texttt{tok}_\looparrowright[\mathcal{S}_{\checkmark}], \mathcal{D}_2 \cup \mathcal{D}_3 \cup \mathcal{D}_4) - \mathfrak{G}_\ell(\texttt{tok}_\looparrowright[\mathcal{S}_{\mathsf{x}}], \mathcal{D}_2 \cup \mathcal{D}_3 \cup \mathcal{D}_4) \leq t(f' + 2(f'' + 3)) \tag{24}$$

As the difference in Eq. (22) is of at least $tf$ tokens, we put these together:

$$\mathfrak{G}_\ell(\texttt{tok}_\looparrowright[\mathcal{S}_{\checkmark}], \mathcal{D}) - \mathfrak{G}_\ell(\texttt{tok}_\looparrowright[\mathcal{S}_{\mathsf{x}}], \mathcal{D}) \leq t(f' + 2(f'' + 3)) - tf \tag{25}$$

As $f > f' + 2(f'' + 3)$, this difference is smaller than zero, implying that $\mathcal{S}_{\checkmark}$ improves on $\mathcal{S}_{\mathsf{x}}$. This shows a contradiction, which completes our proof. □

**LemmaProofStep 2.** (Step ②). *An optimal tokeniser must include all tokens of the form* $1\mathbf{x}_j^{\text{T}}, \mathbf{x}_j^{\text{T}}1, 1\mathbf{x}_j^{\text{F}}, \mathbf{x}_j^{\text{F}}1$, *and further only tokens of the form* $1\mathbf{x}_j^{\text{T}}1, 1\mathbf{x}_j^{\text{F}}1$, *i.e.:*

$$\left\{1\mathbf{x}_j^{\text{T}}, \mathbf{x}_j^{\text{T}}1, 1\mathbf{x}_j^{\text{F}}, \mathbf{x}_j^{\text{F}}1\right\}_{j=1}^{J} \subseteq \mathcal{S}_{\text{opt}} \quad \text{and} \quad \mathcal{S}_{\text{opt}} \subset \left\{1\mathbf{x}_j^{\text{T}}, \mathbf{x}_j^{\text{T}}1, 1\mathbf{x}_j^{\text{F}}, \mathbf{x}_j^{\text{F}}1, 1\mathbf{x}_j^{\text{T}}1, 1\mathbf{x}_j^{\text{F}}1\right\}_{j=1}^{J} \tag{26}$$

*Proof.* As before, we prove this step by contradiction. Given step ①, we know an optimal tokeniser includes all tokens $1\mathbf{x}_j^{\text{T}}, \mathbf{x}_j^{\text{T}}1, 1\mathbf{x}_j^{\text{F}}, \mathbf{x}_j^{\text{F}}1$. Now, assume there exists an optimal tokeniser with vocabulary $\mathcal{S}_{\mathsf{x}}$ with $t > 0$ tokens which are not of the form $1\mathbf{x}_j^{\text{T}}, \mathbf{x}_j^{\text{T}}1, 1\mathbf{x}_j^{\text{F}}, \mathbf{x}_j^{\text{F}}1$ or $1\mathbf{x}_j^{\text{T}}1, 1\mathbf{x}_j^{\text{F}}1$; note that these $t$ tokens are sat-noncompliant. Choose an arbitrary set of $t$ unused compliant tokens—i.e., with form $1\mathbf{x}_j^{\text{T}}1, 1\mathbf{x}_j^{\text{F}}1$—to replace the non-compliant tokens with, forming a new tokeniser's vocabulary $\mathcal{S}_{\checkmark}$. Both these vocabularies compress strings in $\mathcal{D}_1$ equally:

$$\mathfrak{G}_\ell(\texttt{tok}_\looparrowright[\mathcal{S}_{\mathsf{x}}], \mathcal{D}_1) = 4Jf, \qquad \mathfrak{G}_\ell(\texttt{tok}_\looparrowright[\mathcal{S}_{\checkmark}], \mathcal{D}_1) = 4Jf \tag{27}$$

For strings in $\mathcal{D}_2$: if the entire string is in the vocabulary, it is encoded as a single token; otherwise, it is represented with two symbols. Under $\mathcal{S}_{\checkmark}$, there are $J$ tokens covering strings in $\mathcal{D}_2$. Under $\mathcal{S}_{\mathsf{x}}$, there are only $(J - t)$ tokens covering strings in $\mathcal{D}_2$. This implies:

$$\mathfrak{G}_\ell(\texttt{tok}_\looparrowright[\mathcal{S}_{\mathsf{x}}], \mathcal{D}_2) = (3J + t)f', \qquad \mathfrak{G}_\ell(\texttt{tok}_\looparrowright[\mathcal{S}_{\checkmark}], \mathcal{D}_2) = 3Jf' \tag{28}$$

Finally, for strings in $\mathcal{D}_3$ and $\mathcal{D}_4$, a similar argument to the previous step applies: (i) only tokens containing 101-strings can compress these datasets; (ii) each 101-string appears at most $f'' + 3$ times in them; (iii) each 101-string will lead to at most two symbols being saved. As $\mathcal{S}_{\mathsf{x}}$ differs from $\mathcal{S}_{\checkmark}$ in $t$ tokens, we get that it will improve on it by at most:

$$\mathfrak{G}_\ell(\mathtt{tok}_\diamond[\mathcal{S}_{\checkmark}], \mathcal{D}_3 \cup \mathcal{D}_4) - \mathfrak{G}_\ell(\mathtt{tok}_\diamond[\mathcal{S}_{\mathsf{x}}], \mathcal{D}_3 \cup \mathcal{D}_4) \leq 2t(f'' + 3) \tag{29}$$

Summing together the compression on all datasets, we get that their difference is bounded by:

$$\mathfrak{G}_\ell(\mathtt{tok}_\diamond[\mathcal{S}_{\checkmark}], \mathcal{D}) - \mathfrak{G}_\ell(\mathtt{tok}_\diamond[\mathcal{S}_{\mathsf{x}}], \mathcal{D}) \leq 2t(f'' + 3) - tf' \tag{30}$$

As $f' > 2(f'' + 3)$, this difference is smaller than zero, implying that $\mathcal{S}_{\checkmark}$ improves on $\mathcal{S}_{\mathsf{x}}$. This shows a contradiction, which completes our proof. $\qquad\square$

**LemmaProofStep 3.** (Step $\textcircled{3}$). *An optimal tokeniser must be* sat*-compliant: it must contain all tokens of the form* $1\mathbf{x}_j^{\mathsf{T}}, \mathbf{x}_j^{\mathsf{T}}1, 1\mathbf{x}_j^{\mathsf{F}}, \mathbf{x}_j^{\mathsf{F}}1$ *and it must contain either* $1\mathbf{x}_j^{\mathsf{T}}1$ *or* $1\mathbf{x}_j^{\mathsf{F}}1$ *for each* $1 \leq j \leq J$.

*Proof.* As before, we prove this step by contradiction. Given step $\textcircled{1}$, we know an optimal tokeniser includes all tokens $1\mathbf{x}_j^{\mathsf{T}}, \mathbf{x}_j^{\mathsf{T}}1, 1\mathbf{x}_j^{\mathsf{F}}, \mathbf{x}_j^{\mathsf{F}}1$. Further, given step $\textcircled{2}$, we know its other tokens all have form $1\mathbf{x}_j^{\mathsf{T}}1, 1\mathbf{x}_j^{\mathsf{F}}1$. Now, assume there exists an optimal tokeniser with vocabulary $\mathcal{S}_{\mathsf{x}}$ which includes both $1\mathbf{x}_j^{\mathsf{T}}1$ and $1\mathbf{x}_j^{\mathsf{F}}1$ for $t > 0$ variables, and thus neither of those two for $t > 0$ other variables. Then, define $\mathcal{S}_{\checkmark}$ as a vocabulary where the $1\mathbf{x}_j^{\mathsf{F}}1$ token of all $t$ doubly assigned variables are replaced with the $1\mathbf{x}_j^{\mathsf{T}}1$ token of all non-assigned variables. Note that $\mathcal{S}_{\checkmark}$ is sat-compliant. These two tokenisers achieve the same compression on $\mathcal{D}_1$ and $\mathcal{D}_2$:

$$\mathfrak{G}_\ell(\mathtt{tok}_\diamond[\mathcal{S}_{\mathsf{x}}], \mathcal{D}_1 \cup \mathcal{D}_2) = 4Jf + 3Jf' \quad \text{and} \quad \mathfrak{G}_\ell(\mathtt{tok}_\diamond[\mathcal{S}_{\checkmark}], \mathcal{D}_1 \cup \mathcal{D}_2) = 4Jf + 3Jf' \tag{31}$$

The tokeniser with vocabulary $\mathcal{S}_{\checkmark}$ will then compress each string in $\mathcal{D}_3$ to 2 symbols, while $\mathcal{S}_{\mathsf{x}}$ will compress the $t$ strings $1\mathbf{x}_j^{\mathsf{T}}1\mathbf{x}_j^{\mathsf{F}}1$ with unassigned variables to 3 symbols. This will lead to a total compression of:

$$\mathfrak{G}_\ell(\mathtt{tok}_\diamond[\mathcal{S}_{\mathsf{x}}], \mathcal{D}_3) = (2J + t)f'', \qquad \mathfrak{G}_\ell(\mathtt{tok}_\diamond[\mathcal{S}_{\checkmark}], \mathcal{D}_3) = 2Jf'' \tag{32}$$

Finally, the $t$ doubly assigned tokens of the form $1\mathbf{x}_j^{\mathsf{F}}1$ (which $\mathcal{S}_{\checkmark}$ does not contain) appear at most three times in $\mathcal{D}_4$ and will lead to at most one symbol being saved, leading to a bound:

$$\mathfrak{G}_\ell(\mathtt{tok}_\diamond[\mathcal{S}_{\checkmark}], \mathcal{D}_4) - \mathfrak{G}_\ell(\mathtt{tok}_\diamond[\mathcal{S}_{\mathsf{x}}], \mathcal{D}_4) \leq 3t \tag{33}$$

Putting these compressed lengths together, we get:

$$\mathfrak{G}_\ell(\mathtt{tok}_\diamond[\mathcal{S}_{\checkmark}], \mathcal{D}) - \mathfrak{G}_\ell(\mathtt{tok}_\diamond[\mathcal{S}_{\mathsf{x}}], \mathcal{D}) \leq 3t - tf'' \tag{34}$$

As $f'' > 3$, this difference is smaller than zero, implying that $\mathcal{S}_{\checkmark}$ improves on $\mathcal{S}_{\mathsf{x}}$. This shows a contradiction, which completes our proof. $\qquad\square$

**LemmaProofStep 4.** (Step $\textcircled{4}$). *If an optimal tokeniser achieves a compressed length of at most* $329J + 3I - \gamma$, *the original* 3-OCC-MAX2SAT *instance is satisfiable, i.e.:*

$$\left(\mathfrak{G}_\ell(\mathtt{tok}_\diamond[\mathcal{S}_{\mathrm{opt}}], \mathcal{D}) \leq 329J + 3I - \gamma\right) \implies \mathrm{3OM2S}(\mathcal{X}, \mathcal{C}, \gamma) \tag{35}$$

*Proof.* Given steps $\textcircled{1}$ to $\textcircled{3}$, we know that an optimal tokeniser will be sat-compliant. We will now denote this optimal tokeniser's vocabulary by $\mathcal{S}_{\mathrm{opt}}$ and use Eq. (18) to extract a 3-OCC-MAX2SAT assignment $\chi^\star = g(\mathcal{S}_{\mathrm{opt}})$ which corresponds to this tokeniser's vocabulary. From the previous proof steps we see that any sat-compliant tokeniser achieves the following compressed length in $\mathcal{D}_1, \mathcal{D}_2$, and $\mathcal{D}_3$:

$$\mathfrak{G}_\ell(\mathtt{tok}_\diamond[\mathcal{S}_{\mathrm{opt}}], \mathcal{D}_1 \cup \mathcal{D}_2 \cup \mathcal{D}_3) = 4Jf + 3Jf' + 2Jf'' = 329J \tag{36}$$

Now, note that a character-string $1L_i^1 1 L_i^2 1$ in $\mathcal{D}_4$ will be: compressed to two symbols if at least one of the tokens $1L_i^1 1$ or $1L_i^2 1$ exists, or compressed to three symbols if neither exists. Equivalently, a clause $L_i^1 \vee L_i^2$ in 3-OCC-MAX2SAT is: satisfied if either $L_i^1$ or $L_i^2$ evaluates to true, or not satisfied if both evaluate to false. Given our construction of function $g$ above, one of 3-OCC-MAX2SAT's clauses will be satisfied if and only if its corresponding string in $\mathcal{D}_4$ is compressed to two symbols. We can thus state that:

$$\left(\mathfrak{G}_\ell(\mathtt{tok}_\diamond[\mathcal{S}_{\mathrm{opt}}], \mathcal{D}_4) = 3I - \gamma^\star\right) \iff \left(\sum_{i=1}^{I} \mathbb{1}_{\chi^\star}\{L_i^1 \vee L_i^2\} = \gamma^\star\right) \tag{37}$$

Given the construction of $\delta$ as $329J + 3I - \gamma$, we conclude that a sat-compliant tokeniser which compresses the full dataset to at least that size can be mapped to a 3-OCC-MAX2SAT assignment which satisfies at least $\gamma$ clauses. This concludes the proof. $\qquad\square$

# D  PROOF THAT BINARY DIRECT TOKENISATION IS APX-HARD (STEP 1 OF THEOREM 2)

We prove APX-hardness of `D-2-TOK` by providing an L-reduction (defined in Definition 4) from the APX-complete problem `3-OCC-MAX2SAT` (Berman and Karpinski, 1999). For the reduction function $f_{\text{D-2-TOK}}$ of the L-reduction, we do not introduce a new transformation: we directly reuse the reduction from Reduction 1. To complete the L-reduction, we additionally provide a reconstruction function $g_{\text{D-2-TOK}}$ that maps any feasible tokeniser for the `D-2-TOK` instance back to a valid Boolean assignment for the original formula. We then verify these functions satisfy the L-reduction conditions for suitable constants $\alpha$ and $\beta$. Notably, for a given instance $(\mathcal{X}, \mathcal{C})$ and assignment $\chi \in \{\text{F}, \text{T}\}^J$ of the `3-OCC-MAX2SAT` optimisation problem, we will denote its cost function as

$$\mathfrak{G}_{M2S}(\chi, \mathcal{C}) = \sum_{i=1}^{I} \mathbb{1}_\chi \{L_i^1 \vee L_i^2\}. \tag{38}$$

Now that the objective (cost) functions for both problems are fixed, we can prove APX-hardness.

**Lemma 3.** *The binary direct tokenisation problem is* APX-*hard.*

*Proof.* We prove that `D-2-TOK` is APX-hard by constructing an L-reduction from the APX-complete problem `3-OCC-MAX2SAT`. Let $(\mathcal{X}, \mathcal{C})$ be an arbitrary instance of `3-OCC-MAX2SAT`. To establish the L-reduction, we must define a pair of polynomial-time algorithms, $f_{\text{D-2-TOK}}$ and $g_{\text{D-2-TOK}}$, that map instances and solutions between the two problems, and find constants $\alpha$ and $\beta$ such that the conditions in App. A are satisfied. We will prove this holds for $\alpha = 444$ and $\beta = 1$.

**L-Reduction's Condition (i) is Satisfied.** First, we want to prove that algorithm $f_{\text{D-2-TOK}}$ produces an instance $(\mathcal{D}, K) = f_{\text{D-2-TOK}}(\mathcal{X}, \mathcal{C})$ of `D-2-TOK` such that

$$\text{Tok}_\diamond^{2,\star}(f_{\text{D-2-TOK}}(\mathcal{X}, \mathcal{C})) \leq \alpha \cdot 3\text{OM2S}^\star(\mathcal{X}, \mathcal{C}). \tag{39}$$

We define $f_{\text{D-2-TOK}}$ using the polynomial-time construction detailed in Reduction 1. Since this mapping is formulated for the decision problem as $\text{R1}(\mathcal{X}, \mathcal{C}, \gamma) = (\mathcal{D}, K, \delta)$, we discard the target values from the function definition for the optimisation problem. Formally, we define $f_{\text{D-2-TOK}}(\mathcal{X}, \mathcal{C}) = (\mathcal{D}, K)$, where the dataset $\mathcal{D}$ and the token budget $K = 5J$ remain identical to the original construction.

As shown by Eqs. (36) and (37), the size of any dataset tokenised by a `sat`-compliant tokeniser $\mathcal{S}_{\checkmark}$ is linearly related to the number of satisfied clauses $\gamma^\star$ by assignment $\chi^\star = g(\mathcal{S}_{\checkmark})$:

$$\mathfrak{G}_\ell(\text{tok}_\diamond[\mathcal{S}_{\checkmark}], \mathcal{D}) = 329J + 3I - \gamma^\star \tag{40}$$

As there is a one-to-one mapping between `sat`-compliant tokenisers, and assignments $\chi^\star$, it trivially follows that the optimal tokenisation size is $\text{Tok}_\diamond^{2,\star}(\mathcal{D}, K) = 329J + 3I - 3\text{OM2S}^\star(\mathcal{X}, \mathcal{C})$.

By definition, each variable in $\mathcal{X}$ appears exactly three times in $\mathcal{C}$, and each clause contains two literals. This implies $3J = 2I$. Furthermore, there always exists an (efficiently computable) assignment satisfying at least half the clauses, meaning that the problem is trivial for $\gamma < \frac{1}{2}I$. We thus restrict our attention to cases where $3\text{OM2S}^\star(\mathcal{X}, \mathcal{C}) = \gamma_{\text{opt}} \geq \frac{1}{2}I$, which constitute the interesting instances of `3-OCC-MAX2SAT`. Substituting $J = \frac{2}{3}I$ into the optimality equation, we get:

$$\text{Tok}_\diamond^{2,\star}(f_{\text{D-2-TOK}}(\mathcal{X}, \mathcal{C})) = \text{Tok}_\diamond^{2,\star}(\mathcal{D}, K) \tag{41a}$$

$$= 329\left(\frac{2}{3}I\right) + 3I - 3\text{OM2S}^\star(\mathcal{X}, \mathcal{C}) \tag{41b}$$

$$= \frac{667}{3}I - 3\text{OM2S}^\star(\mathcal{X}, \mathcal{C}) \tag{41c}$$

$$\leq \frac{1331}{3} 3\text{OM2S}^\star(\mathcal{X}, \mathcal{C}) \tag{41d}$$

Thus, the first condition of the L-reduction holds with $\alpha = 444$.

**L-Reduction's Condition (ii) is Satisfied.** Second, let $(\mathcal{D}, K) = f_{\text{D-2-TOK}}(\mathcal{X}, \mathcal{C})$, we want to prove that algorithm $g_{\text{D-2-TOK}}$ maps any solution $\mathcal{S}$ of $(\mathcal{D}, K)$ to a solution $\chi^\star$ of $(\mathcal{X}, \mathcal{C})$ such that

$$
\begin{aligned}
|\mathfrak{G}_{M2S}(\chi^\star, \mathcal{C}) - 3\text{OM2S}^\star(\mathcal{X}, \mathcal{C})| & \\
\leq \beta \cdot |\mathfrak{G}_\ell(\text{tok}_\circlearrowleft[\mathcal{S}], \mathcal{D}) - \text{Tok}_\circlearrowleft^{2,\star}(\mathcal{D}, K)|, & \text{ where } & \begin{array}{ll} (\mathcal{D}, K) & = f_{\text{D-2-TOK}}(\mathcal{X}, \mathcal{C}), \\ \chi^\star & = g_{\text{D-2-TOK}}((\mathcal{X}, \mathcal{C}), \mathcal{S}) \end{array}
\end{aligned} \quad (42)
$$

We define the polynomial-time algorithm $g_{\text{D-2-TOK}}$ to map any valid D-2-TOK solution $\mathcal{S}$ of a mapped instance $(\mathcal{D}, K)$ back to a 3-OCC-MAX2SAT assignment $\chi^\star$ for $(\mathcal{X}, \mathcal{C})$. As established in Lemma 2 (Steps 1–4), an arbitrary solution $\mathcal{S}$ may contain sat-noncompliant tokens, which must be systematically switched to compliant ones before extracting truth values.

Let $\mathcal{S}$ be any feasible solution to the mapped instance $(\mathcal{D}, K, \delta) = \text{R1}(\mathcal{X}, \mathcal{C}, \gamma)$. Note that, by the problem definition, these tokenisers have vocabulary size $|\mathcal{S}| = |\Sigma| + K$. We then define:

$$
\mathcal{S}' = \left\{ 1\mathbf{x}_j^{\text{T}}, \, \mathbf{x}_j^{\text{T}}1, \, 1\mathbf{x}_j^{\text{F}}, \, \mathbf{x}_j^{\text{F}}1 \right\}_{j=1}^J, \qquad \mathcal{S}'' = \left\{ 1\mathbf{x}_j^{\text{T}}1, \, 1\mathbf{x}_j^{\text{F}}1 \right\}_{j=1}^J \quad (43)
$$

Using Lemma 2, we construct in polynomial time a vocabulary $\mathcal{S}_{\checkmark}$ that is sat-compliant and satisfies

$$
\mathfrak{G}_\ell(\text{tok}_\circlearrowleft[\mathcal{S}_{\checkmark}], \mathcal{D}) \leq \mathfrak{G}_\ell(\text{tok}_\circlearrowleft[\mathcal{S}], \mathcal{D}). \quad (44)
$$

The construction is as follows:

1. While there exists $t \in \mathcal{S}' \setminus \mathcal{S}$, choose any $u \in \mathcal{S} \setminus \mathcal{S}'$ and set $\mathcal{S} \leftarrow (\mathcal{S} \setminus \{u\}) \cup \{t\}$. By Step ①, this replacement does not increase $\mathfrak{G}_\ell(\text{tok}_\circlearrowleft[\mathcal{S}], \mathcal{D})$.

2. Let $\mathcal{S}_{\text{non-compl}} = \mathcal{S} \setminus (\mathcal{S}' \cup \mathcal{S}'')$. Choose any set $\mathcal{S}_{\text{missing}} \subseteq \mathcal{S}'' \setminus \mathcal{S}$ such that $|\mathcal{S}_{\text{missing}}| = |\mathcal{S}_{\text{non-compl}}|$, and define: $\mathcal{S} \leftarrow (\mathcal{S} \setminus \mathcal{S}_{\text{non-compl}}) \cup \mathcal{S}_{\text{missing}}$. By Step ②, this replacement does not increase $\mathfrak{G}_\ell(\text{tok}_\circlearrowleft[\mathcal{S}], \mathcal{D})$.

3. While there exist indices $j \neq k$ such that $\{1\mathbf{x}_j^{\text{T}}1, \, 1\mathbf{x}_j^{\text{F}}1\} \subseteq \mathcal{S}$ and $\{1\mathbf{x}_k^{\text{T}}1, \, 1\mathbf{x}_k^{\text{F}}1\} \cap \mathcal{S} = \emptyset$, choose any $r \in \{1\mathbf{x}_j^{\text{T}}1, \, 1\mathbf{x}_j^{\text{F}}1\}$ and any $t \in \{1\mathbf{x}_k^{\text{T}}1, \, 1\mathbf{x}_k^{\text{F}}1\}$, and set $\mathcal{S} \leftarrow (\mathcal{S} \setminus \{r\}) \cup \{t\}$. By Step ③, this replacement does not increase $\mathfrak{G}_\ell(\text{tok}_\circlearrowleft[\mathcal{S}], \mathcal{D})$.

Let $\mathcal{S}_{\checkmark}$ be the final vocabulary. By construction, $\mathcal{S}' \subseteq \mathcal{S}_{\checkmark}$, $\mathcal{S}_{\checkmark} \subseteq \mathcal{S}' \cup \mathcal{S}''$, and for each $j$ exactly one of $1\mathbf{x}_j^{\text{T}}1$ or $1\mathbf{x}_j^{\text{F}}1$ belongs to $\mathcal{S}_{\checkmark}$; hence $\mathcal{S}_{\checkmark}$ is sat-compliant, and the objective inequality follows from the three nonincreasing replacements. Further, this polynomial-time transformation guarantees that the total compressed length does not increase, ensuring $\mathfrak{G}_\ell(\text{tok}_\circlearrowleft[\mathcal{S}_{\checkmark}], \mathcal{D}) \leq \mathfrak{G}_\ell(\text{tok}_\circlearrowleft[\mathcal{S}], \mathcal{D})$.

We can now use the algorithm $g$ defined in Eq. (18) to extract the variable assignment $\chi^\star$ directly from the compliant tokens in $\mathcal{S}_{\checkmark}$. Thus, we define $g_{\text{D-2-TOK}}$ as the application of our extraction function to this sat-compliant vocabulary:

$$
g_{\text{D-2-TOK}}((\mathcal{X}, \mathcal{C}), \mathcal{S}) = g(\mathcal{S}_{\checkmark}) = \{x_j\}_{j=1}^J, \text{ where } \begin{array}{ll} x_j = \text{T} & \text{if } 1\mathbf{x}_j^{\text{T}}1 \in \mathcal{S}_{\checkmark} \\ x_j = \text{F} & \text{elif } 1\mathbf{x}_j^{\text{F}}1 \in \mathcal{S}_{\checkmark} \end{array} \quad (45)
$$

From Eqs. (36) and (37), the objective value of the sat-compliant tokeniser $\mathcal{S}_{\checkmark}$ has a linear relation to the value of assignment $\chi^\star$:

$$
\mathfrak{G}_\ell(\text{tok}_\circlearrowleft[\mathcal{S}_{\checkmark}], \mathcal{D}) = 329J + 3I - \mathfrak{G}_{M2S}(\chi^\star, \mathcal{C}) \quad (46)
$$

We can now prove our desired bound as:

$$
\begin{aligned}
\mathfrak{G}_\ell(\text{tok}_\circlearrowleft[\mathcal{S}], \mathcal{D}) - \text{Tok}_\circlearrowleft^{2,\star}(\mathcal{D}, K) & \qquad \text{for } (\mathcal{D}, K) = f_{\text{D-2-TOK}}(\mathcal{X}, \mathcal{C}) \\
\geq \mathfrak{G}_\ell(\text{tok}_\circlearrowleft[\mathcal{S}_{\checkmark}], \mathcal{D}) - \text{Tok}_\circlearrowleft^{2,\star}(\mathcal{D}, K) & \qquad\qquad\qquad\qquad (47a) \\
= (329J + 3I - \mathfrak{G}_{M2S}(\chi^\star, \mathcal{C})) - (329J + 3I - 3\text{OM2S}^\star(\mathcal{X}, \mathcal{C})) & \qquad (47b) \\
= 3\text{OM2S}^\star(\mathcal{X}, \mathcal{C}) - \mathfrak{G}_{M2S}(\chi^\star, \mathcal{C}) & \qquad \text{for } \chi^\star = g_{\text{D-2-TOK}}((\mathcal{X}, \mathcal{C}), \mathcal{S})) \ (47c)
\end{aligned}
$$

Taking absolute values, we see the inequality in Eq. (42)—and thus the second condition for L-reductions—holds with $\beta = 1$. Since 3-OCC-MAX2SAT is APX-complete, this L-reduction completes the proof that D-2-TOK is APX-hard. $\qquad \square$

# E   PROOF THAT THE BINARY DIRECT TOKENISATION GAP PROBLEM IS NP-HARD (STEP 2 OF THEOREM 2)

**Lemma 4.** *The binary direct tokenisation gap problem is* NP-*hard.*

*Proof.* For this proof, we rely on a result by Berman and Karpinski (1998; 1999) that, for specific instances of 3-OCC-MAX2SAT with $I = 2016n$ clauses, it is NP-hard to distinguish whether at least $(2012 - \varepsilon)n$ or at most $(2011 + \varepsilon)n$ of these clauses are satisfiable, for any $\varepsilon > 0$. We will denote this 3-OCC-MAX2SAT gap problem by $3\text{OM2S}(\mathcal{X}, \mathcal{C}, (\gamma^-, \gamma^+))$, with $\gamma^- = (2011 + \varepsilon)n$ and $\gamma^+ = (2012 - \varepsilon)n$. We can now prove NP-hardness of the binary direct tokenisation gap problem by reducing 3-OCC-MAX2SAT's gap problem to it. To this end, we rely on a reduction identical to $\text{R1}(\mathcal{X}, \mathcal{C}, \gamma)$, but where we define:

$$\delta^- = 329J + 3I - \gamma^- \qquad\qquad \delta^+ = 329J + 3I - \gamma^+ \qquad (48)$$
$$= 329J + 3I - \frac{2011 + \varepsilon}{2016}I, \qquad\qquad = 329J + 3I - \frac{2012 - \varepsilon}{2016}I$$

Lemmas 1 and 2 trivially show the validity of this reduction:

$$3\text{OM2S}(\mathcal{X}, \mathcal{C}, (\gamma^-, \gamma^+)) \iff \text{Tok}^2_{\updownarrow}(\mathcal{D}, K, (\delta^-, \delta^+)) \qquad (49)$$

which holds since $3\text{OM2S}(\mathcal{X}, \mathcal{C}, \gamma^+) \iff \text{Tok}^2_{\updownarrow}(\mathcal{D}, K, \delta^+)$ and the same for $\gamma^-$ and $\delta^-$. It is therefore NP-hard to distinguish whether a dataset can be compressed to at most $329J + 3I - \frac{2012 - \varepsilon}{2016}I$ symbols, or if at least $329J + 3I - \frac{2011 + \varepsilon}{2016}I$ symbols remain (with an allowed vocabulary size $K = 5J$). Since each variable occurs exactly three times in 3-OCC-MAX2SAT, we have that $\frac{3}{2}J = I$. We now compute a lower bound on the best achievable compression ratio:

$$\frac{\delta^-}{\delta^+} = \frac{329J + 3I - \frac{2011 + \varepsilon}{2016}I}{329J + 3I - \frac{2012 - \varepsilon}{2016}I} \qquad (50\text{a})$$

$$= \frac{667 - \frac{6033 + 3\varepsilon}{2016}}{667 - \frac{6036 - \varepsilon}{2016}} \qquad (50\text{b})$$

$$= \frac{1338639 - 3\varepsilon}{1338636 + 3\varepsilon} \qquad (50\text{c})$$

$$= \frac{446213 - \varepsilon}{446212 + \varepsilon} \qquad (50\text{d})$$

Thus, binary direct tokenisation cannot be approximated in polynomial time with an approximation ratio better than $\frac{446213}{446212} > 1.000002$, unless $\text{P} = \text{NP}$.

$\square$

# F   PROOF OF FORWARD STEP OF THEOREM 3

We first need another lemma in preparation for the actual proof of this forward step. Note that Reduction 2 produces character-strings $\mathbf{x}^{\text{T}}_j$ and $\mathbf{x}^{\text{F}}_j$, with form $\{0^j \mid 1 \leq j \leq 2J\}$, which our tokeniser must compress. However, a merge-sequence $\mathbf{m} = \bigcirc_{j=1}^{2J-1}[0^j \odot 0]$ does not compress all these targets into a single symbol; character-string $0000$, for instance, would be merged into $\langle 00, 00 \rangle$ by the first merge $0 \odot 0$ in this sequence, and merge $0^3 \odot 0$ would not be applied to it. Thus, we need to describe a more unwieldy merge sequence to achieve this with the same number of merges. We do so in App. F.1, where we show that exactly $2J - 1$ merges are required to compress all these strings into a single symbol. With this, we now prove the forward step of Theorem 3 in the following lemma.

**Lemma 5.** *If a* 3-OCC-MAX2SAT *instance is satisfiable, then the* B-2-TOK *instance output by Reduction 2 is also satisfiable. Formally:* $3\text{OM2S}(\mathcal{X}, \mathcal{C}, \gamma) \implies \text{Tok}^2_{\uparrow}(\text{R1}(\mathcal{X}, \mathcal{C}, \gamma))$.

*Proof.* Assume this $(\mathcal{X}, \mathcal{C}, \gamma)$ instance of the 3-OCC-MAX2SAT decision problem is satisfiable, i.e., that $3\text{OM2S}(\mathcal{X}, \mathcal{C}, \gamma)$ is true. We must prove that in this case, $\text{Tok}^2_{\uparrow}(\text{R1}(\mathcal{X}, \mathcal{C}, \gamma))$ is also true. We

define the following list of merges which, as shown in SubLemma 1, compresses every target of type $\mathbf{x}_j^{\mathtt{T}}$ or $\mathbf{x}_j^{\mathtt{F}}$ into a single token:

$$\mathbf{m}_1 = (1 \odot 1) \circ \underbrace{\bigcirc_{j=1}^{\lfloor \log_2 J \rfloor} [(0^{2^j} \odot 0^{2^j})] \circ \bigcirc_{j=1}^{\lfloor \log_2 J \rfloor} \bigcirc_{j'=1}^{2^j-1} [(0^{2^j} \odot 0^{j'})]}_{2J-1 \text{ merges which compress each } \mathbf{x}_j^{\mathtt{T}} \text{ and } \mathbf{x}_j^{\mathtt{F}} \text{ to a single token}} \tag{51}$$

Note that the merge $1 \odot 1$ is independent of the merges on $0$ and could thus be placed at any point in the sequence. We also define the following lists of merges, which will be included in any satisfying solution to the tokenisation problem:

$$\mathbf{m}_2 = \bigcirc_{j=1}^{J}[(11 \odot \mathbf{x}_j^{\mathtt{F}}), (\mathbf{x}_j^{\mathtt{T}} \odot 11)], \qquad \mathbf{m}_4 = \bigcirc_{j=1}^{J}[(\mathbf{x}_j^{\mathtt{F}} \odot 1), (1 \odot \mathbf{x}_j^{\mathtt{T}})], \tag{52a}$$

$$\mathbf{m}_6 = \bigcirc_{j=1}^{J}[(1 \odot \mathbf{x}_j^{\mathtt{F}}), (\mathbf{x}_j^{\mathtt{T}} \odot 1)] \tag{52b}$$

Now, let $\chi^\star = \{x_j^\star\}_{j=1}^{J}$ be any satisfying solution to the 3-OCC-MAX2SAT instance $(\mathcal{X}, \mathcal{C}, \gamma)$. We define the following instance-specific merges:

$$\mathbf{m}_3 = \bigcirc_{j=1}^{J} \begin{bmatrix} (1 \odot \mathbf{x}_j^{\mathtt{T}}11) & \texttt{if } x_j^\star = \mathtt{T} \\ (11\mathbf{x}_j^{\mathtt{F}} \odot 1) & \texttt{else} \end{bmatrix}, \qquad \mathbf{m}_5 = \bigcirc_{j=1}^{J} \begin{bmatrix} (1\mathbf{x}_j^{\mathtt{T}} \odot 1) & \texttt{if } x_j^\star = \mathtt{T} \\ (1 \odot \mathbf{x}_j^{\mathtt{F}}1) & \texttt{else} \end{bmatrix} \tag{53}$$

In words, we include merges $1 \odot \mathbf{x}_j^{\mathtt{T}}11$ and $1\mathbf{x}_j^{\mathtt{T}} \odot 1$ if $x_j^\star$ is true, or $11\mathbf{x}_j^{\mathtt{F}} \odot 1$ and $1 \odot \mathbf{x}_j^{\mathtt{F}}1$ if $x_j^\star$ is false. We then create a merge sequence by concatenating these lists in order:

$$\mathbf{m} = \mathbf{m}_1 \circ \mathbf{m}_2 \circ \mathbf{m}_3 \circ \mathbf{m}_4 \circ \mathbf{m}_5 \circ \mathbf{m}_6 \tag{54}$$

This gives us a total of $|\mathbf{m}| = K = 10J$ merges. Now we just need to count the symbols output by this solution to check whether the bound is satisfied.

By applying the merges $\mathbf{m}$, each string in $\mathcal{D}_1$ will be compressed into a single symbol, obtaining:

$$\mathfrak{G}_\ell(\mathsf{tok}_\uparrow[\mathbf{m}], \mathcal{D}_1) = (1 + 8J)f \tag{55}$$

For each pair of strings $1\mathbf{x}_j^{\mathtt{T}}1$ and $1\mathbf{x}_j^{\mathtt{F}}1$ in $\mathcal{D}_2$, one is compressed into a single symbol while the other is only compressed to two symbols—the one with $\mathbf{x}_j^{\mathtt{T}}$ is compressed into a single symbol if $x_j^\star = \mathtt{T}$ and the one with $\mathbf{x}_j^{\mathtt{F}}$ otherwise. The same is true for each pair of strings $1\mathbf{x}_j^{\mathtt{T}}11$ and $11\mathbf{x}_j^{\mathtt{F}}1$, also in $\mathcal{D}_2$. We thus have that, for each variable $X_j$, the strings in $\mathcal{D}_2$ will occupy a total of $(1 + 2 + 1 + 2)f'$ symbols, and:

$$\mathfrak{G}_\ell(\mathsf{tok}_\uparrow[\mathbf{m}], \mathcal{D}_2) = 6f'J \tag{56}$$

Similarly, each string in $\mathcal{D}_3$ and $\mathcal{D}_4$ will be compressed into only 2 symbols after this tokeniser is applied to it. We thus have:

$$\mathfrak{G}_\ell(\mathsf{tok}_\uparrow[\mathbf{m}], \mathcal{D}_3) = 4f''J, \qquad \mathfrak{G}_\ell(\mathsf{tok}_\uparrow[\mathbf{m}], \mathcal{D}_4) = 4f'''J \tag{57}$$

Finally, we have the strings in $\mathcal{D}_5$. These strings are constructed such that they will be compressed into 2 symbols if either $L_i^1$ or $L_i^2$ evaluates to $\mathtt{T}$, and kept with 3 symbols otherwise; see Tab. 1 for a detailed simulation of why this is the case. We thus have:

$$\mathfrak{G}_\ell(\mathsf{tok}_\uparrow[\mathbf{m}], \mathcal{D}_5) = \sum_{i=1}^{I} \left( 3 - \mathbb{1} \begin{cases} L_i^1 = X_j & \texttt{and } (1 \odot \mathbf{x}_j^{\mathtt{T}}11), (1\mathbf{x}_j^{\mathtt{T}} \odot 1) \in \mathbf{m} \\ & \text{or} \\ L_i^1 = \neg X_j & \texttt{and } (11\mathbf{x}_j^{\mathtt{F}} \odot 1), (1 \odot \mathbf{x}_j^{\mathtt{F}}1) \in \mathbf{m} \\ & \text{or} \\ L_i^2 = X_{j'} & \texttt{and } (1 \odot \mathbf{x}_{j'}^{\mathtt{T}}11), (1\mathbf{x}_{j'}^{\mathtt{T}} \odot 1) \in \mathbf{m} \\ & \text{or} \\ L_i^2 = \neg X_{j'} & \texttt{and } (11\mathbf{x}_{j'}^{\mathtt{F}} \odot 1), (1 \odot \mathbf{x}_{j'}^{\mathtt{F}}1) \in \mathbf{m} \end{cases} \right) \tag{58a}$$

$$= 3I - \sum_{i=1}^{I} \mathbb{1}_{\chi^\star}\{L_i^1 \vee L_i^2\} \tag{58b}$$

$$\leq 3I - \gamma \tag{58c}$$

where, by construction, we have a merge in our sequence (e.g., $1 \odot \mathbf{x}_j^{\mathtt{T}}11$ or $11\mathbf{x}_j^{\mathtt{F}} \odot 1$) if and only if its value is in a satisfying assignment (e.g., $x_j^\star = \mathtt{T}$ or $x_j^\star = \mathtt{F}$, respectively). Summing together the lengths in Eqs. (55) to (58), we get that:

$$\mathfrak{G}_\ell(\mathsf{tok}_\uparrow[\mathbf{m}], \mathcal{D}) \leq \delta = (1 + 8J)f + (6f' + 4f'' + 4f''')J + 3I - \gamma \tag{59}$$

which concludes the proof. $\qquad \square$

| Assignment | Condition | $c$ | $\mathrm{tok}_\uparrow[\mathbf{m}_1](c)$ | $\mathrm{tok}_\uparrow[\mathbf{m}_1\circ\mathbf{m}_2](c)$ | $\mathrm{tok}_\uparrow[\mathbf{m}_1\circ\mathbf{m}_2\circ\mathbf{m}_3](c)$ | $\mathrm{tok}_\uparrow[\mathbf{m}_1\circ\cdots\circ\mathbf{m}_4](c)$ | $\mathrm{tok}_\uparrow[\mathbf{m}_1\circ\cdots\circ\mathbf{m}_5](c)$ | $|\mathrm{tok}_\uparrow[\mathbf{m}](c)|$ |
|---|---|---|---|---|---|---|---|---|
| $L_i^1 = X_j$ and $L_i^2 = \neg X_{j'}$ | $x_j^\star = T \wedge x_{j'}^\star = T$ | $\langle 1,\underbrace{0,\ldots,0}_{2j-1},1,\underbrace{0,\ldots,0}_{2j'},1\rangle$ | $\langle 1,\mathbf{x}_j^T,1,\mathbf{x}_{j'}^F,1\rangle$ | · | · | $\langle 1\mathbf{x}_j^T,1,\mathbf{x}_{j'}^F,1\rangle$ | $\langle 1\mathbf{x}_j^T,1,\mathbf{x}_{j'}^F,1\rangle$ | 2 |
| | $x_j^\star = F \wedge x_{j'}^\star = T$ | | | | | | $\langle 1\mathbf{x}_j^T,1,\mathbf{x}_{j'}^F,1\rangle$ | 3 |
| | $x_j^\star = T \wedge x_{j'}^\star = F$ | | | | | | $\langle 1\mathbf{x}_j^T,1,\mathbf{x}_{j'}^F,1\rangle$ | 2 |
| | $x_j^\star = F \wedge x_{j'}^\star = F$ | | | | | | $\langle 1\mathbf{x}_j^T,1\mathbf{x}_{j'}^F,1\rangle$ | 2 |
| $L_i^1 = \neg X_j$ and $L_i^2 = X_{j'}$ | $x_j^\star = T \wedge x_{j'}^\star = T$ | $\langle 1,\underbrace{0,\ldots,0}_{2j'-1},1,\underbrace{0,\ldots,0}_{2j},1\rangle$ | $\langle 1,\mathbf{x}_{j'}^T,1,\mathbf{x}_j^F,1\rangle$ | · | · | $\langle 1\mathbf{x}_{j'}^T,1,\mathbf{x}_j^F,1\rangle$ | $\langle 1\mathbf{x}_{j'}^T,1,\mathbf{x}_j^F,1\rangle$ | 2 |
| | $x_j^\star = F \wedge x_{j'}^\star = T$ | | | | | | $\langle 1\mathbf{x}_{j'}^T,1,\mathbf{x}_j^F,1\rangle$ | 2 |
| | $x_j^\star = T \wedge x_{j'}^\star = F$ | | | | | | $\langle 1\mathbf{x}_{j'}^T,1,\mathbf{x}_j^F,1\rangle$ | 3 |
| | $x_j^\star = F \wedge x_{j'}^\star = F$ | | | | | | $\langle 1\mathbf{x}_{j'}^T,1\mathbf{x}_j^F,1\rangle$ | 2 |
| $L_i^1 = \neg X_j$ and $L_i^2 = \neg X_{j'}$ | $x_j^\star = T \wedge x_{j'}^\star = T$ | $\langle 1,\underbrace{0,\ldots,0}_{2j},1,\underbrace{0,\ldots,0}_{2j'},1\rangle$ | $\langle 11,\mathbf{x}_j^F,1,\mathbf{x}_{j'}^F,1\rangle$ | $\langle 11\mathbf{x}_j^F,1,\mathbf{x}_{j'}^F,1\rangle$ | $\langle 11\mathbf{x}_j^F,1,\mathbf{x}_{j'}^F,1\rangle$ | $\langle 11\mathbf{x}_j^F,1,\mathbf{x}_{j'}^F,1\rangle$ | · | 3 |
| | $x_j^\star = F \wedge x_{j'}^\star = T$ | | | | $\langle 11\mathbf{x}_j^F,1,\mathbf{x}_{j'}^F,1\rangle$ | | $\langle 11\mathbf{x}_j^F,1\mathbf{x}_{j'}^F,1\rangle$ | 2 |
| | $x_j^\star = T \wedge x_{j'}^\star = F$ | | | | · | | | 2 |
| | $x_j^\star = F \wedge x_{j'}^\star = F$ | | | $\langle 11\mathbf{x}_j^F,1,\mathbf{x}_{j'}^F,1\rangle$ | $\langle 11\mathbf{x}_j^F,1,\mathbf{x}_{j'}^F,1\rangle$ | | | 2 |
| $L_i^1 = X_j$ and $L_i^2 = X_{j'}$ | $x_j^\star = T \wedge x_{j'}^\star = T$ | $\langle 1,\underbrace{0,\ldots,0}_{2j-1},1,\underbrace{0,\ldots,0}_{2j'-1},1\rangle$ | $\langle 1,\mathbf{x}_j^T,1,\mathbf{x}_{j'}^T,11\rangle$ | $\langle 1,\mathbf{x}_j^T,1\mathbf{x}_{j'}^T,11\rangle$ | $\langle 1\mathbf{x}_j^T,1\mathbf{x}_{j'}^T,11\rangle$ | $\langle 1\mathbf{x}_j^T,1\mathbf{x}_{j'}^T,11\rangle$ | · | 2 |
| | $x_j^\star = F \wedge x_{j'}^\star = T$ | | | | | | | 2 |
| | $x_j^\star = T \wedge x_{j'}^\star = F$ | | | | $\langle 1\mathbf{x}_j^T,1,\mathbf{x}_{j'}^T,11\rangle$ | | | 2 |
| | $x_j^\star = F \wedge x_{j'}^\star = F$ | | | · | | | | 3 |

Table 1: Performance of merges on strings in $\mathcal{D}_5$, adapted from Whittington et al. (2025). The dot symbol · denotes the string not changing under the given merge.

## F.1 Proof that exactly $2J-1$ merges optimally compress the $2J$ $0^j$ strings

**SubLemma 1.** *Given character-strings $\{0^j \mid 1 \le j \le 2J\}$, an optimal bottom-up tokeniser requires exactly $2J-1$ merges to encode all these strings into a single token.*[6]

*Proof.* We establish the result in two steps:

   ① we prove that at *least* $2J-1$ merges are *required* to reduce these strings to a single token;

   ② we prove that $2J-1$ merges are *sufficient* to reduce these character-strings to a single token.

Given these upper and lower bounds, we conclude exactly $2J-1$ merges are required to reduce these character-strings to a single token. This completes the proof. ☐

**SubLemmaProofStep 1.** (Step ①). *Given character-strings $\{0^j \mid 1 \le j \le 2J\}$, an optimal bottom-up tokeniser requires at* least $2J-1$ *merges to encode all these strings into a single token.*

*Proof.* The target set contains $2J$ distinct values, one of which is the base symbol $0$. Whenever a target becomes a single token through a merge, that merge must combine exactly two tokens whose concatenation equals that specific target. Since all targets are distinct, this concatenation cannot simultaneously equal any other target. Hence, a single merge can complete at most one target. It follows that at least $2J-1$ merges are required to reduce all targets to single tokens. ☐

**SubLemmaProofStep 2.** (Step ②). *Given character-strings $\{0^j \mid 1 \le j \le 2J\}$, there exists an explicit merge sequence that reduces all these strings to single tokens in exactly $2J-1$ merges.*

*Proof.* For the matching upper bound, we will construct an explicit merge sequence, composed of two types of merges:

1. **Binary stage.** For each power of two up to $J$—i.e., with $j \in \mathbb{N}$ such that $2^j \le J$—incrementally include binary merges as $0^{2^j} \odot 0^{2^j}$.

2. **Extension stage.** For each power of two up to $J$—i.e., with $j \in \mathbb{N}$ such that $2^j \le J$—incrementally create non-binary merges by merging binary tokens $0^{2^j}$ with non-binary ones $0^{j'}$ (with $j' \in \mathbb{N}$ such that $j' < 2^j$). These merges will thus be $0^{2^j} \odot 0^{j'}$.[7]

These merges are combined as:

$$\underbrace{\bigcirc_{j=1}^{\lfloor \log_2 J \rfloor}[(0^{2^j} \odot 0^{2^j})]}_{\text{binary merges}} \quad \circ \quad \underbrace{\bigcirc_{j=1}^{\lfloor \log_2 J \rfloor}\bigcirc_{j'=1}^{2^j-1}[(0^{2^j} \odot 0^{j'})]}_{\text{extension merges}} \tag{60}$$

---

[6]Note the same is true for both optimal direct tokenisers, and optimal OPE tokenisers (defined in §5.2).

[7]As we will see in the proof, the smallest non-fully merged value always consists of only two symbols, such that the "rightmost" tiebreaker is not necessary.

Note that, as merges are created incrementally, token $0^{2^j}$ always exists when merge $0^{2^j} \circledcirc 0^{2^j}$ is applied. Similarly, for $j' < 2^j$, both token $0^{2^j}$ and $0^{j'}$ will exist when merge $0^{2^j} \circledcirc 0^{j'}$ is applied. Finally, for $j', j'' < 2^j$, tokens $0^{j'}$ and $0^{j'''}$ will both be created before any extension merge with left-side $0^{2^j}$ is applied; thus, a merge $0^{2^j} \circledcirc 0^{j'}$ will never affect subword-string $\langle 0^{2^j}, 0^{j''} \rangle$ or *vice-versa*. After these merges are applied, it is easy to see that each character-string $\{0^j \mid 1 \leq j \leq 2J\}$ will be represented as a single symbol. As the merge-sequence above contains $2J - 1$ merges, this completes the proof. $\quad\square$

## G    PROOF OF BACKWARD STEP OF THEOREM 3

We again start with defining some useful notions and redefine compliant tokenisers.

First, even though this section is about bottom-up tokenisers, we define a term addressing direct tokenisers, meaning this definition describes tokenisers with tokens instead of merges. We will use this definition in our lemma to prove a fact about direct tokenisers, which we later generalise by showing that it applies to bottom-up tokenisers as well. We define a **sat-compliant (direct) tokeniser** to be any tokeniser which: (i) contains all tokens of the form $11, \mathbf{x}_j^\mathsf{T}, \mathbf{x}_j^\mathsf{F}, 1\mathbf{x}_j^\mathsf{T}, \mathbf{x}_j^\mathsf{T}1, 1\mathbf{x}_j^\mathsf{F}, \mathbf{x}_j^\mathsf{F}1, \mathbf{x}_j^\mathsf{T}11, 11\mathbf{x}_j^\mathsf{F}$; and (ii) contains either $1\mathbf{x}_j^\mathsf{T}1, 1\mathbf{x}_j^\mathsf{T}11$ or $1\mathbf{x}_j^\mathsf{F}1, 11\mathbf{x}_j^\mathsf{F}1$ for each $j \in \{1, \dots, J\}$. Otherwise, we call the tokeniser **sat-noncompliant**. Again, given the vocabulary of a `sat`-compliant tokeniser, we can easily build an assignment to a `3-OCC-MAX2SAT` instance with the same function as for direct tokenisation, defined in Eq. (18):

$$g(\mathcal{S}) = \{x_j\}_{j=1}^J, \text{ where } \begin{cases} x_j = \mathsf{T} & \text{if } 1\mathbf{x}_j^\mathsf{T}1 \in \mathcal{S} \\ x_j = \mathsf{F} & \text{elif } 1\mathbf{x}_j^\mathsf{F}1 \in \mathcal{S} \end{cases} \tag{61}$$

We further adapt the definition of 101-strings to also include all character-strings of the form $110^+1$ and $10^+11$. Considering the datasets output by Reduction 2, we know that there are no 101-strings in dataset $\mathcal{D}_1$. Further, we know that each unique 101-string appears in datasets $\mathcal{D}_2, \mathcal{D}_3$, and $\mathcal{D}_4$ exactly $2f', 2f''$, and $2f'''$ times, respectively, and exactly 3 times in $\mathcal{D}_5$ (this is due to us working with the three-occurrences variant of `MAX2SAT` and to the fact that $\mathbf{x}_j^\mathsf{T} = 0^{2j-1}$ and $\mathbf{x}_j^\mathsf{F} = 0^{2j}$). We now prove the following lemma.

**Lemma 6.** *If the `B-2-TOK` instance output by Reduction 2 is satisfiable, the `3-OCC-MAX2SAT` instance which generated it is as well. Formally:* $\mathrm{Tok}_\uparrow^2(\mathrm{R2}(\mathcal{X}, \mathcal{C}, \gamma)) \implies \mathrm{3OM2S}(\mathcal{X}, \mathcal{C}, \gamma)$.

*Proof.* Assume this `B-2-TOK` instance $(\mathcal{D}, K, \delta)$—where $(\mathcal{D}, K, \delta) = \mathrm{R2}(\mathcal{X}, \mathcal{C}, \gamma)$—is satisfiable, i.e., that $\mathrm{Tok}_\uparrow^2(\mathrm{R2}(\mathcal{X}, \mathcal{C}, \gamma))$ evaluates to true. We must prove that, in this case, $\mathrm{3OM2S}(\mathcal{X}, \mathcal{C}, \gamma)$ also evaluates to true. Now, let $\mathbf{m}_{\mathrm{opt}}$ be an arbitrary optimal solution to $(\mathcal{D}, K, \delta)$. We know, by definition, that:

$$\mathrm{Tok}_\uparrow^2(\mathrm{R2}(\mathcal{X}, \mathcal{C}, \gamma)) \iff \left( \mathfrak{G}_\ell(\mathtt{tok}_\uparrow[\mathbf{m}_{\mathrm{opt}}], \mathcal{D}) \leq \delta \right) \tag{62}$$

We can thus prove this lemma by showing the following implication:

$$\left( \mathfrak{G}_\ell(\mathtt{tok}_\uparrow[\mathbf{m}_{\mathrm{opt}}], \mathcal{D}) \leq \delta \right) \implies \mathrm{3OM2S}(\mathcal{X}, \mathcal{C}, \gamma) \tag{63}$$

When comparing two bottom-up tokenisers, things quickly get messy, because we have to not only consider the merges, but also their order. For this reason, we show that `sat`-compliant direct tokenisers can be transformed into bottom-up tokenisers without loss of compression quality. Thus, for the `sat`-compliant tokeniser, we can consider the direct tokeniser instead. The key idea for this to work is that all target strings are hit via a sequence of merges such that each intermediate merge also hits a target (which has high multiplicity), such that this target must also be included as a token by a direct tokeniser. We also compare to `sat`-noncompliant direct tokenisers, which are by definition at least as strong as bottom-up tokenisers; thus we can compute upper bounds on their performance.

Let $\mathcal{S}_{\mathrm{opt}}$ again be the optimal direct tokeniser for $(\mathcal{D}, K, \delta)$. We will proceed in six steps:

① we prove that $\mathcal{S}_{\mathrm{opt}}$ must include all tokens of the form $11, \mathbf{x}_j^\mathsf{T}, \mathbf{x}_j^\mathsf{F}, 1\mathbf{x}_j^\mathsf{T}, \mathbf{x}_j^\mathsf{T}1, 1\mathbf{x}_j^\mathsf{F}, \mathbf{x}_j^\mathsf{F}1, \mathbf{x}_j^\mathsf{T}11, 11\mathbf{x}_j^\mathsf{F}$;

② we prove that $\mathcal{S}_{\text{opt}}$ must, in addition to the tokens above, only include tokens of the form $1\mathbf{x}_j^{\text{T}}1, 1\mathbf{x}_j^{\text{F}}1, 1\mathbf{x}_j^{\text{T}}11, 11\mathbf{x}_j^{\text{F}}1$;

③ we prove that $\mathcal{S}_{\text{opt}}$ may only include, for each $j$, either token $1\mathbf{x}_j^{\text{T}}1$ or $1\mathbf{x}_j^{\text{F}}1$, and either token $11\mathbf{x}_j^{\text{T}}1$ or $1\mathbf{x}_j^{\text{F}}11$;

④ we prove that $\mathcal{S}_{\text{opt}}$ may only include, for each $j$, either tokens $11\mathbf{x}_j^{\text{T}}1, 1\mathbf{x}_j^{\text{T}}1$ or $1\mathbf{x}_j^{\text{F}}1, 1\mathbf{x}_j^{\text{F}}11$

⑤ we prove that for any sat-compliant $\mathcal{S}_{\text{opt}}$, there exists a merge sequence $\mathbf{m}_{\text{opt}}$ with the same performance;

⑥ finally, we prove that if $\left( \mathfrak{G}_\ell(\text{tok}_\uparrow[\mathbf{m}_{\text{opt}}], \mathcal{D}) \leq \delta \right)$, we can build a variable assignment which satisfies this 3-OCC-MAX2SAT instance $(\mathcal{X}, \mathcal{C}, \gamma)$.

Note that, together, Steps ① to ④ show that $\mathcal{S}_{\text{opt}}$ must be the vocabulary of a sat-compliant direct tokeniser; in Step ⑤, we show that we can convert any sat-compliant $\mathcal{S}_{\text{opt}}$ to a merge sequence $\mathbf{m}_{\text{opt}}$ without losing compression, showing an equivalence between the two types of tokenisers for these reduced instances; and in Step ⑥, we will then rely on function $g$ (defined above) to convert this merge sequence into a satisfying assignment $\chi = g(\mathcal{S}_{\text{opt}})$ for the instance $(\mathcal{X}, \mathcal{C}, \gamma)$. □

**LemmaProofStep 1.** (Step ①). *An optimal (direct) tokeniser must include all tokens of the form* $11, \mathbf{x}_j^{\text{T}}, \mathbf{x}_j^{\text{F}}, 1\mathbf{x}_j^{\text{T}}, \mathbf{x}_j^{\text{T}}1, 1\mathbf{x}_j^{\text{F}}, \mathbf{x}_j^{\text{F}}1, \mathbf{x}_j^{\text{T}}11, 11\mathbf{x}_j^{\text{F}}$, *i.e.:*

$$\left\{ 11, \mathbf{x}_j^{\text{T}}, \mathbf{x}_j^{\text{F}}, 1\mathbf{x}_j^{\text{T}}, \mathbf{x}_j^{\text{T}}1, 1\mathbf{x}_j^{\text{F}}, \mathbf{x}_j^{\text{F}}1, \mathbf{x}_j^{\text{T}}11, 11\mathbf{x}_j^{\text{F}} \right\}_{j=1}^{J} \subseteq \mathcal{S}_{\text{opt}} \tag{64}$$

*Proof.* We prove this step by contradiction. Assume there exists an optimal tokeniser with vocabulary $\mathcal{S}_{\textbf{✗}}$ which does not include $t > 0$ of the tokens above. Now, remove $t$ arbitrarily chosen tokens in this vocabulary which are not of the form above, and replace them with the missing tokens in this set. We denote this new tokeniser's vocabulary by $\mathcal{S}_{\textbf{✓}}$. Note that the strings in $\mathcal{D}_1$ with these missing tokens were represented with at least 2 symbols under $\mathcal{S}_{\textbf{✗}}$, but with a single token under $\mathcal{S}_{\textbf{✓}}$, i.e.:

$$\mathfrak{G}_\ell(\text{tok}_\circ[\mathcal{S}_{\textbf{✗}}], \mathcal{D}_1) \geq (8J + 1 + t)f, \qquad \mathfrak{G}_\ell(\text{tok}_\circ[\mathcal{S}_{\textbf{✓}}], \mathcal{D}_1) = (8J + 1)f \tag{65}$$

Further, note that under $\mathcal{S}_{\textbf{✓}}$, we have that strings in dataset $\mathcal{D}_2$ are compressed to at most two symbols, while strings in $\mathcal{D}_3, \mathcal{D}_4$, and $\mathcal{D}_5$ are compressed to at most three symbols:

$$\forall \mathbf{c} \in \mathcal{D}_2: \mathfrak{G}_\ell(\text{tok}_\circ[\mathcal{S}_{\textbf{✓}}], \mathbf{c}) \leq 2, \quad \forall \mathbf{c} \in \mathcal{D}_3 \cup \mathcal{D}_4 \cup \mathcal{D}_5: \mathfrak{G}_\ell(\text{tok}_\circ[\mathcal{S}_{\textbf{✓}}], \mathbf{c}) \leq 3 \tag{66}$$

To improve on this compressed length, $\mathcal{S}_{\textbf{✗}}$ must, consequently, compress strings in $\mathcal{D}_2$ to a single symbol, or strings in $\mathcal{D}_3, \mathcal{D}_4$, and $\mathcal{D}_5$ to one or two symbols. As before, this can only be done if the noncompliant tokens in $\mathcal{S}_{\textbf{✗}}$ contain 101-strings.[8] As discussed above, however, each 101-string appears, as a prefix or suffix, at most: $2f'$ times in $\mathcal{D}_2$, $2f''$ times in $\mathcal{D}_3$, $2f'''$ times in $\mathcal{D}_4$, and 3 times in $\mathcal{D}_5$. This gives us a best case scenario—in which all the strings in which a new token appears are compressed to a single symbol—where:

$$\mathfrak{G}_\ell(\text{tok}_\circ[\mathcal{S}_{\textbf{✓}}], \mathcal{D}_2 \cup \mathcal{D}_3 \cup \mathcal{D}_4 \cup \mathcal{D}_5) - \mathfrak{G}_\ell(\text{tok}_\circ[\mathcal{S}_{\textbf{✗}}], \mathcal{D}_2 \cup \mathcal{D}_3 \cup \mathcal{D}_4 \cup \mathcal{D}_5) \tag{67}$$
$$\leq t(2f' + 2(2f'' + 2f''' + 3))$$

As the difference in Eq. (65) is of at least $tf$ tokens, we put these together:

$$\mathfrak{G}_\ell(\text{tok}_\circ[\mathcal{S}_{\textbf{✓}}], \mathcal{D}) - \mathfrak{G}_\ell(\text{tok}_\circ[\mathcal{S}_{\textbf{✗}}], \mathcal{D}) \leq t(2f' + 2(2f'' + 2f''' + 3)) - tf \tag{68}$$

Since $f > 2(2f' + 2f'' + 2f''' + 3)$, this difference is smaller than zero, implying that $\mathcal{S}_{\textbf{✓}}$ improves on $\mathcal{S}_{\textbf{✗}}$. This shows a contradiction, which completes our proof. □

---

[8]This follows the same argument as in the proof of Lemma 2. Note that the extension of 101-strings to include strings of the form $110^{+}1$ and $10^{+}11$ does not break the argument, as the substring has to appear as a prefix or suffix to yield a saving. Thus, even though the strings of form $10^{+}1$ are included in the new strings, we do not have to count those occurrences.

**LemmaProofStep 2.** (Step ②). *An optimal tokeniser must include all tokens of the form* $11, \mathbf{x}_j^\mathsf{T}, \mathbf{x}_j^\mathsf{F}, 1\mathbf{x}_j^\mathsf{T}, \mathbf{x}_j^\mathsf{T}1, 1\mathbf{x}_j^\mathsf{F}, \mathbf{x}_j^\mathsf{F}1, \mathbf{x}_j^\mathsf{T}11, 11\mathbf{x}_j^\mathsf{F}$, *and further only tokens of the form* $1\mathbf{x}_j^\mathsf{T}1, 1\mathbf{x}_j^\mathsf{F}1, 1\mathbf{x}_j^\mathsf{T}11, 11\mathbf{x}_j^\mathsf{F}1$, *i.e.:*

$$\left\{ 11, \mathbf{x}_j^\mathsf{T}, \mathbf{x}_j^\mathsf{F}, 1\mathbf{x}_j^\mathsf{T}, \mathbf{x}_j^\mathsf{T}1, 1\mathbf{x}_j^\mathsf{F}, \mathbf{x}_j^\mathsf{F}1, \mathbf{x}_j^\mathsf{T}11, 11\mathbf{x}_j^\mathsf{F} \right\}_{j=1}^J \subseteq \mathcal{S}_{\text{opt}} \quad \text{and} \tag{69}$$

$$\mathcal{S}_{\text{opt}} \subset \left\{ 11, \mathbf{x}_j^\mathsf{T}, \mathbf{x}_j^\mathsf{F}, 1\mathbf{x}_j^\mathsf{T}, \mathbf{x}_j^\mathsf{T}1, 1\mathbf{x}_j^\mathsf{F}, \mathbf{x}_j^\mathsf{F}1, \mathbf{x}_j^\mathsf{T}11, 11\mathbf{x}_j^\mathsf{F}, 1\mathbf{x}_j^\mathsf{T}1, 1\mathbf{x}_j^\mathsf{F}1, 1\mathbf{x}_j^\mathsf{T}11, 11\mathbf{x}_j^\mathsf{F}1 \right\}_{j=1}^J$$

*Proof.* As before, we prove this step by contradiction. Given Step ①, we know that an optimal tokeniser includes all tokens $11, \mathbf{x}_j^\mathsf{T}, \mathbf{x}_j^\mathsf{F}, 1\mathbf{x}_j^\mathsf{T}, \mathbf{x}_j^\mathsf{T}1, 1\mathbf{x}_j^\mathsf{F}, \mathbf{x}_j^\mathsf{F}1, \mathbf{x}_j^\mathsf{T}11, 11\mathbf{x}_j^\mathsf{F}$. Now, assume there exists an optimal tokeniser with vocabulary $\mathcal{S}_{\text{✗}}$ with $t > 0$ tokens which are not of the form $11, \mathbf{x}_j^\mathsf{T}, \mathbf{x}_j^\mathsf{F}, 1\mathbf{x}_j^\mathsf{T}, \mathbf{x}_j^\mathsf{T}1, 1\mathbf{x}_j^\mathsf{F}, \mathbf{x}_j^\mathsf{F}1, \mathbf{x}_j^\mathsf{T}11, 11\mathbf{x}_j^\mathsf{F}$ or $1\mathbf{x}_j^\mathsf{T}1, 1\mathbf{x}_j^\mathsf{F}1, 1\mathbf{x}_j^\mathsf{T}11, 11\mathbf{x}_j^\mathsf{F}1$; we will call these tokens non-compliant here. Choose an arbitrary set of $t$ unused compliant tokens—i.e., with form $1\mathbf{x}_j^\mathsf{T}1, 1\mathbf{x}_j^\mathsf{F}1, 1\mathbf{x}_j^\mathsf{T}11, 11\mathbf{x}_j^\mathsf{F}1$—to replace the non-compliant tokens with, forming a new tokeniser's vocabulary $\mathcal{S}_{\text{✓}}$. Both these vocabularies compress strings in $\mathcal{D}_1$ equally:

$$\mathfrak{G}_\ell(\text{tok}_\Leftrightarrow[\mathcal{S}_{\text{✗}}], \mathcal{D}_1) = (8J+1)f, \qquad \mathfrak{G}_\ell(\text{tok}_\Leftrightarrow[\mathcal{S}_{\text{✓}}], \mathcal{D}_1) = (8J+1)f \tag{70}$$

For strings in $\mathcal{D}_2$: if the entire string is in the vocabulary, it is encoded as a single token; else, it is represented with two symbols. Under $\mathcal{S}_{\text{✓}}$, there are $2J$ tokens covering strings in $\mathcal{D}_2$. Under $\mathcal{S}_{\text{✗}}$, there are only $(2J-t)$ tokens covering strings in $\mathcal{D}_2$. This implies:

$$\mathfrak{G}_\ell(\text{tok}_\Leftrightarrow[\mathcal{S}_{\text{✗}}], \mathcal{D}_2) = (6J+t)f', \qquad \mathfrak{G}_\ell(\text{tok}_\Leftrightarrow[\mathcal{S}_{\text{✓}}], \mathcal{D}_2) = 6Jf' \tag{71}$$

For strings in $\mathcal{D}_3, \mathcal{D}_4$, and $\mathcal{D}_5$, an argument similar to the previous step applies: (i) only tokens containing 101-strings can compress these datasets; (ii) each 101-string appears, as a prefix or suffix, at most $2f'' + 2f''' + 3$ times in them; (iii) each 101-string will lead to at most two symbols being saved. As $\mathcal{S}_{\text{✗}}$ differs from $\mathcal{S}_{\text{✓}}$ in $t$ tokens, we get that it will improve on it by at most:

$$\mathfrak{G}_\ell(\text{tok}_\Leftrightarrow[\mathcal{S}_{\text{✓}}], \mathcal{D}_3 \cup \mathcal{D}_4 \cup \mathcal{D}_5) - \mathfrak{G}_\ell(\text{tok}_\Leftrightarrow[\mathcal{S}_{\text{✗}}], \mathcal{D}_3 \cup \mathcal{D}_4 \cup \mathcal{D}_5) \leq 2t(2f'' + 2f''' + 3) \tag{72}$$

Summing together the compression on all datasets, we get that their difference is bounded by:

$$\mathfrak{G}_\ell(\text{tok}_\Leftrightarrow[\mathcal{S}_{\text{✓}}], \mathcal{D}) - \mathfrak{G}_\ell(\text{tok}_\Leftrightarrow[\mathcal{S}_{\text{✗}}], \mathcal{D}) \leq 2t(2f'' + 2f''' + 3) - tf' \tag{73}$$

As $f' > 2(2f'' + 2f''' + 3)$, this difference is smaller than zero, implying that $\mathcal{S}_{\text{✓}}$ improves on $\mathcal{S}_{\text{✗}}$. This shows a contradiction, which completes our proof. □

**LemmaProofStep 3.** (Step ③). *An optimal tokeniser must contain all tokens of the form* $11, \mathbf{x}_j^\mathsf{T}, \mathbf{x}_j^\mathsf{F}, 1\mathbf{x}_j^\mathsf{T}, \mathbf{x}_j^\mathsf{T}1, 1\mathbf{x}_j^\mathsf{F}, \mathbf{x}_j^\mathsf{F}1, \mathbf{x}_j^\mathsf{T}11, 11\mathbf{x}_j^\mathsf{F}$ *and further only tokens of the form* $1\mathbf{x}_j^\mathsf{T}1, 1\mathbf{x}_j^\mathsf{F}1, 1\mathbf{x}_j^\mathsf{T}11, 11\mathbf{x}_j^\mathsf{F}1$, *and for each* $1 \leq j \leq J$, *it must contain exactly one of* $1\mathbf{x}_j^\mathsf{T}1, 1\mathbf{x}_j^\mathsf{F}1$, *and exactly one of* $1\mathbf{x}_j^\mathsf{T}11, 11\mathbf{x}_j^\mathsf{F}1$.

*Proof.* As before, we prove this step by contradiction. Given Step ①, we know an optimal tokeniser includes all tokens $11, \mathbf{x}_j^\mathsf{T}, \mathbf{x}_j^\mathsf{F}, 1\mathbf{x}_j^\mathsf{T}, \mathbf{x}_j^\mathsf{T}1, 1\mathbf{x}_j^\mathsf{F}, \mathbf{x}_j^\mathsf{F}1, \mathbf{x}_j^\mathsf{T}11, 11\mathbf{x}_j^\mathsf{F}$. Further, given Step ②, we know its other tokens all have form $1\mathbf{x}_j^\mathsf{T}1, 1\mathbf{x}_j^\mathsf{F}1, 1\mathbf{x}_j^\mathsf{T}11, 11\mathbf{x}_j^\mathsf{F}1$. Now, assume there exists an optimal tokeniser with vocabulary $\mathcal{S}_{\text{✗}}$ which includes both tokens in a pair $1\mathbf{x}_j^\mathsf{T}1, 1\mathbf{x}_j^\mathsf{F}1$ or $11\mathbf{x}_j^\mathsf{T}1, 1\mathbf{x}_j^\mathsf{F}11$ for $t > 0$ such pairs, and thus neither of those two for $t > 0$ other such pairs. Then, define $\mathcal{S}_{\text{✓}}$ as a vocabulary where the $1\mathbf{x}_j^\mathsf{F}1$ respectively $1\mathbf{x}_j^\mathsf{F}11$ token of all $t$ doubly assigned pairs are replaced with the $1\mathbf{x}_j^\mathsf{T}1$ respectively $11\mathbf{x}_j^\mathsf{T}1$ token of all uncovered pairs.

These two tokenisers achieve the same compression on $\mathcal{D}_1$ and $\mathcal{D}_2$:

$$\mathfrak{G}_\ell(\text{tok}_\Leftrightarrow[\mathcal{S}_{\text{✗}}], \mathcal{D}_1 \cup \mathcal{D}_2) = (8J+1)f + 6Jf', \quad \mathfrak{G}_\ell(\text{tok}_\Leftrightarrow[\mathcal{S}_{\text{✓}}], \mathcal{D}_1 \cup \mathcal{D}_2) = (8J+1)f + 6Jf' \tag{74}$$

The tokeniser with vocabulary $\mathcal{S}_{\text{✓}}$ will then compress each string in $\mathcal{D}_3$ to 2 symbols, while $\mathcal{S}_{\text{✗}}$ will compress the $t$ strings of the form $1\mathbf{x}_j^\mathsf{T}1\mathbf{x}_j^\mathsf{F}1$ or $11\mathbf{x}_j^\mathsf{F}1\mathbf{x}_j^\mathsf{T}11$ for which their respective pair $1\mathbf{x}_j^\mathsf{T}1, 1\mathbf{x}_j^\mathsf{F}1$ or $11\mathbf{x}_j^\mathsf{T}1, 1\mathbf{x}_j^\mathsf{F}11$ is uncovered, to 3 symbols. This will lead to a total compression of:

$$\mathfrak{G}_\ell(\text{tok}_\Leftrightarrow[\mathcal{S}_{\text{✗}}], \mathcal{D}_3) = (4J+t)f'', \qquad \mathfrak{G}_\ell(\text{tok}_\Leftrightarrow[\mathcal{S}_{\text{✓}}], \mathcal{D}_3) = 4Jf'' \tag{75}$$

Finally, the $t$ doubly assigned tokens (which $\mathcal{S}_{\checkmark}$ does not contain) appear at most $f'''$ times in $\mathcal{D}_4$ and three times in $\mathcal{D}_5$, and will lead to at most one symbol being saved, leading to a bound:

$$\mathfrak{G}_\ell(\mathtt{tok}_\looparrowright[\mathcal{S}_{\checkmark}], \mathcal{D}_4 \cup \mathcal{D}_5) - \mathfrak{G}_\ell(\mathtt{tok}_\looparrowright[\mathcal{S}_{\text{\sf X}}], \mathcal{D}_4 \cup \mathcal{D}_5) \leq t(f''' + 3) \tag{76}$$

Putting these compressed lengths together, we get:

$$\mathfrak{G}_\ell(\mathtt{tok}_\looparrowright[\mathcal{S}_{\checkmark}], \mathcal{D}) - \mathfrak{G}_\ell(\mathtt{tok}_\looparrowright[\mathcal{S}_{\text{\sf X}}], \mathcal{D}) \leq t(f''' + 3) - tf'' \tag{77}$$

As $f'' > f''' + 3$, this difference is smaller than zero, implying that $\mathcal{S}_{\checkmark}$ improves on $\mathcal{S}_{\text{\sf X}}$. This shows a contradiction, which completes our proof. $\square$

**LemmaProofStep 4.** (Step ④). *An optimal tokeniser must be* $\mathtt{sat}$*-compliant: it must contain all tokens of the form* $11, \mathbf{x}_j^{\mathrm{T}}, \mathbf{x}_j^{\mathrm{F}}, 1\mathbf{x}_j^{\mathrm{T}}, \mathbf{x}_j^{\mathrm{T}}1, 1\mathbf{x}_j^{\mathrm{F}}, \mathbf{x}_j^{\mathrm{F}}1, \mathbf{x}_j^{\mathrm{T}}11, 11\mathbf{x}_j^{\mathrm{F}}$ *and further only tokens of the form* $1\mathbf{x}_j^{\mathrm{T}}1, 1\mathbf{x}_j^{\mathrm{F}}1, 1\mathbf{x}_j^{\mathrm{T}}11, 11\mathbf{x}_j^{\mathrm{F}}1$*, and for each* $1 \leq j \leq J$*, it must include either* $1\mathbf{x}_j^{\mathrm{T}}1, 1\mathbf{x}_j^{\mathrm{T}}11$ *or* $1\mathbf{x}_j^{\mathrm{F}}1, 11\mathbf{x}_j^{\mathrm{F}}1$.

*Proof.* As before, we prove this step by contradiction. Given Step ①, we know that an optimal tokeniser includes all tokens $11, \mathbf{x}_j^{\mathrm{T}}, \mathbf{x}_j^{\mathrm{F}}, 1\mathbf{x}_j^{\mathrm{T}}, \mathbf{x}_j^{\mathrm{T}}1, 1\mathbf{x}_j^{\mathrm{F}}, \mathbf{x}_j^{\mathrm{F}}1, \mathbf{x}_j^{\mathrm{T}}11, 11\mathbf{x}_j^{\mathrm{F}}$. Further, given Step ②, we know that its other tokens all have form $1\mathbf{x}_j^{\mathrm{T}}1, 1\mathbf{x}_j^{\mathrm{F}}1, 1\mathbf{x}_j^{\mathrm{T}}11, 11\mathbf{x}_j^{\mathrm{F}}1$. Finally, given Step ③, we know that it contains exactly one token for each pair $1\mathbf{x}_j^{\mathrm{T}}1, 1\mathbf{x}_j^{\mathrm{F}}1$ and $1\mathbf{x}_j^{\mathrm{T}}11, 11\mathbf{x}_j^{\mathrm{F}}1$.

Now, assume there exists an optimal tokeniser with vocabulary $\mathcal{S}_{\text{\sf X}}$ which includes both tokens in a pair $1\mathbf{x}_j^{\mathrm{F}}1, 1\mathbf{x}_j^{\mathrm{T}}1$ or $1\mathbf{x}_j^{\mathrm{T}}11, 11\mathbf{x}_j^{\mathrm{F}}1$ for $t > 0$ such pairs, and thus neither of those two for $t > 0$ other such pairs. Then, define $\mathcal{S}_{\checkmark}$ as a vocabulary where the $1\mathbf{x}_j^{\mathrm{F}}1$ respectively, $11\mathbf{x}_j^{\mathrm{F}}1$) token of all $t$ doubly assigned pairs are replaced with the $1\mathbf{x}_j^{\mathrm{T}}1$ (respectively, $1\mathbf{x}_j^{\mathrm{T}}11$) token of all uncovered pairs. These two tokenisers achieve the same compression on $\mathcal{D}_1, \mathcal{D}_2$, and $\mathcal{D}_3$:

$$\mathfrak{G}_\ell(\mathtt{tok}_\looparrowright[\mathcal{S}_{\text{\sf X}}], \mathcal{D}_1 \cup \mathcal{D}_2 \cup \mathcal{D}_3) = (8J + 1)f + 6Jf' + 4Jf'' \tag{78}$$

$$\mathfrak{G}_\ell(\mathtt{tok}_\looparrowright[\mathcal{S}_{\checkmark}], \mathcal{D}_1 \cup \mathcal{D}_2 \cup \mathcal{D}_3) = (8J + 1)f + 6Jf' + 4Jf'' \tag{79}$$

The tokeniser with vocabulary $\mathcal{S}_{\checkmark}$ will then compress each string in $\mathcal{D}_4$ to 2 symbols, while $\mathcal{S}_{\text{\sf X}}$ will only compress the $t$ strings of the form $1\mathbf{x}_j^{\mathrm{F}}1\mathbf{x}_j^{\mathrm{T}}11$ or $11\mathbf{x}_j^{\mathrm{F}}1\mathbf{x}_j^{\mathrm{T}}1$, for which the pair is uncovered, to 3 symbols. This will lead to a total compression of:

$$\mathfrak{G}_\ell(\mathtt{tok}_\looparrowright[\mathcal{S}_{\text{\sf X}}], \mathcal{D}_4) = (4J + t)f''', \qquad \mathfrak{G}_\ell(\mathtt{tok}_\looparrowright[\mathcal{S}_{\checkmark}], \mathcal{D}_4) = 4Jf''' \tag{80}$$

Finally, the $t$ doubly assigned tokens of the form $1\mathbf{x}_j^{\mathrm{F}}1$ or $11\mathbf{x}_j^{\mathrm{F}}1$ (which $\mathcal{S}_{\checkmark}$ does not contain) appear, as a prefix or suffix, at most three times in $\mathcal{D}_5$, and will lead to at most one symbol being saved, leading to a bound:

$$\mathfrak{G}_\ell(\mathtt{tok}_\looparrowright[\mathcal{S}_{\checkmark}], \mathcal{D}_5) - \mathfrak{G}_\ell(\mathtt{tok}_\looparrowright[\mathcal{S}_{\text{\sf X}}], \mathcal{D}_5) \leq 3t \tag{81}$$

Putting these compressed lengths together, we get:

$$\mathfrak{G}_\ell(\mathtt{tok}_\looparrowright[\mathcal{S}_{\checkmark}], \mathcal{D}) - \mathfrak{G}_\ell(\mathtt{tok}_\looparrowright[\mathcal{S}_{\text{\sf X}}], \mathcal{D}) \leq 3t - tf''' \tag{82}$$

As $f''' > 3$, this difference is smaller than zero, implying that $\mathcal{S}_{\checkmark}$ improves on $\mathcal{S}_{\text{\sf X}}$. This shows a contradiction, which completes our proof. $\square$

**LemmaProofStep 5.** (Step ⑤). *A* $\mathtt{sat}$*-compliant direct tokeniser can be transformed into a bottom-up tokeniser without changing its performance. As any optimal direct tokeniser is* $\mathtt{sat}$*-compliant, this implies:*

$$\mathrm{Tok}_\uparrow^2(\mathrm{R2}(\mathcal{X}, \mathcal{C}, \gamma)) \iff \mathrm{Tok}_\looparrowright^2(\mathrm{R2}(\mathcal{X}, \mathcal{C}, \gamma)) \tag{83}$$

*Proof.* As every bottom-up tokeniser can be interpreted as a direct tokeniser with the same vocabulary size and a possibly suboptimal application of its tokens, it holds that:

$$\mathrm{Tok}_\uparrow^2(\mathrm{R2}(\mathcal{X}, \mathcal{C}, \gamma)) \implies \mathrm{Tok}_\looparrowright^2(\mathrm{R2}(\mathcal{X}, \mathcal{C}, \gamma)) \tag{84}$$

This is to say, direct tokenisers always compress at least as well as bottom-up tokenisers, when allowed the same vocabulary size.

Given a `sat`-compliant direct tokeniser, we first describe how to transform it into a bottom-up tokeniser. We always include merges:

$$\mathbf{m}_1 = (1 \odot 1) \circ \bigcirc_{j=1}^{\lfloor \log(2J-1) \rfloor} [(0^{2^i} \odot 0^{2^i})] \circ \bigcirc_{j=1}^{\lfloor \log(2J-1) \rfloor} \bigcirc_{j'=1}^{2^j-1} [(0^{2^j} \odot 0^{j'})], \tag{85a}$$

$$\mathbf{m}_2 = \bigcirc_{j=1}^{J} [(11 \odot \mathbf{x}_j^{\mathrm{F}}), (\mathbf{x}_j^{\mathrm{T}} \odot 11)], \tag{85b}$$

$$\mathbf{m}_4 = \bigcirc_{j=1}^{J} [(\mathbf{x}_j^{\mathrm{F}} \odot 1), (1 \odot \mathbf{x}_j^{\mathrm{T}})], \tag{85c}$$

$$\mathbf{m}_6 = \bigcirc_{j=1}^{J} [(1 \odot \mathbf{x}_j^{\mathrm{F}}), (\mathbf{x}_j^{\mathrm{T}} \odot 1)] \tag{85d}$$

and additionally, depending on which tokens are included in the direct tokeniser's vocabulary $\mathcal{S}$:

$$\mathbf{m}_3 = \bigcirc_{j=1}^{J} \left[ \begin{array}{ll} (1 \odot \mathbf{x}_j^{\mathrm{T}} 11) & \text{if } 1\mathbf{x}_j^{\mathrm{T}}11 \in \mathcal{S} \\ (11\mathbf{x}_j^{\mathrm{F}} \odot 1) & \text{else} \end{array} \right], \tag{85e}$$

$$\mathbf{m}_5 = \bigcirc_{j=1}^{J} \left[ \begin{array}{ll} (1\mathbf{x}_j^{\mathrm{T}} \odot 1) & \text{if } 1\mathbf{x}_j^{\mathrm{T}}1 \in \mathcal{S} \\ (1 \odot \mathbf{x}_j^{\mathrm{F}}1) & \text{else} \end{array} \right] \tag{85f}$$

Note that since the tokeniser is `sat`-compliant, we have:

$$1\mathbf{x}_j^{\mathrm{T}}11 \in \mathcal{S} \iff 1\mathbf{x}_j^{\mathrm{T}}1 \in \mathcal{S} \tag{86}$$

It is easy to verify that the resulting bottom-up tokeniser has the same performance on datasets $\mathcal{D}_1$ to $\mathcal{D}_4$. For $\mathcal{D}_5$, Tab. 1 shows that each string which could be reduced to two symbols by the direct tokeniser is also reduced to two symbols by the bottom-up tokeniser. Thus, we get:

$$\mathrm{Tok}_{\uparrow}^2(\mathrm{R2}(\mathcal{X}, \mathcal{C}, \gamma)) \iff \mathrm{Tok}_{\looparrowright}^2(\mathrm{R2}(\mathcal{X}, \mathcal{C}, \gamma)) \tag{87}$$

which completes this proof. $\qquad \square$

**LemmaProofStep 6.** (Step ⑥). *If an optimal (direct) tokeniser achieves a compressed length of at most $5398J + 575 + 3I - \gamma$, the original* `3-OCC-MAX2SAT` *instance is satisfiable, i.e.:*

$$\left( \mathfrak{G}_{\ell}(\mathrm{tok}_{\looparrowright}[\mathcal{S}_{\mathsf{opt}}], \mathcal{D}) \le 5398J + 575 + 3I - \gamma \right) \implies \mathrm{3OM2S}(\mathcal{X}, \mathcal{C}, \gamma) \tag{88}$$

*Proof.* Given Steps ① to ④, we know that an optimal tokeniser will be `sat`-compliant. We will now denote this optimal tokeniser's vocabulary by $\mathcal{S}_{\mathsf{opt}}$ and use Eq. (18) to extract a `3-OCC-MAX2SAT` assignment $\chi^{\star} = g(\mathcal{S}_{\mathsf{opt}})$ which corresponds to this tokeniser's vocabulary. From the previous proof steps, we know that any `sat`-compliant tokeniser achieves the following compressed length in $\mathcal{D}_1$, $\mathcal{D}_2$, $\mathcal{D}_3$, and $\mathcal{D}_4$:

$$\mathfrak{G}_{\ell}(\mathrm{tok}_{\looparrowright}[\mathcal{S}_{\mathsf{opt}}], \mathcal{D}_1 \cup \mathcal{D}_2 \cup \mathcal{D}_3 \cup \mathcal{D}_4) = (8J + 1)f + 6Jf' + 4Jf'' + 4Jf''' \tag{89a}$$

$$= (8 \cdot 575 + 6 \cdot 115 + 4 \cdot 23 + 4 \cdot 4)J + 575 \tag{89b}$$

$$= 5398J + 575 \tag{89c}$$

Now, note that any target string in $\mathcal{D}_5$ will be: (i) compressed to two symbols either if $L_i^1$ or $L_i^2$ is $X_j$ and tokens $1\mathbf{x}_j^{\mathrm{T}}1$ and $1\mathbf{x}_j^{\mathrm{T}}11$ exist or if $L_i^1$ or $L_i^2$ is $\neg X_j$ and tokens $1\mathbf{x}_j^{\mathrm{F}}1$ and $11\mathbf{x}_j^{\mathrm{F}}1$ exist; or (ii) compressed to three symbols if neither case is satisfied. Equivalently, a clause $L_i^1 \lor L_i^2$ in `3-OCC-MAX2SAT` is: satisfied if either $L_i^1$ or $L_i^2$ evaluates to true; or not satisfied if both evaluate to false. Given our construction of function $g$ above, one of `3-OCC-MAX2SAT`'s clauses will be satisfied if and only if its corresponding string in $\mathcal{D}_4$ is compressed to two symbols. We can thus state that:

$$\left( \mathfrak{G}_{\ell}(\mathrm{tok}_{\looparrowright}[\mathcal{S}_{\mathsf{opt}}], \mathcal{D}_5) = 3I - \gamma^{\star} \right) \iff \left( \sum_{i=1}^{I} \mathbb{1}_{\chi^{\star}}\{L_i^1 \lor L_i^2\} = \gamma^{\star} \right) \tag{90}$$

Given the construction of $\delta$ as $5398J + 575 + 3I - \gamma$, we conclude that a `sat`-compliant tokeniser which compresses the full dataset to at most that size can be mapped to a `3-OCC-MAX2SAT` assignment which satisfies at least $\gamma$ clauses. This concludes the proof. $\qquad \square$

# H PROOF THAT BINARY BOTTOM-UP TOKENISATION IS APX-HARD (STEP 1 OF THEOREM 4)

We prove APX-hardness of `B-2-TOK` by providing an L-reduction (Definition 4) from the APX-complete problem `3-OCC-MAX2SAT` (Berman and Karpinski, 1999). For the instance mapping $f_{\text{D-2-TOK}}$, we reuse the construction of Reduction 2. To complete the L-reduction, we define a reconstruction mapping $g_{\text{D-2-TOK}}$ that converts any feasible merge sequence (or induced tokeniser) for the `B-2-TOK` instance into a Boolean assignment, and we relate objective gaps via the `3-OCC-MAX2SAT` cost function defined in Eq. (38), which we repeat here for convenience:

$$\mathfrak{G}_{M2S}(\mathcal{X}, \mathcal{C}) = \sum_{i=1}^{I} \mathbb{1}_{\mathcal{X}}\{L_i^1 \vee L_i^2\}. \tag{91}$$

**Lemma 7.** *The bottom-up binary tokenisation problem is* APX-*hard.*

*Proof.* We prove that `B-2-TOK` is APX-hard by constructing an L-reduction from the APX-complete problem `3-OCC-MAX2SAT`. Let $(\mathcal{X}, \mathcal{C})$ be an arbitrary instance of `3-OCC-MAX2SAT`. To establish the L-reduction, we must define a pair of polynomial-time algorithms, $f_{\text{B-2-TOK}}$ and $g_{\text{B-2-TOK}}$, that map instances and solutions between the two problems, and find constants $\alpha$ and $\beta$ such that the conditions in App. A are satisfied. Below, we prove these conditions hold for $\alpha = 7778$ and $\beta = 1$.

**L-Reduction's Condition (i) is Satisfied.** First, we want to prove that algorithm $f_{\text{B-2-TOK}}$ produces an instance $(\mathcal{D}, K) = f_{\text{B-2-TOK}}(\mathcal{X}, \mathcal{C})$ of `B-2-TOK` such that

$$\text{Tok}_{\uparrow}^{2,\star}(f_{\text{B-2-TOK}}(\mathcal{X}, \mathcal{C})) \leq \alpha \cdot 3\text{OM2S}^{\star}(\mathcal{X}, \mathcal{C}). \tag{92}$$

We define $f_{\text{B-2-TOK}}$ using the polynomial-time construction detailed in Reduction 2. Since this mapping is formulated for the decision problem as $\text{R2}(\mathcal{X}, \mathcal{C}, \gamma) = (\mathcal{D}, K, \delta)$, we adapt it for the optimisation setting by discarding the target values (i.e., $\gamma$ and $\delta$) from the function definition. Formally, we thus define $f_{\text{B-2-TOK}}(\mathcal{X}, \mathcal{C}) = (\mathcal{D}, K)$, where the dataset $\mathcal{D} = \mathcal{D}_1 \cup \mathcal{D}_2 \cup \mathcal{D}_3 \cup \mathcal{D}_4 \cup \mathcal{D}_5$ and the token budget $K = 10J$ remain identical to the original construction.

As shown by Eqs. (89) and (90) in the previous section, for any `sat`-compliant tokeniser, the size of the dataset tokenised with $\mathbf{m}_{\checkmark}$ is linearly related to the number of satisfied clauses $\gamma^{\star}$ by the boolean assignment it corresponds to $\mathcal{X}^{\star} = g(\mathcal{S}_{\mathbf{m}_{\checkmark}})$, where $g$ is defined in Eq. (61):

$$\mathfrak{G}_{\ell}(\text{tok}_{\uparrow}[\mathbf{m}_{\checkmark}], \mathcal{D}) = \mathfrak{G}_{\ell}(\text{tok}_{\hookrightarrow}[\mathcal{S}_{\mathbf{m}_{\checkmark}}], \mathcal{D}) \qquad \text{by Lemma 6 Step ⑤} \tag{93}$$

$$= 5398J + 575 + 3I - \gamma^{\star} \qquad \text{by Lemma 6 Step ⑥} \tag{94}$$

As an optimal tokeniser must be `sat`-compliant (by Lemma 6 Step ①–④), it follows that the compression it achieves is

$$\text{Tok}_{\uparrow}^{2,\star}(\mathcal{D}, K) = 5398J + 575 + 3I - 3\text{OM2S}^{\star}(\mathcal{X}, \mathcal{C}) \tag{95}$$

By the definition of `3-OCC-MAX2SAT`, each variable in $\mathcal{X}$ appears exactly three times in the clauses $\mathcal{C}$, and each clause contains exactly two literals; this implies $3J = 2I$. Furthermore, it is trivial to find an assignment that satisfies at least half the clauses. We now thus restrict ourselves to the non-trivial subset of instances with: $3\text{OM2S}^{\star}(\mathcal{X}, \mathcal{C}) \geq \frac{1}{2}I$. We also note that we can always satisfy at least one clause: $3\text{OM2S}^{\star}(\mathcal{X}, \mathcal{C}) \geq 1$. We can now manipulate the optimality equation to get:

$$\text{Tok}_{\uparrow}^{2,\star}(\mathcal{D}, K) = 5398J + 575 + 3I - 3\text{OM2S}^{\star}(\mathcal{X}, \mathcal{C}) \tag{96a}$$

$$= 5398\left(\frac{2}{3}I\right) + 575 + 3I - 3\text{OM2S}^{\star}(\mathcal{X}, \mathcal{C}) \qquad \text{since } J = \frac{2}{3}I \tag{96b}$$

$$\leq \frac{10805}{3}I + 574\, 3\text{OM2S}^{\star}(\mathcal{X}, \mathcal{C}) \qquad \text{since } 1 \leq 3\text{OM2S}^{\star}(\mathcal{X}, \mathcal{C}) \tag{96c}$$

$$\leq \frac{21610}{3}3\text{OM2S}^{\star}(\mathcal{X}, \mathcal{C}) + 574\, 3\text{OM2S}^{\star}(\mathcal{X}, \mathcal{C}) \qquad \text{since } I \leq 2\, 3\text{OM2S}^{\star}(\mathcal{X}, \mathcal{C}) \tag{96d}$$

$$= \left(\frac{21610}{3} + 574\right) 3\text{OM2S}^{\star}(\mathcal{X}, \mathcal{C}) \tag{96e}$$

$$\leq 7778 \cdot 3\text{OM2S}^{\star}(\mathcal{X}, \mathcal{C}) \tag{96f}$$

Thus, the first condition of the L-reduction holds with $\alpha = 7778$.

**L-Reduction's Condition (ii) is Satisfied.** Second, let $(\mathcal{D}, K) = f_{\text{B-2-TOK}}(\mathcal{X}, \mathcal{C})$, we want to build an algorithm $g_{\text{B-2-TOK}}$ which maps any solution $\mathbf{m}$ of $(\mathcal{D}, K)$ to a solution $\chi^\star$ of $(\mathcal{X}, \mathcal{C})$ such that

$$
\begin{aligned}
&|\mathfrak{G}_{M2S}(\chi^\star, \mathcal{C}) - 3\text{OM2S}^\star(\mathcal{X}, \mathcal{C})| \\
&\quad \le \ \beta \cdot |\mathfrak{G}_\ell(\text{tok}_\uparrow[\mathbf{m}], \mathcal{D}) - \text{Tok}_\uparrow^{2,\star}(\mathcal{D}, K)|,
\end{aligned}
\quad \text{where} \quad
\begin{aligned}
(\mathcal{D}, K) &= f_{\text{B-2-TOK}}(\mathcal{X}, \mathcal{C}), \\
\chi^\star &= g_{\text{B-2-TOK}}((\mathcal{X}, \mathcal{C}), \mathbf{m})
\end{aligned}
\tag{97}
$$

We will now define the polynomial-time reconstruction algorithm $g_{\text{B-2-TOK}}$ to map any valid B-2-TOK solution—i.e., a merge sequence $\mathbf{m}$ for the reduced 3-OCC-MAX2SAT instance $(\mathcal{D}, K)$—back to a Boolean assignment $\chi^\star$ with bounded quality.

First, remember that bottom-up solutions are specified by merges rather than an explicit vocabulary. The first step in our construction will thus be to map these $\mathbf{m}$ to their induced token set $\mathcal{S}_\mathbf{m}$ (the alphabet $\Sigma$ together with every token created by a merge in $\mathbf{m}$). Notably, this transformation is cost-nonincreasing, i.e.,:

$$
\mathfrak{G}_\ell(\text{tok}_\uparrow[\mathbf{m}], \mathcal{D}) \ge \mathfrak{G}_\ell(\text{tok}_\diamond[\mathcal{S}_\mathbf{m}], \mathcal{D})
\tag{98}
$$

Notably, an induced vocabulary may contain sat-noncompliant tokens. We now transform this vocabulary $\mathcal{S}_\mathbf{m}$ into sat-compliant via the cost-nonincreasing replacement procedures of Lemma 6's Steps ①–④. This will later allow us to extract a Boolean assignments $\chi^\star$ with bounded quality from this sat-compliant vocabulary. Before defining our cost-nonincreasing transformations, however, we define the following sub-vocabularies:

$$
\mathcal{S}' = \left\{ 11, \mathbf{x}_j^\text{T}, \mathbf{x}_j^\text{F}, 1\mathbf{x}_j^\text{T}, \mathbf{x}_j^\text{T}1, 1\mathbf{x}_j^\text{F}, \mathbf{x}_j^\text{F}1, \mathbf{x}_j^\text{T}11, 11\mathbf{x}_j^\text{F} \right\}_{j=1}^J,
\tag{99a}
$$

$$
\mathcal{S}'' = \left\{ 1\mathbf{x}_j^\text{T}1, 1\mathbf{x}_j^\text{F}1, 1\mathbf{x}_j^\text{T}11, 11\mathbf{x}_j^\text{F}1 \right\}_{j=1}^J
\tag{99b}
$$

We can now apply the following cost-nonincreasing steps to make the vocabulary sat-compliant:

1. While there exists $t \in \mathcal{S}' \setminus \mathcal{S}_\mathbf{m}$, choose any $u \in \mathcal{S}_\mathbf{m} \setminus \mathcal{S}'$ and set $\mathcal{S}_\mathbf{m} \leftarrow (\mathcal{S}_\mathbf{m} \setminus \{u\}) \cup \{t\}$. By Step ①, this replacement does not increase $\mathfrak{G}_\ell(\text{tok}_\diamond[\mathcal{S}_\mathbf{m}], \mathcal{D})$.

2. Let $\mathcal{S}_{\text{non-compl}} = \mathcal{S}_\mathbf{m} \setminus (\mathcal{S}' \cup \mathcal{S}'')$. Choose any set $\mathcal{S}_{\text{missing}} \subseteq \mathcal{S}'' \setminus \mathcal{S}_\mathbf{m}$ such that $|\mathcal{S}_{\text{missing}}| = |\mathcal{S}_{\text{non-compl}}|$, and set $\mathcal{S}_\mathbf{m} \leftarrow (\mathcal{S}_\mathbf{m} \setminus \mathcal{S}_{\text{non-compl}}) \cup \mathcal{S}_{\text{missing}}$. By Step ②, this replacement does not increase $\mathfrak{G}_\ell(\text{tok}_\diamond[\mathcal{S}_\mathbf{m}], \mathcal{D})$.

3. While there exist indices $j \ne k$ such that $\{1\mathbf{x}_j^\text{T}1, 1\mathbf{x}_j^\text{F}1\} \subseteq \mathcal{S}_\mathbf{m}$ and $\{1\mathbf{x}_k^\text{T}1, 1\mathbf{x}_k^\text{F}1\} \cap \mathcal{S}_\mathbf{m} = \emptyset$, choose any $r \in \{1\mathbf{x}_j^\text{T}1, 1\mathbf{x}_j^\text{F}1\}$ and any $t \in \{1\mathbf{x}_k^\text{T}1, 1\mathbf{x}_k^\text{F}1\}$, and set $\mathcal{S}_\mathbf{m} \leftarrow (\mathcal{S}_\mathbf{m} \setminus \{r\}) \cup \{t\}$. By Step ③, this replacement does not increase $\mathfrak{G}_\ell(\text{tok}_\diamond[\mathcal{S}_\mathbf{m}], \mathcal{D})$.

4. While there exist indices $j \ne k$ such that $\{1\mathbf{x}_j^\text{T}11, 11\mathbf{x}_j^\text{F}1\} \subseteq \mathcal{S}_\mathbf{m}$ and $\{1\mathbf{x}_k^\text{T}11, 11\mathbf{x}_k^\text{F}1\} \cap \mathcal{S}_\mathbf{m} = \emptyset$, choose any $r \in \{1\mathbf{x}_j^\text{T}11, 11\mathbf{x}_j^\text{F}1\}$ and any $t \in \{1\mathbf{x}_k^\text{T}11, 11\mathbf{x}_k^\text{F}1\}$, and set $\mathcal{S}_\mathbf{m} \leftarrow (\mathcal{S}_\mathbf{m} \setminus \{r\}) \cup \{t\}$. By Step ③, this replacement does not increase $\mathfrak{G}_\ell(\text{tok}_\diamond[\mathcal{S}_\mathbf{m}], \mathcal{D})$.

5. While there exist an index $j$ such that $\{1\mathbf{x}_j^\text{T}1, 11\mathbf{x}_j^\text{F}1\} \subset \mathcal{S}_\mathbf{m}$, set $\mathcal{S}_\mathbf{m} \leftarrow (\mathcal{S}_\mathbf{m} \setminus \{11\mathbf{x}_j^\text{F}1\}) \cup \{1\mathbf{x}_j^\text{T}11\}$ Similarly, if $\{1\mathbf{x}_j^\text{F}1, 1\mathbf{x}_j^\text{T}11\} \subset \mathcal{S}_\mathbf{m}$, set $\mathcal{S}_\mathbf{m} \leftarrow (\mathcal{S}_\mathbf{m} \setminus \{1\mathbf{x}_j^\text{T}11\}) \cup \{11\mathbf{x}_j^\text{F}1\}$. By Step ④, this replacement does not increase $\mathfrak{G}_\ell(\text{tok}_\diamond[\mathcal{S}_\mathbf{m}], \mathcal{D})$.

Now, let $\mathcal{S}_{\mathbf{m}\checkmark}$ be the vocabulary resulting from these transformations. By Lemma 6 (Steps ①–④), $\mathcal{S}_{\mathbf{m}\checkmark}$ is sat-compliant (for the bottom-up gadget), and thus for each $j$ it contains either $\{1\mathbf{x}_j^\text{T}1, 1\mathbf{x}_j^\text{T}11\}$ or $\{1\mathbf{x}_j^\text{F}1, 11\mathbf{x}_j^\text{F}1\}$. Further, by this same Lemma, the cost of $\mathcal{S}_{\mathbf{m}\checkmark}$ is smaller or equal to $\mathcal{S}_\mathbf{m}$, as each transformation is cost-nonincreasing. We now define an algorithm $g_{\text{B-2-TOK}}$ which, given a merge-sequence $\mathbf{m}$, transforms it into $\mathcal{S}_{\mathbf{m}\checkmark}$ and then extracts the variable assignment $\chi^\star$ from these sat-compliant tokens:

$$
g_{\text{B-2-TOK}}((\mathcal{X}, \mathcal{C}), \mathbf{m}) = g(\mathcal{S}_{\mathbf{m}\checkmark}) = \{x_j\}_{j=1}^J, \quad \text{where} \quad
\begin{aligned}
x_j &= \text{T} && \text{if } 1\mathbf{x}_j^\text{T}1 \in \mathcal{S}_{\mathbf{m}\checkmark} \\
x_j &= \text{F} && \text{elif } 1\mathbf{x}_j^\text{F}1 \in \mathcal{S}_{\mathbf{m}\checkmark}
\end{aligned}
\tag{100}
$$

Finally, from Eq. (94), the cost of a tokeniser with sat-compliant tokens $\mathcal{S}_{\mathbf{m}\checkmark}$ has a a linear relationship with the cost of assignment $\chi^\star = g(\mathcal{S}_{\mathbf{m}\checkmark})$:

$$\mathfrak{G}_\ell(\text{tok}_\Rightarrow[\mathcal{S}_{\mathbf{m}\checkmark}], \mathcal{D}) = 5398J + 575 + 3I - \mathfrak{G}_{M2S}(\chi^\star, \mathcal{C}) \tag{101}$$

Now, with some simple operations:

$$\mathfrak{G}_\ell(\text{tok}_\uparrow[\mathbf{m}], \mathcal{D}) - \text{Tok}_\uparrow^{2,\star}(\mathcal{D}, K) \tag{102a}$$

$$\geq \mathfrak{G}_\ell(\text{tok}_\Rightarrow[\mathcal{S}_{\mathbf{m}}], \mathcal{D}) - \text{Tok}_\uparrow^{2,\star}(\mathcal{D}, K) \qquad \text{tok}_\Rightarrow \text{ applies tokens optimally} \tag{102b}$$

$$\geq \mathfrak{G}_\ell(\text{tok}_\Rightarrow[\mathcal{S}_{\mathbf{m}\checkmark}], \mathcal{D}) - \text{Tok}_\uparrow^{2,\star}(\mathcal{D}, K) \qquad \text{by Lemma 6 Steps ①–④} \tag{102c}$$

$$= (5398J + 575 + 3I - \mathfrak{G}_{M2S}(\chi^\star, \mathcal{C})) \tag{102d}$$
$$\quad - (5398J + 575 + 3I - 3\text{OM2S}^\star(\mathcal{X}, \mathcal{C}))$$

$$= 3\text{OM2S}^\star(\mathcal{X}, \mathcal{C}) - \mathfrak{G}_{M2S}(\chi^\star, \mathcal{C}) \tag{102e}$$

Taking absolute values, we get:

$$|3\text{OM2S}^\star(\mathcal{X}, \mathcal{C}) - \mathfrak{G}_{M2S}(\chi^\star, \mathcal{C})| \leq 1 \cdot |\mathfrak{G}_\ell(\text{tok}_\uparrow[\mathbf{m}], \mathcal{D}) - \text{Tok}_\uparrow^{2,\star}(\mathcal{D}, K)| \tag{103}$$

This confirms that the second condition holds with $\beta = 1$. Since 3-OCC-MAX2SAT is APX-complete, this L-reduction to B-2-TOK proves that B-2-TOK is APX-hard. $\qquad\square$

## I  PROOF THAT THE BINARY BOTTOM-UP TOKENISATION GAP PROBLEM IS NP-HARD (STEP 2 OF THEOREM 4)

**Lemma 8.** *The binary bottom-up tokenisation gap problem is* NP-*hard.*

*Proof.* For this proof, we again rely on the result by Berman and Karpinski (1998; 1999) that, for specific instances of 3-OCC-MAX2SAT with $I = 2016n$ clauses, it is NP-hard to distinguish whether at least $(2012 - \varepsilon)n$ or at most $(2011 + \varepsilon)n$ of these clauses are satisfiable, for any $\varepsilon > 0$. We will denote this 3-OCC-MAX2SAT gap problem by $3\text{OM2S}(\mathcal{X}, \mathcal{C}, (\gamma^-, \gamma^+))$, with $\gamma^- = (2011 + \varepsilon)n$ and $\gamma^+ = (2012 - \varepsilon)n$. We can now prove NP-hardness of the binary bottom-up tokenisation gap problem by reducing 3-OCC-MAX2SAT's gap problem to it. To this end, we rely on a reduction identical to $\text{R2}(\mathcal{X}, \mathcal{C}, \gamma)$, but where we define:

$$\delta^- = 5398J + 575 + 3I - \gamma^- \qquad\qquad \delta^+ = 5398J + 575 + 3I - \gamma^+ \tag{104}$$
$$= 5398J + 575 + 3I - \frac{2011 + \varepsilon}{2016}I, \qquad = 5398J + 575 + 3I - \frac{2012 - \varepsilon}{2016}I$$

Lemmas 5 and 6 trivially show the validity of this reduction:

$$3\text{OM2S}(\mathcal{X}, \mathcal{C}, (\gamma^-, \gamma^+)) \iff \text{Tok}_\uparrow^2(\mathcal{D}, K, (\delta^-, \delta^+)) \tag{105}$$

which holds since $3\text{OM2S}(\mathcal{X}, \mathcal{C}, \gamma^+) \iff \text{Tok}_\uparrow^2(\mathcal{D}, K, \delta^+)$ and the same for $\gamma^-$ and $\delta^-$. It is therefore NP-hard to distinguish whether a dataset can be compressed to at most $5398J + 575 + 3I - \frac{2012-\varepsilon}{2016}I$ symbols, or if at least $5398J + 575 + 3I - \frac{2011+\varepsilon}{2016}I$ symbols remain (with an allowed vocabulary size $K = 10J$). Since each variable occurs exactly three times in any 3-OCC-MAX2SAT instance, we have that $\frac{3}{2}J = I$. We now compute a lower bound on the best achievable compression ratio:

$$\frac{\delta^-}{\delta^+} = \frac{5398J + 575 + 3I - \frac{2011+\varepsilon}{2016}I}{5398J + 575 + 3I - \frac{2012-\varepsilon}{2016}I} \tag{106a}$$

$$= \frac{10805J + 575 - \frac{6033+3\varepsilon}{2016}J}{10805J + 575 - \frac{6036-3\varepsilon}{2016}J} \tag{106b}$$

$$\geq \frac{10805 - \frac{6033+3\varepsilon'}{2016}}{10805 - \frac{6036-3\varepsilon'}{2016}} \qquad \text{additive 575 omitted in } \varepsilon' \text{ for sufficiently large } J \tag{106c}$$

$$= \frac{7258949 - \varepsilon'}{7258948 + \varepsilon'} \tag{106d}$$

We conclude that binary bottom-up tokenisation cannot be approximated in polynomial time with an approximation ratio better than $\frac{7258949}{7258948} > 1.0000001$, unless P = NP. $\qquad\square$

## J  PROOF THAT UNARY DIRECT TOKENISATION IS IN NP

A decision problem is in the nondeterministic polynomial-time class (NP) if it can be verified in polynomial time in the presence of a **certificate**: a string designed to verify that the current instance is a "yes"-instance, typically encoding an optimal solution to its search problem. When inputs are represented as strings, the following lemma follows trivially from the unbounded-alphabet case discussed by Whittington et al. (2025). Their proof, however, relies on the explicit computation of the direct tokenisation function:

$$\texttt{tok}_{\phi}[\mathcal{S}](\mathbf{c}) = \arg\min_{\mathbf{s}\in\mathcal{S}^*} |\mathbf{s}|, \qquad \text{s.t. } \mathbf{c}\overset{\circ}{=}\mathbf{s} \tag{107}$$

While we can efficiently compute this when inputs are given in string form, this is not known to be the case for the string-length representation. In this case, the optimal application of tokens corresponds to the change-making problem, which is itself weakly NP-hard, as shown by Lueker (1975). Thus, we now include the optimal application of tokens as a part of the certificate, in order to prove that this decision problem is still in NP when inputs are represented as string-lengths.

**Lemma 9.** *The unary direct tokenisation decision problem is in* NP.

*Proof.* We use as certificate a set of string-lengths composing the tokeniser's vocabulary $\mathcal{S}_{\mathbb{N}}$, as well as a set $\mathcal{Z} = \{\mathbf{z}_m\}_{m=1}^M$, where each $\mathbf{z}_m \in \mathbb{N}^{|\mathcal{S}_{\mathbb{N}}|}$ shows how many tokens of each length should be used to tokenise each string in the dataset $\mathcal{D}_{\mathbb{N}}$. Verifying this certificate then simply requires computing the sum of tokens used for each target.

If $|\mathcal{D}_{\mathbb{N}}| \leq K$, each $\ell \in \mathcal{D}_{\mathbb{N}}$ can be included as a token, and thus all entries in our dataset can be compressed into single token; consequently, the certificate can simply be empty and we verify the problem's satisfiability by checking whether $\delta \geq |\mathcal{D}_{\mathbb{N}}|$ holds. Assuming $|\mathcal{D}_{\mathbb{N}}| > K$—and therefore that $K$'s value is polynomial in the input—we have that the certificate also has polynomial length, and that, in particular, the size of $\mathcal{Z}$ is bounded by $|\mathcal{D}_{\mathbb{N}}| K \log \ell_{\max}$, where $\ell_{\max}$ is the maximum string-length in $\mathcal{D}_{\mathbb{N}}$. Thus, all that is left is to compute the sum of $\mathcal{Z}$ and check whether $\sum_{\mathbf{z}\in\mathcal{Z}} \texttt{sum}(\mathbf{z}) \leq \delta$. □

## K  PROOF OF FORWARD STEP OF THEOREM 5

**Lemma 10.** *If a* vertex-cover *instance is satisfiable, then the* D-1-TOK *instance output by Reduction 3 is also satisfiable. Formally:* $\mathrm{VC}(\mathcal{V},\mathcal{E},\psi) \implies \mathrm{Tok}_{\phi}^1(\mathrm{R3}(\mathcal{V},\mathcal{E},\psi))$.

*Proof.* Suppose the given instance of vertex-cover is satisfiable. Then there exists a vertex cover $\mathcal{C}^{\star} \subseteq \mathcal{V}$ which uses $\psi$ vertices. As a consequence, we can choose as tokens those with the following lengths:

$$\mathcal{S}_{\mathbb{N}} = \{\ell_j \mid v_j \in \mathcal{V}\} \cup \{B\} \cup \{\ell_j' \mid v_j \in \mathcal{C}^{\star}\} \tag{108}$$

We have that every vertex-string in $\mathcal{D}_1$ is covered by a single token, either of length $\ell_j$ or $B$.

All $\psi$ cover-strings $a^{\ell_j'}$ in $\mathcal{D}_2$ which encode a vertex that belongs to $\mathcal{C}^{\star}$ are also covered by a single token. The remaining $J - \psi$ cover-strings in $\mathcal{D}_2$ are covered by 2 tokens, as for every target of length $\ell_j' = \texttt{enc}(v_j) + N^3$ there exist the tokens of length $\ell_j = \texttt{enc}(v_j)$ and $B = N^3$.

As $\mathcal{C}^{\star}$ is a vertex cover, we have that for every edge $(v_j, v_{j'}) \in \mathcal{E}$ at least one of the vertices belongs to $\mathcal{C}^{\star}$. It follows that for every edge-string of length $\ell_{j,j'}'' = \texttt{enc}(v_j) + \texttt{enc}(v_{j'}) + N^3$ in $\mathcal{D}_3$, at least one of the tokens of length $\ell_j' = \texttt{enc}(v_j) + B$ or $\ell_{j'}' = \texttt{enc}(v_{j'}) + B$ belongs to $\mathcal{S}_{\mathbb{N}}$. Thus, all edge-strings are covered by two tokens.

We can now count the number of symbols used in each dataset: $\mathcal{D}_1$ uses $J + 1$ symbols; $\mathcal{D}_2$ uses $2J - \psi$ symbols; and $\mathcal{D}_3$ uses $2I$ symbols. This gives us a total of $3J + 2I + 1 - \psi = \delta$ symbols, which satisfies this tokenisation instance. Thus, we have that $\mathrm{Tok}_{\phi}^1(\mathcal{D}, K, \delta) = \texttt{T}$. □

## L  PROOF OF BACKWARD STEP OF THEOREM 5

**Lemma 11.** *If the* D-1-TOK *instance output by Reduction 3 is satisfiable, then the* vertex-cover *instance which generated it is as well. Formally:* $\mathrm{Tok}_{\phi}^1(\mathrm{R3}(\mathcal{V},\mathcal{E},\psi)) \implies \mathrm{VC}(\mathcal{V},\mathcal{E},\psi)$.

*Proof.* Assume this instance $(\mathcal{D}_{\mathbb{N}}, K, \delta)$ of D-1-TOK—where $(\mathcal{D}_{\mathbb{N}}, K, \delta) = \mathrm{R3}(\mathcal{V}, \mathcal{E}, \psi)$—is satisfiable, i.e., that $\mathrm{Tok}_{\hookrightarrow}^{1}(\mathrm{R3}(\mathcal{V}, \mathcal{E}, \psi))$ evaluates to true. We must prove that, in this case, $\mathrm{VC}(\mathcal{V}, \mathcal{E}, \psi)$ also evaluates to true. Now, let $\mathcal{S}_{\mathbb{N}}$ be an arbitrary optimal solution to $(\mathcal{D}_{\mathbb{N}}, K, \delta)$. By construction, we have that $K = J + 1 + \psi$ and $\delta = 3J + 2I + 1 - \psi$. We proceed in four steps:

①  We prove that all target strings in $\mathcal{D}_{\mathbb{N}}$ are unique;

②  We prove that an optimal tokeniser must only include full target strings in its vocabulary;

③  We prove that an optimal tokeniser will include all target strings in $\mathcal{D}_1$ in its vocabulary;

④  We prove that, if an optimal tokeniser achieves compression $\delta$, then the instance of vertex-cover which was reduced to it is satisfiable.

These steps first show that an optimal tokeniser must admit a certain form (Steps ① – ③), and that from this form (Step ④), we can deduce a valid vertex cover, which concludes the proof. □

**LemmaProofStep 1.** (Step ①). *All strings in $\mathcal{D}_{\mathbb{N}}$ are unique.*

*Proof.* Now we show that all strings in $\mathcal{D}_{\mathbb{N}}$ are unique. These strings all have lengths:

$$\ell_{j_1} = \mathrm{enc}(v_{j_1}), \quad B = N^4, \quad \ell'_{j_1} = \mathrm{enc}(v_{j_1}) + B, \quad \ell''_{j_1, j_2} = \mathrm{enc}(v_{j_1}) + \mathrm{enc}(v_{j_2}) + B \tag{109}$$

for $1 \le j_1, j_2 \le J$ and with $\mathrm{enc}(v_{j_1}) = j_1 + j_1^2 N + j_1^3 N^2$. Notably, our reduction defines $N \gg J^3$ and it will be useful to think about these lengths in base $N$. Let a number $(a, b, c, d, e)_N$ denote $aN^4 + bN^3 + cN^2 + dN^1 + e$. For example, we can write:

$$(a, b, c, d, N-1)_N + (a, 0, 0, 0, 1)_N = (2a, b, c, d+1, 0)_N \tag{110}$$

We can similarly write in this base:

$$\ell_{j_1} = (0, 0, j_1^3, j_1^2, j_1)_N, \quad B = (1, 0, 0, 0, 0)_N, \quad \ell'_{j_1} = (1, 0, j_1^3, j_1^2, j_1)_N, \tag{111a}$$

$$\ell''_{j_1, j_2} = (1, 0, j_1^3 + j_2^3, j_1^2 + j_2^2, j_1 + j_2)_N, \quad \ell'_{j_1} + \ell'_{j_2} = (2, 0, j_1^3 + j_2^3, j_1^2 + j_2^2, j_1 + j_2)_N \tag{111b}$$

Two numbers are the same only if each "digit" in this base system is the same. Given this structure, we see that $\ell_{j_1}$ and $B$ are all unique string-lengths. Further, the string-lengths $\ell'_{j_1}$ are all different from one another. It is left to show that: (i) all string-lengths $\ell'_{j_1}$ are different from all $\ell''_{j_1, j_2}$; and (ii) that string-lengths $\ell''_{j_1, j_2}$ are different among themselves. Requirement (i) is proven by SubLemma 2, which shows that there is no set of numbers $j_1, j_2, j_3 \in \mathbb{N}$ for which $j_1 = j_2 + j_3$ and $j_1^2 = j_2^2 + j_3^2$. Requirement (ii) is proven by SubLemma 3, which shows that there is no set of numbers $j_1, j_2, j_3, j_4 \in \mathbb{N}$ for which $j_1 + j_2 = j_3 + j_4$ and $j_1^2 + j_2^2 = j_3^2 + j_4^2$. It follows that all strings in $\mathcal{D}_{\mathbb{N}}$ are unique. □

**LemmaProofStep 2.** (Step ②). *An optimal tokeniser must only include full character-strings in $\mathcal{D}_{\mathbb{N}}$ and compress all other strings to two symbols.*

*Proof.* Note that, since all strings in $\mathcal{D}_{\mathbb{N}}$ are unique, the best compression one could possibly achieve would result from compressing $K$ strings into a single symbol, and the remaining $|\mathcal{D}_{\mathbb{N}}| - K$ to two symbols. As $|\mathcal{D}_{\mathbb{N}}| = 2J + I + 1$, this (hypothetical) optimal compression would lead to:

$$\mathfrak{G}_{\ell}(\mathcal{S}_{\mathrm{opt}}, \mathcal{D}_{\mathbb{N}}) = K + 2(|\mathcal{D}_{\mathbb{N}}| - K) \tag{112a}$$

$$= J + 1 + \psi + 2(2J + I + 1 - J - 1 - \psi) \tag{112b}$$

$$= 3J + 2I + 1 + \psi - \psi) = \delta \tag{112c}$$

As by assumption $\mathrm{Tok}_{\hookrightarrow}^{1}(\mathrm{R3}(\mathcal{V}, \mathcal{E}, \psi))$ evaluates to true, our tokeniser must achieve this compression, and is thus composed of $K$ full strings in $\mathcal{D}_{\mathbb{N}}$. Further, it must compress all other strings to at most two symbols. □

**LemmaProofStep 3.** (Step ③). *An optimal tokeniser selects every vertex-string in $\mathcal{D}_1$ as a token.*

*Proof.* Suppose that some vertex-string of length $\ell_{j_1}$ in $\mathcal{D}_1$ is not chosen as a token. Then $\ell_{j_1}$ must be the sum of two tokens. No tokens of cover- or vertex-strings (in datasets $\mathcal{D}_2$ and $\mathcal{D}_3$) can be used, since such tokens contain a summand $B$, which is significantly larger than $\ell_{j_1}$. Hence, both summands would have to also be vertex-strings $\ell_{j_2}, \ell_{j_3}$. These string-lengths have values:

$$\ell_{j_1} = (1, 0, j_1^3, j_1^2, j_1)_N, \quad \ell_{j_2} = (1, 0, j_2^3, j_2^2, j_2)_N, \quad \ell_{j_3} = (1, 0, j_3^3, j_3^2, j_3)_N \tag{113}$$

Again, by SubLemma 2, it is impossible that $\ell_{j_1} = \ell_{j_2} + \ell_{j_3}$. Thus, no target string in $\mathcal{D}_1$ can be covered by two other tokens; but, as argued in Step ②, the tokeniser may use at most two symbols per target. This concludes the proof that all character-strings in $\mathcal{D}_1$ must be included in the vocabulary $\mathcal{S}_\mathbb{N}$. Further, every cover and edge-string is larger than $B$, while all vertex-strings are significantly smaller than it; $B$ thus cannot be written as two other tokens and must hence also be part of $\mathcal{S}_\mathbb{N}$. $\square$

**LemmaProofStep 4.** (Step ④). *If an optimal tokeniser achieves compression $\delta$, then the original* `vertex-cover` *instance is satisfiable.*

*Proof.* Step ③ shows that $J+1$ tokens in any optimal tokeniser must correspond to the target strings in $\mathcal{D}_1$. Using only these tokens, every target in $\mathcal{D}_2$ needs two symbols, and every target in $\mathcal{D}_3$ needs three symbols. With Step ②, the remaining $\psi$ tokens must correspond to target strings from $\mathcal{D}_2 \cup \mathcal{D}_3$. We are going to show that: a token corresponding to an edge target can only contribute to itself; a token corresponding to a cover target can only contribute to itself and edge targets which include the vertex the cover consists of. Without loss of generality, let $t$ with $0 \le t \le \psi$ of these remaining tokens be edge-strings (from $\mathcal{D}_3$) and let the remaining $\psi - t$ be cover-strings (from $\mathcal{D}_2$).

We are now going to show that a token from $\mathcal{D}_3$ can only improve the solution by reducing its target string to a single token. Pick any of the selected edge tokens, having a length of $\ell''_{j_1, j_2} = (1, 0, j_1^3 + j_2^3, j_1^2 + j_2^2, j_1 + j_2)_N$. For this edge token to contribute to compressing a target string, it must be combined with another token; let its length be $\ell_\oplus$, such that their sum equals the length of one of the target strings already present in the dataset. Since $\ell''_{j_1, j_2}$ already contains a summand $B$, the token $\ell_\oplus$ cannot contain $B$, as any target string length in the dataset is strictly less than $2B$. As established in Step ②, tokens must correspond to target string-lengths themselves. Since any non-vertex target string contains a summand $B$, the additional token $\ell_\oplus$ must therefore be a vertex token, with length $\ell_\oplus = (0, 0, j_\oplus^3, j_\oplus^2, j_\oplus)_N$. Now, assume that the edge token $\ell''_{j_1, j_2}$ is combined with this vertex token $\ell_\oplus$ to compress an arbitrary target from one of the three datasets:

$$\ell_{j_3} = \ell''_{j_1, j_2} + \ell_\oplus \iff \tag{114a}$$
$$(0, 0, j_3^3, j_3^2, j_3)_N = (1, 0, j_\oplus^3 + j_1^3 + j_2^3, j_\oplus^2 + j_1^2 + j_2^2, j_\oplus + j_1 + j_2)_N,$$
$$\ell'_{j_3} = \ell''_{j_1, j_2} + \ell_\oplus \iff \tag{114b}$$
$$(1, 0, j_3^3, j_3^2, j_3)_N = (1, 0, j_\oplus^3 + j_1^3 + j_2^3, j_\oplus^2 + j_1^2 + j_2^2, j_\oplus + j_1 + j_2)_N,$$
$$\ell''_{j_3, j_4} = \ell''_{j_1, j_2} + \ell_\oplus \iff \tag{114c}$$
$$(1, 0, j_3^3 + j_4^3, j_3^2 + j_4^2, j_3 + j_4)_N = (1, 0, j_\oplus^3 + j_1^3 + j_2^3, j_\oplus^2 + j_1^2 + j_2^2, j_\oplus + j_1 + j_2)_N$$

The first case clearly cannot be satisfied, as any vertex target has length strictly smaller than any edge target. Additionally, the two other cases cannot be satisfied per SubLemmas 4 and 5. In other words, this shows that edge-strings cannot contribute to any other target value.

We are now left with $\psi - t$ tokens formed of cover-strings. Recall that using only the tokens obtained from Step ③, every target in $\mathcal{D}_2$ needs two symbols, and every target in $\mathcal{D}_3$ needs three symbols. Having the newly obtained tokens corresponding to a target from $\mathcal{D}_2$ we will show that they can only be applied optimally on: themselves, resulting in one symbol used; target strings from $\mathcal{D}_3$, reducing the symbols to two. Any other application of these tokens would not yield a better compression, as improving compression is only possible if the token obtained from $\mathcal{D}_2$ compresses a target from $\mathcal{D}_2 \cup \mathcal{D}_3$ other than itself to a single token. But Step ① shows that all target strings are unique, meaning a token cannot reduce any target string apart from itself to a single symbol.

Finally, these selected tokens must be used, in conjunction with vertex-strings, to compress all the remaining edge-strings to two tokens. Note that only composing a cover and a vertex-string can compress an edge-string to two symbols:

$$\ell_{j_1} + \ell_{j_2} = (0, 0, j_1^3 + j_2^3, j_1^2 + j_2^2, j_1 + j_2)_N, \tag{115a}$$

$$\ell'_{j_1} + \ell'_{j_2} = (2, 0, j_1^3 + j_2^3, j_1^2 + j_2^2, j_1 + j_2)_N, \tag{115b}$$

$$\ell'_{j_1} + \ell_{j_2} = (1, 0, j_1^3 + j_2^3, j_1^2 + j_2^2, j_1 + j_2)_N \tag{115c}$$

Additionally, an edge-string can only be compressed by a cover- or vertex-string subword which contains exactly the vertices that the edge consists of. Assume there exists another set of vertex- and cover-strings such that their tokens $\ell_{j_1}, \ell_{j_2}$ can compress an edge-string consisting of different vertices $\ell''_{j_3, j_4}$. Then we have that:

$$\ell''_{j_3, j_4} = \ell_{j_1} + \ell_{j_2} \iff (1, 0, j_3^3 + j_4^3, j_3^2 + j_4^2, j_3 + j_4)_N = (1, 0, j_1^3 + j_2^3, j_1^2 + j_2^2, j_1 + j_2)_N \tag{116}$$

From SubLemma 3 it follows that this cannot be the case. This means that, for each edge-string of length $\ell''_{j_1, j_2}$ not in our tokeniser, we must have a subword (of length either $\ell'_{j_1}$ or $\ell'_{j_2}$) which "covers" it to obtain our target compression. Consider thus, the subgraph $(\mathcal{V}, \mathcal{E}')$, where:

$$\mathcal{E}' = \mathcal{E} \setminus \{(v_{j_1}, v_{j_2}) \in \mathcal{E} \mid \ell''_{j_1, j_2} \notin \mathcal{S}_{\mathbb{N}}\} \tag{117}$$

There exists a vertex cover of size $\psi - t$ for this subgraph composed of vertices $\{v_j \in \mathcal{V} \mid \ell'_j \in \mathcal{S}_{\mathbb{N}}\}$. Now, if we expand this set of $\psi - t$ vertices by picking one arbitrary vertex, $v_{j_1}$ or $v_{j_2}$, for each edge-string of length $\ell''_{j_1, j_2}$ in our vocabulary, we get a cover $\mathcal{C} = \{v_j \in \mathcal{V} \mid \ell'_j \in \mathcal{S}_{\mathbb{N}}\} \cup \{v_{j_1} \mid \ell''_{j_1, j_2} \in \mathcal{S}_{\mathbb{N}}\}$ of size at most $\psi$ for the original graph. Thus, it follows that $\mathrm{VC}(\mathcal{V}, \mathcal{E}, \psi) = \mathtt{T}$. $\square$

## L.1 Proofs that String-lengths in Reduction 3 are Unique

We now show the technical sublemmas used in the previous proof.

**SubLemma 2.** *For any $r \in \mathbb{N}$, there do not exist non-zero $i, j \in \mathbb{N}$ such that:*

$$i + j = r, \qquad i^2 + j^2 = r^2 \tag{118}$$

*Proof.* From $(i + j)^2 = i^2 + 2ij + j^2$ and the equations $i + j = r$ and $i^2 + j^2 = r^2$, we obtain:

$$r^2 = i^2 + j^2 + 2ij = r^2 + 2ij \Rightarrow ij = 0, \tag{119}$$

contradicting $i, j > 0$. $\square$

**SubLemma 3.** *There do not exist two distinct pairs $\{i, j\} \neq \{a, b\}$ of positive integers such that:*

$$i + j = a + b, \qquad i^2 + j^2 = a^2 + b^2 \tag{120}$$

*Proof.* Due to $i + j = a + b$, there is an integer $s$ such that:

$$a = i - s, \qquad b = j + s \tag{121}$$

Then, replacing $a$ and $b$ with their corresponding expressions from Eq. (121), we obtain:

$$a^2 + b^2 - (i^2 + j^2) = (i - s)^2 + (j + s)^2 - i^2 - j^2 = 2s^2 + 2s(j - i) \tag{122}$$

By the lemma statement $a^2 + b^2 = i^2 + j^2$, so $2s^2 + 2s(j - i) = 0$, i.e.:

$$s(s + j - i) = 0 \tag{123}$$

Hence either $s = 0$ or $s = i - j$. If $s = 0$, then $a = i$ and $b = j$. If $s = i - j$, then $a = i - (i - j) = j$ and $b = j + (i - j) = i$. In both cases, it follows that $\{a, b\} = \{i, j\}$, contradicting distinctness. $\square$

**SubLemma 4.** *For any $r \in \mathbb{N}$, there do not exist non-zero $i, j, k \in \mathbb{N}$ such that:*

$$i + j + k = r, \qquad i^2 + j^2 + k^2 = r^2 \tag{124}$$

*Proof.* Using $(i + j + k)^2 = i^2 + j^2 + k^2 + 2(ij + ik + jk)$ and the given equations:

$$r^2 = r^2 + 2(ij + ik + jk) \implies ij + ik + jk = 0 \tag{125}$$

With $i, j, k > 0$, each product $ij, ik, jk$ is positive, which yields a contradiction. Hence, no solution exists. $\square$

**SubLemma 5.** *Let $r, p \in \mathbb{N}$. There do not exist non-zero $i, j, k \in \mathbb{N}$ such that:*

$$i + j + k = r + p, \qquad i^2 + j^2 + k^2 = r^2 + p^2, \qquad i^3 + j^3 + k^3 = r^3 + p^3 \qquad (126)$$

*Proof.* Let $p_1 = i + j + k$, $p_2 = i^2 + j^2 + k^2$, $p_3 = i^3 + j^3 + k^3$, and $e_1 = i + j + k$, $e_2 = ij + ik + jk$, $e_3 = ijk$. From the first two equations:

$$e_1 = r + p, \qquad e_2 = \frac{p_1^2 - p_2}{2} = \frac{(r+p)^2 - (r^2+p^2)}{2} = rp \qquad (127)$$

Newton's identity for three variables gives:

$$p_3 = e_1 p_2 - e_2 p_1 + 3 e_3 \qquad (128)$$

Substituting $p_1 = r + p$, $p_2 = r^2 + p^2$, $e_2 = rp$ yields:

$$p_3 = (r + p)(r^2 + p^2) - rp(r + p) + 3e_3 = (r + p)(r^2 + p^2 - rp) + 3e_3 = r^3 + p^3 + 3e_3 \qquad (129)$$

By the third equation in the lemma statement, $p_3 = r^3 + p^3$; hence $3e_3 = 0$ and so $e_3 = ijk = 0$, contradicting $i, j, k > 0$. $\qquad \square$

## M   DEFINITION OF THE ADDITION CHAIN PROBLEM

An addition chain is a sequence of integers that provides an efficient way to "build" a target set of numbers starting from 1.

**Definition 5.** *Let $\mathbf{t} = \{t_1, t_2, \ldots, t_J\}$ be a finite set of positive integer targets. An **addition chain** for $\mathbf{t}$ is a sequence of integers $\mathbf{b} = \langle b_0, b_1, \ldots, b_R \rangle$ with the following properties:*

1. *The sequence starts with $b_0 = 1$.*

2. *Every subsequent element $b_i$ is the sum of two preceding elements:*

$$b_i = b_j + b_k, \quad \text{for some } 0 \le k \le j < i \qquad (130)$$

3. *The sequence contains all targets: for every $t_j \in \mathbf{t}$, there is some $b_r \in \mathbf{b}$ such that $t_j = b_r$.*

4. *By convention, the **length** of the chain $\mathbf{b}$ is $R$.*

**Definition 6.** *Let $\mathbf{t}$ be a set of positive integers. Given a maximum length $\zeta$, the **addition chain decision problem (Add-Chain)** requires deciding whether there exists an addition chain for $\mathbf{t}$ with length at most $\zeta$. The **addition chain optimisation problem** is to find the minimal length for such an addition chain.*

We denote by $\mathrm{AddChain}(\mathbf{t}, \zeta)$ a function which returns T if such a chain exists (meaning the Add-Chain instance is satisfied), and F otherwise.

## N   PROOF THAT THE UNARY OPE DECISION PROBLEM IS (AT LEAST) WEAKLY NP-COMPLETE

**Theorem 6.** *The unary optimal pair encoding decision problem is (at least) weakly NP-complete.*

*Proof.* We write $\mathrm{Tok}^1_{\mathrm{OPE}}(\mathcal{D}, K, \delta)$ for a function which returns T if its input corresponds to a satisfiable instance of the OPE-1-TOK decision problem, and F otherwise. To prove weak NP-completeness, we must show that the problem is in NP and that it is weakly NP-hard. Inclusion in NP was already established by Kozma and Voderholzer (2024). We prove weak NP-hardness via a reduction from the Add-Chain decision problem. First, we define this reduction.

**Reduction 4.** *Given an instance of Add-Chain consisting of a targets $\mathbf{t} = \{t_1, \ldots, t_J\}$ and a length limit $\zeta$, we construct an instance of the OPE-1-TOK problem with the following parameters. The dataset $\mathcal{D}$ is the set of unary strings corresponding to the targets: $\mathcal{D} = \{a^{t_1}, a^{t_2}, \ldots, a^{t_J}\}$; the merge budget is set to the addition chain length limit: $K = \zeta$; and the token count threshold is set to the number of targets $\delta = J$.*

Note that setting the threshold $\delta$ to the number of strings in the dataset implies that a valid solution must represent every string as a single token. The proof proceeds in two parts, showing both directions of the equivalence:

$$\text{AddChain}(\mathbf{t}, \zeta) \iff \text{Tok}_{\text{OPE}}^1(\mathcal{D}, K, \delta) \tag{131}$$

**Forward Step** ($\text{AddChain}(\mathbf{t}, \zeta) \implies \text{Tok}_{\text{OPE}}^1(\mathcal{D}, K, \delta)$). We first show that a solution to the Add-Chain problem implies a solution to the OPE-1-TOK problem. Assume there exists a valid addition chain $\mathbf{b}^\star = \langle b_0, b_1, \ldots, b_R \rangle$ of length $R \leq \zeta$ for the target set $\mathbf{t}$. By definition, for each element $b_r^\star \in \mathbf{b}^\star$ (where $r \geq 1$), there exist indices $r', r'' < r$ such that $b_r^\star = b_{r'}^\star + b_{r''}^\star$. We construct a merge sequence $\mathbf{m} = \langle m_1, \ldots, m_R \rangle$ of length $R$ where each merge is defined as $m_r = (a^{b_{r'}^\star} \odot a^{b_{r''}^\star})$, corresponding to the predecessors of $b_r$ in the addition chain. The length of this merge sequence $\mathbf{m}$ is $R \leq \zeta$, satisfying the merge budget $K = \zeta$. By the iterative definition of the merge-extracted vocabulary, the resulting vocabulary $\mathcal{S}_\mathbf{m} = \{a^{b_{r'}^\star + b_{r''}^\star} \mid m_r \in \mathbf{m}\}$ will contain a token $a^{b_r^\star}$ for every element $b_r^\star$ in the addition chain $\mathbf{b}^\star$. Since the addition chain $\mathbf{b}^\star$ contains all targets $t_j \in \mathbf{t}$, the vocabulary $\mathcal{S}_\mathbf{m}$ is guaranteed to contain a single token for each target string $a^{t_j} \in \mathcal{D}$. Consequently, the direct encoding function $\text{tok}_{\looparrowright}[\mathcal{S}_\mathbf{m}]$ can represent each string $\mathbf{c} \in \mathcal{D}$ with exactly one token. The total token count is therefore:

$$\sum_{\mathbf{c} \in \mathcal{D}} |\text{tok}_{\looparrowright}[\mathcal{S}_\mathbf{m}](\mathbf{c})| = \sum_{\mathbf{c} \in \mathcal{D}} 1 = |\mathcal{D}| \tag{132}$$

By the construction used in our reduction, $|\mathcal{D}| = \delta$. The condition is met, thus proving the implication.

**Backward Step** ($\text{Tok}_{\text{OPE}}^1(\mathcal{D}, K, \delta) \implies \text{AddChain}(\mathbf{t}, \zeta)$). Next, we show that a solution to the OPE-1-TOK problem implies a solution to the Add-Chain problem. Assume there exists a merge sequence $\mathbf{m}_{\text{opt}}$ of length $K = R \leq \zeta$ that satisfies the OPE-1-TOK decision problem. The condition is:

$$\sum_{\mathbf{c} \in \mathcal{D}} |\text{tok}_{\looparrowright}[\mathcal{S}_\mathbf{m}](\mathbf{c})| \leq \delta \tag{133}$$

where, by the reduction's construction, we have $\delta = |\mathcal{D}|$. Since the tokenisation of any string must contain at least one token, this sum is also lower-bounded by $|\mathcal{D}|$. Therefore, the inequality must hold with equality, which is only possible if every string is tokenised into exactly one token:

$$|\text{tok}_{\looparrowright}[\mathcal{S}_\mathbf{m}](\mathbf{c})| = 1 \quad \text{for all } \mathbf{c} \in \mathcal{D} \tag{134}$$

This implies that for every target $t_j \in \mathbf{t}$, the corresponding string $a^{t_j}$ must exist as a single token in the merge-extracted vocabulary, i.e., $a^{t_j} \in \mathcal{S}_\mathbf{m}$. Now, construct an addition chain from $\mathcal{S}_\mathbf{m}$ as: $\mathbf{b} = \langle \ell_r \mid a^{\ell_r} \in \mathcal{S}_\mathbf{m} \rangle$. The construction of $\mathcal{S}_\mathbf{m}$ from merges guarantees that $\mathbf{b}$ is a valid addition chain.[9] Since $\mathcal{S}_\mathbf{m}$ contains all strings $a^{t_j}$, then $\mathbf{b}$ must contain all targets $t_j \in \mathbf{t}$. Further, $\mathbf{b}$ was constructed from $K \leq \zeta$ merges. There is thus an addition chain for $\mathbf{t}$ of length at most $\zeta$, which concludes the proof. $\quad\square$

---

[9] If any subword produced by a non-reachable merge exists in $\mathcal{S}_\mathbf{m}$, it should be pruned from $\mathbf{b}$. A non-reachable merge is a merge $m_r = (a^{\ell_{r'}} \odot a^{\ell_{r''}})$ whose pair of subwords $a^{\ell_{r'}}$ and $a^{\ell_{r''}}$ cannot be generated.

