# OpenReview forum: "Tokenisation over Bounded Alphabets is Hard"
_ICLR.cc/2026/Conference — ICLR 2026 Poster_

### Official Review · Reviewer_gmyW · 2025-10-19

**Soundness:** 3
**Presentation:** 3
**Contribution:** 3
**Rating:** 8
**Confidence:** 3

**Summary:**

This paper explores the computational complexity of tokenisation when restricted to bounded alphabets, addressing a critical gap in prior research that focused on unbounded alphabets.
The authors analyze both direct and bottom-up tokenisation and demonstrate that these problems remain NP-complete even for binary and unary alphabets. Furthermore, they prove that neither admits a polynomial-time approximation scheme (PTAS) unless P = NP.
Through reductions from classic NP-hard problems (such as 3-OCC-MAX2SAT and Vertex Cover), the paper rigorously establishes the intractability of tokenisation even under practical, real-world constraints.

**Strengths:**

The connection drawn between tokenisation and classical NP-hard problems is conceptually elegant and insightful.

The paper tackles a relevant and previously unresolved problem, making a clear theoretical contribution to both computational complexity and NLP.

The authors thoughtfully discuss implications for practical tokenisation algorithms, encouraging further research on approximation and relaxation methods.

**Weaknesses:**

The discussion of approximation hardness constants (e.g., very tight lower bounds) could be expanded to give more intuition on their practical implications.

**Questions:**

Could the authors clarify whether their hardness results extend to probabilistic or heuristic tokenisation schemes commonly used in NLP (e.g., BPE variants with stochastic merges)?

Do the reductions rely crucially on compression-based objectives, or might similar hardness results hold for alternative objectives like frequency balancing or entropy minimization?

Given the established hardness, do the authors foresee any provably efficient approximation algorithms for special cases (e.g., very small datasets or restricted merge operations)?

---

> ### Author Response · Authors · 2025-11-25
>
> We thank the reviewer for their feedback and we are glad they appreciated our paper. We respond to each point in detail below.
>
> > W1. The discussion of approximation hardness constants (e.g., very tight lower bounds) could be expanded to give more intuition on their practical implications.
>
> We thank the reviewer for their suggestion, and we will expand this discussion.
>
> > Q1. Could the authors clarify whether their hardness results extend to probabilistic or heuristic tokenisation schemes commonly used in NLP (e.g., BPE variants with stochastic merges)?
>
> In general, a randomised tokeniser (such as BPE dropout) cannot compress a text better than an optimal deterministic tokeniser. This is because there exists one (or multiple) optimally compressed subword-strings per sentence, and randomly choosing between these cannot improve compression. As such, finding an optimal stochastic tokeniser is as hard as finding a deterministic one (at least as it regards compression). These randomisations, however, may still improve downstream language modelling performance (due to, e.g., regularising the model). Thinking about how to capture those effects by designing new objective functions for tokenisation would be interesting.
>
> > Q2. Do the reductions rely crucially on compression-based objectives, or might similar hardness results hold for alternative objectives like frequency balancing or entropy minimization?
>
> The specific reductions rely on compression-based objectives, which are easier to analyse as they are quite local in nature. Adding a merge to a bottom-up tokeniser affects the compression on the merge’s characters, but not on unrelated characters. Adding a merge to a bottom-up tokeniser, though, may affect the unigram probability of all tokens, which makes proofs based on unigram log-probability objectives (such as used by UnigramLM) more involved. However, we believe that similar results should hold for these other objectives, and we are actively trying to solve this as well.
>
> > Q3. Given the established hardness, do the authors foresee any provably efficient approximation algorithms for special cases (e.g., very small datasets or restricted merge operations)?
>
> We are not sure whether such algorithms exist, but we are actively working to find out (for better or worse)!
>
> Please let us know if we misunderstood any of your questions, or if you have other ones.

---

### Official Review · Reviewer_Zs3e · 2025-10-22

**Soundness:** 3
**Presentation:** 2
**Contribution:** 2
**Rating:** 2
**Confidence:** 4

**Summary:**

This paper addresses the computational complexity of bottom-up and direct tokenisation over bounded alphabet. The authors develop elaborate proofs to show that this problem is NP-complete even with alphabets consisting only of 2 characters. Also they show that there is no polynomial time approximation unless P=NP. The main optimization function is compression. The paper contains elaborate formal definitions of the problems addressed and provides elaborate technical details underlying the main results.

**Strengths:**

The paper presents rigorous proofs for tokenisation over bounded alphabets with compression as the optimization function.

**Weaknesses:**

In my opinion, this paper will not be of interest to the main ICLR community and is only of theoretical interest with no clear practical impact on learning or NLP.

**Questions:**

How does this work fit under learning representation?
There are already heuristics to perform tokenisation which work very well in practice with no theoretical guarantees. How does this work impact the existing vast literature?

---

> ### Author Response · Authors · 2025-11-25
>
> We thank the reviewer for their feedback. We respond to each point in detail below.
>
> > W1. In my opinion, this paper will not be of interest to the main ICLR community and is only of theoretical interest with no clear practical impact on learning or NLP.
> >
> > Q1. How does this work fit under learning representation?
>
> Tokenisation is a technique that is subject to active research, and existing tokenisation algorithms are often blamed for many common problems with LLMs (Rust et al., 2021; Toraman et al., 2023; Ali et al., 2024; Lesci et al., 2025). Thus, the community still actively tries to find new, practical algorithms (Liu et al., 2025; Chizhov et al., 2025). For this, it is important to understand the problem at hand and which techniques might be suitable to solve it. For example, following the previous results on NP-hardness, it would be very reasonable to think about tokenisation algorithms that are parameterised by the size of the alphabet used by the tokenised strings. However, our results show that this cannot result in efficient algorithms (unless P=NP), thus practitioners need not waste their time on such approaches. In fact, we ourselves did waste quite some time on such unsuccessful attempts, before proving they were fruitless. While our results might not directly inform us on how to improve tokenisation, they do inform us about how it cannot be improved, which is (or at least should be) of interest to practitioners.
>
>
> > Q2. There are already heuristics to perform tokenisation which work very well in practice with no theoretical guarantees. How does this work impact the existing vast literature?
>
> We respectfully disagree that the existence of heuristics which work relatively well should be reason enough to stop trying to improve them. In fact, a similar statement can be made for the entire literature around machine learning: large language models work well enough in practice, but we shouldn’t stop trying to improve transformers, optimisation algorithms, etc. Finding optimal tokenisers could be one such potential direction for improvement. We prove, however, that this would not be computationally viable. Overall, we thus believe that our work motivates the use of heuristic methods in the literature, and informs NLP practitioners of an avenue which should perhaps not be pursued.
>
> Please let us know if we misunderstood any of your questions, or if you have other ones.
>
> References:
>
>
> * Phillip Rust, Jonas Pfeiffer, Ivan Vulic, Sebastian Ruder, and Iryna Gurevych. 2021. How good is your tokenizer? On the monolingual performance of multilingual language models. ACL 2021 https://aclanthology.org/2021.acl-long.243
> * Cagri Toraman, Eyup Halit Yilmaz, Furkan Şahinuç, and Oguzhan Ozcelik. 2023. Impact of tokenization on language models: An analysis for Turkish. ACM Trans. Asian Low-Resour. Lang. Inf. Process., 22(4). https://dl.acm.org/doi/10.1145/3578707
> * Mehdi Ali, Michael Fromm, Klaudia Thellmann, Richard Rutmann, Max Lübbering, Johannes Leveling, Katrin Klug, Jan Ebert, Niclas Doll, Jasper Buschhoff, Charvi Jain, Alexander Weber, Lena Jurkschat, Hammam Abdelwahab, Chelsea John, Pedro Ortiz Suarez, Malte Ostendorff, Samuel Weinbach, Rafet Sifa, Stefan Kesselheim, and Nicolas FloresHerr. 2024. Tokenizer choice for LLM training: Negligible or crucial? NAACL 2024
> * Pietro Lesci, Clara Meister, Thomas Hofmann, Andreas Vlachos, and Tiago Pimentel. 2025. Causal Estimation of Tokenisation Bias. ACL 2025 https://aclanthology.org/2025.acl-long.1374 https://aclanthology.org/2024.findings-naacl.247
> * Alisa Liu, Jonathan Hayase, Valentin Hofmann, Sewoong Oh, Noah A. Smith, Yejin Choi. SuperBPE: Space Travel for Language Models https://arxiv.org/abs/2503.13423
> * Pavel Chizhov, Catherine Arnett, Elizaveta Korotkova, Ivan P. Yamshchikov. BPE Gets Picky: Efficient Vocabulary Refinement During Tokenizer Training https://arxiv.org/abs/2409.04599

---

### Official Review · Reviewer_bxh6 · 2025-10-30

**Soundness:** 4
**Presentation:** 4
**Contribution:** 3
**Rating:** 8
**Confidence:** 4

**Summary:**

The authors consider the problem of tokenization: given a database of words, how can one most efficiently tokenize the words? Tokenization is an important first step in most natural language processing pipelines, and finding the optimal tokenization is potentially useful to optimizing language model performance. Thus, it is important to understand how well we can hope to compute good tokenizations. Formally the authors consider two problems:

1. Direct encoding: given a set of tokens, the tokenization of a database is the optimal splitting of the words in the database to minimize the number of tokens used.
2. Bottom-up encoding: given a set of tokens, the tokenization of a database is represented by a set of merge operations. The tokenized database is obtained by sequentially applying the merge operations greedily to the database.

For both formulations of the tokenization function, the optimization problem asks to find the tokenization (i.e. the set of tokens in the former case, the merge operations in the second case) minimizing the number of tokens. Naturally, one can also consider approximations of this problem. The main result of this work establishes that unless P = NP, there is no PTAS for tokenization i.e. there exists a constant 1.00001 such that finding a 1.00001-approximation to the optimal tokenization is NP-hard. The proofs follow via reduction to a special case of the MAX-2-SAT problem. Whereas previous work obtained lower bound for unbounded alphabets, this work gives the first hardness result for bounded alphabets, and in fact gives a strong result ruling out polynomial algorithms for binary alphabets. Furthermore, they show that for (1) even unary tokenization is NP-complete.

The authors consider an interesting problem, and give strong results. The paper is well written and easy to follow. I therefore recommend accept.

**Strengths:**

The paper studies a practically motivated problem which is a key step in training natural language processing models. The result is strong and essentially resolves the question of tokenization with compression as an objective. The paper is well written, and the proofs are well motivated and easy to follow.

**Weaknesses:**

No clear weaknesses (see minor comments below)

**Questions:**

Minor Comments

Reference to Lemma 1 and 2 - I think better to reference Reduction 1 and Theorem 2

Maybe good to state explicitly what all merge operations are in forward step of bottom-up tokenization proof.

---

> ### Author Response · Authors · 2025-11-25
>
> We thank the reviewer for their feedback and we are glad they appreciated our paper. We respond to the reviewer’s comments below.
>
> > Q1. Reference to Lemma 1 and 2 - I think better to reference Reduction 1 and Theorem 2
>
> We have updated the text to refer directly to our reductions and theorems, instead of the lemmas, as suggested.
>
> > Q2.  Maybe good to state explicitly what all merge operations are in forward step of bottom-up tokenization proof.
>
> We assume the reviewer’s suggestion is to provide them explicitly in the proof sketch (in the main text), since this is already provided in our full proof in the appendix. We agree that this would make the sketch easier to follow, and we will do so as suggested.
>
> Please let us know if we misunderstood any of your questions, or if you have other ones.

---

> > ### Comment · Reviewer_bxh6 · 2025-11-25
> >
> > Thanks for the rebuttal! I remain positive about the paper.
> >
> > Regarding practicality and whether ICLR is a good fit (as brought up by other reviewers) I am not experienced in empirical work so will defer to others. However, I believe that theoretical work about a key step in the learning process is valuable. Even if the results are negative, they shed light on why current tokenization algorithms are heuristic and may explain why some practical trade offs are necessary.

---

### Official Review · Reviewer_GzVs · 2025-10-30

**Soundness:** 3
**Presentation:** 3
**Contribution:** 3
**Rating:** 8
**Confidence:** 4

**Summary:**

This paper investigates the tokenization problem, which is the first step in most NLP pipeline. The problem (somewhat involved to describe) is the following: Let $\Sigma$ be a finite alphabet  and let $D = \{c_1, c_2, \dots, c_M\}$ be a dataset of character strings
$c_i \in \Sigma^{*}$. A *tokenizer* is a tuple $(S,dt,t)$ where $S$ is a *vocabulary* which is a finite set $S \subset \Sigma^{+}$ of nonempty substrings over $\Sigma$. $dt$ (detokenizer) is a function from $S^{\*}$ to $\Sigma^{\*}$ which is a string concatenation operator. The tokenizer encoder unction $t$ is a function from $\Sigma^{\*}$ to $S^{\*}$ such that  for any $c \in \Sigma^{\*}$, $c = \mathrm{concat}(t(c))$.  Fix a natural encoding function (eg direct encoding) and a budget $K$. For a given vocabulary $S$ so that $|S| = K+|\Sigma|$, the cost of $S$ is the minimum,  total number of tokens produced over all the dataset $D$ -- i.e., achieves maximal compression over $D$. The optimization problem is to find cost of the best-cost vocabulary of size $K+\Sigma$.

The paper considers two types of encodings -- direct encoding and bottom-up encoding, which are used in practice and investigated in the literature. Prior work has shown that these optimization problems are NP-hard when the alphabet size is unbounded. The present work builds on this hardness result and  strengthen them in several ways. They show that in both cases, even if the alphabet is binary, these problems remain NP-complete. More interestingly to me, they show that even approximating the optimum value up to arbitrary accuracy (PTAS) is NP-hard. In other words, there exists a constant $\varepsilon > 0$ such that no polynomial-time algorithm can approximate the optimum value within a factor of $(1+\varepsilon)$ unless P=NP. The paper leaves open a very interesting theoretical question, is there a constant ratio approximation algorithm for this problem? Concretely, is there an approximation algorithm that finds a vocabulary whose cost is  almost twice the optimum?

**Strengths:**

The computational problem studied in this paper is highly relevant, and, given the prior work, the demonstrated impossibility of approximation algorithms with arbitrary precision represents a significant theoretical contribution. The results provide valuable insights into the computational complexity of a fundamental step in modern AI and NLP models. The paper is clearly structured and written fairly well. While I did not verify all proofs in detail (as they are presented in the appendix), the claims appear sound and consistent with the constructions outlined in the main text. I also find the open question regarding the existence of a constant-factor approximation algorithm particularly intriguing—and I even suspect that the answer might turn out to be negative.

**Weaknesses:**

While theoretically it is an interesting paper, it appears like the practical tokenizers work very well and not clear whether these results will have any impact on the progress of modern NLP models. Another weakness I find is that it appears complicated than it needs to be to define the computational problems. Some notational use appears non-standard. For example while $tok$ is a function by definition, they also use it as a set and use notations such as $|tok|$ which, to my understanding is the cardinality of the vocabulary. I find such notional use a bit confusing. Another minor weakness is that, the paper is dense and proof-heavy with limited examples illustrating the reductions. Hence a small running example of the reduction would improve readability.

**Questions:**

From a theoretical viewpoint, It will be very nice to see the exact constant beyond which one cannot approximate. You give a bound 1.000002, which is theoretically sufficient for the claim, but practically a 1.000002 approximation algorithm is very good.  Have you considered improving the constant? What is your insight as to the existence of constant factor approximation algorithm?

**Details Of Ethics Concerns:**

No Concern.

---

> ### Author Response · Authors · 2025-11-25
>
> We thank the reviewer for their feedback and we are glad they appreciated our paper. We respond to each point in detail below.
>
> > W1. While theoretically it is an interesting paper, it appears like the practical tokenizers work very well and not clear whether these results will have any impact on the progress of modern NLP models.
>
> While we agree that practical tokenisers work very well (which could also be said about LLMs’ architectures, optimisation processes, etc), we also believe that they can be improved further and hope that our work acts as a pointer on how this could be done. In particular, we believe the design space of tokenisation algorithms has not been explored comprehensively enough for us to be fully satisfied with the current state of affairs. That said, we also believe that our paper motivates—to a certain extent—the current reliance we have on approximation algorithms, like BPE, as finding optimal tokenisers would be intractable.
>
> > W2. Another weakness I find is that it appears complicated than it needs to be to define the computational problems. Some notational use appears non-standard. For example while tok  is a function by definition, they also use it as a set and use notations such as |tok|  which, to my understanding is the cardinality of the vocabulary. I find such notional use a bit confusing.
>
> Thank you for this valuable feedback.  While we agree the paper may be a bit “notation heavy”, we tried to simplify exposition as much as possible while keeping precision and formalisation, and we will try to improve this further. We also agree that the use of |tok| to measure the tokeniser’s size is non-standard and we will try to clarify that in our manuscript.
>
> > W3. Another minor weakness is that, the paper is dense and proof-heavy with limited examples illustrating the reductions. Hence a small running example of the reduction would improve readability.
>
> We agree a running example would help make the reductions clearer. This is non-trivial, however, as some of our reductions rely on different base problems. We tried to brainstorm a few ideas so far, but without success. We will keep trying to do so. If you have a more concrete suggestion of how this should look, we'll definitely try to incorporate it.
>
> > Q1. From a theoretical viewpoint, It will be very nice to see the exact constant beyond which one cannot approximate. You give a bound 1.000002, which is theoretically sufficient for the claim, but practically a 1.000002 approximation algorithm is very good. Have you considered improving the constant?
>
> Small improvements to this constant should be relatively easy to achieve. Some of our proofs, for instance, rely on showing contradictions when an optimality condition is not met. In those proofs, we were somewhat “generous” with the compression achieved by the “counterfactual” cases, to make the proofs less involved (and easier for us to manage). We then set the multiplicities $f$ to be sufficiently large to ensure the optimality conditions were satisfied. By being generous with the counterfactual cases, however, we indirectly inflated the multiplicities $f$, which impacts this constant. It should be relatively easy to reduce those multiplicities and thus lift the lower bound.
>
> However, significantly improving the resulting bound is a significant challenge, which we considered, but unsuccessfully so far. The main difficulty stems from the nature of our reduction, which relies on the above-mentioned multiplicities $f$. These must be large enough to guarantee that optimal tokenisers correspond to the solutions of the reduced problem. To further lift the lower bound, we believe a significantly different proof strategy would be necessary. While we did attempt reductions from other problems, we were not able to find satisfying solutions yet, largely due to these same encoding constraints. We believe that a more promising next step in this direction would be not necessarily to improve this constant, but to show that no constant-factor approximation exists, as we argue in the next paragraph.
>
> > Q2. What is your insight as to the existence of constant factor approximation algorithm?
>
> Our intuition is that a constant-factor approximation probably doesn't exist, as the reviewer also suspects. We base this on our own early attempts. We tried to bound the approximation ratios of simple, greedy-like approaches, for the unary direct case, and we quickly found dataset constructions which served as counterexamples, where the approximation ratio significantly increased. Even for vocabulary sizes of just two or three (i.e., for $K=2$ or $K=3$), the greedy-like choices led to significantly worse results than the optimal one. Our feeling is that, if it's this hard to get a constant bound for the simple unary direct case, it's highly unlikely that one exists for the more complex binary cases.
>
>
> Please let us know if we misunderstood any of your questions, or if you have other ones.

---

### Official Review · Reviewer_jKLj · 2025-11-01

**Soundness:** 3
**Presentation:** 3
**Contribution:** 3
**Rating:** 6
**Confidence:** 3

**Summary:**

The paper investigates the computational complexity of tokenisation when applied to inputs drawn from bounded alphabets, addressing a gap in earlier NP-completeness results that assumed alphabets of unbounded size. Tokenisation, a core component of natural language processing pipelines, converts character strings into subword sequences. Many tokenisation methods—such as Byte Pair Encoding (BPE) and Unigram Language Models—aim to compress data, thereby improving efficiency in model training and inference. While previous work had established that finding an optimally compressive tokeniser is NP-complete, those proofs relied on the unrealistic assumption of infinitely large alphabets.

The authors define and analyse two bounded-alphabet tokenisation problems: bottom-up tokenisation, where an optimal sequence of merge operations must be selected, and direct tokenisation, where the optimal vocabulary must be chosen directly. They demonstrate that even when the alphabet is extremely limited, the problems remain computationally intractable. Specifically, for binary alphabets (only two characters), both bottom-up and direct tokenisation are shown to be NP-complete and to lack any polynomial-time approximation scheme, unless P = NP. The authors also show that direct tokenisation remains NP-complete even for unary alphabets (containing a single symbol). Because unary and binary alphabets are the simplest cases, these hardness results automatically extend to all larger alphabets.

The findings imply that the difficulty of tokenisation does not stem from large alphabets or complex merge strategies, but from the inherent structure of the optimisation problem itself. Consequently, it is unlikely that any efficient algorithm can find or even closely approximate an optimal tokeniser under a compression objective.

**Strengths:**

The main strength of the paper is that it closes a gap in earlier NP-completeness results, which assumed alphabets of unbounded size. The current paper shows that these hardness results hold even for small size alphabets.

**Weaknesses:**

The scope of the paper may be more suitable for a conference on computational complexity. On the other hand the results are about an important problem in natural language processing. Therefore it may fit a section dedicated to computational complexity results within natural language processing.

**Questions:**

No question

---

> ### Author Response · Authors · 2025-11-25
>
> We thank the reviewer for their feedback and we are glad they appreciated our paper. Regarding the pointed-out weakness:
>
> > W1: The scope of the paper may be more suitable for a conference on computational complexity.
>
> Although our results are theoretical, they directly shed light on why all popular tokenisation algorithms rely on heuristics, providing the first formal proof (for bounded alphabet sizes) that the problem is intractably hard. This means that the widespread use of approximation methods, like BPE, isn’t just a practical convenience or compromise, but a fundamental necessity. We believe this result is thus important for the NLP community (as the reviewer says, it is about an important problem in natural language processing) and builds on work published at ACL, which is why we have decided to submit our paper to ICLR.

---

### Author Response · Authors · 2025-12-04
**Discussion Summary**

Dear area chair, we would like to first thank you and the five reviewers for their careful reading of our paper. Here is a short summary of our view of the discussion period and how we will integrate the reviewers’ suggestions.

First, we are happy that four out of the five reviewers appreciated our paper and believe it should be presented at ICLR. They highlight the importance of the problem we tackle, the strength and rigour of our proofs, the easiness to read our paper, and our discussion of our results’ impact. The main weakness noted by the reviewers (modulo some small requests for clarifications) is the relevance of our paper to ICLR; this was noted as a small weakness by jKLj and as an important weakness by Zs3e. We respectfully disagree with this perspective: while being a theoretical result, our work sheds light on an important area of NLP and strengthens past results on the topic (one of which was published at an NLP conference,  ACL). As four of the reviewers also highlight the importance of tokenisation for NLP, we believe that we have a strong claim that at least part of the ICLR community would be interested in our work. Further, we note that even the negative reviewer kindly described our proofs as rigorous and elaborate.

For an updated version of our paper, we are working to further increase its readability, cleaning up some small notation issues highlighted by the reviewers, and using the increased page limit to include some more explanations and helpful details. Further, we are currently trying to construct an intuitive running example for the reductions, which would be included in our manuscript.

Thank you again for your time,

The authors

---

### Meta-Review · Area_Chair_DJdu · 2025-12-19

**Summary:**

The paper concerns proving NP-hardness of (approximate and exact) optimization of compressed length objective for two formalizations of the problem: bottom-up and direct tokenization. Relative to prior work, this paper considers bounded alphabet sizes, closing a gap in the literature.

The primary weaknesses of the paper were pointed out by reviewers Zs3e and jKLj --- regarding the practical impact of the paper, as well as the appropriateness of ICLR as a venue. This is a worry I share as well: Schmidt et al '24, an empirical paper the authors cite, quite convincingly demonstrate that compression is unlikely to be the main property of a good tokenization. Given this, it's unclear what the hardness of optimal compression suggests for developing new tokenizers.

On a purely technical level, the results seem sound and clearly written, though it's not very easy for a reader to understand what are the salient differences in the reductions from Kozma and Voderholzer & Whittington et al. (The proof sketches make it seem like the constructions follow a similar idea.)  Also on a technical level, as the authors note, different compression-like objectives may behave differently with respect to the approximation factor. Given that, it seems to me a more reasonable notion of "optimality" is closeness of the encoders themselves to the optimal encoder.

**Reviewer Concerns:**

Practical impact, and appropriateness of the venue.

**Reviewer Scores:**

I don't think the scores would have changed.

---

### Decision · Program_Chairs · 2026-01-26

Accept (Poster)